# GLOBAL CONVERGENCE IN NEURAL ODEs: IMPACT OF ACTIVATION FUNCTIONS

**Tianxiang Gao**
DePaul University
tgao9@depaul.edu

**Siyuan Sun**
Iowa State University
sxs14473@iastate.edu

**Hailiang Liu**
Iowa State University
hliu@iastate.edu

**Hongyang Gao**
Iowa State University
hygao@iastate.edu

## ABSTRACT

Neural Ordinary Differential Equations (ODEs) have been successful in various applications due to their continuous nature and parameter-sharing efficiency. However, these unique characteristics also introduce challenges in training, particularly with respect to gradient computation accuracy and convergence analysis. In this paper, we address these challenges by investigating the impact of activation functions. We demonstrate that the properties of activation functions—specifically smoothness and nonlinearity—are critical to the training dynamics. Smooth activation functions guarantee globally unique solutions for both forward and backward ODEs, while sufficient nonlinearity is essential for maintaining the spectral properties of the Neural Tangent Kernel (NTK) during training. Together, these properties enable us to establish the global convergence of Neural ODEs under gradient descent in overparameterized regimes. Our theoretical findings are validated by numerical experiments, which not only support our analysis but also provide practical guidelines for scaling Neural ODEs, potentially leading to faster training and improved performance in real-world applications.

## 1 INTRODUCTION

In recent years, deep neural networks have achieved remarkable success across a wide range of applications. Among these advancements, Neural Ordinary Differential Equations (ODEs) (Chen et al., 2018b) stand out due to their continuous nature and parameter efficiency through shared parameters. Unlike conventional neural networks with discrete layers, Neural ODEs model the evolution of hidden states as a continuous-time differential equation, allowing them to better capture dynamic systems. This parameter-sharing mechanism ensures consistent dynamics throughout the continuous transformation and reduces the number of parameters, improving both memory efficiency and computational complexity. These unique properties make Neural ODEs particularly effective not only for traditional machine learning tasks like image classification (Chen et al., 2018b) and natural language processing (Rubanova et al., 2019), but also for more complex tasks involving continuous processes, such as time series analysis (Kidger et al., 2020), reinforcement learning (Du et al., 2020), and diffusion models (Song et al., 2020). However, while these features offer flexibility and efficiency, they also introduce significant challenges during training, particularly in gradient computation and convergence analysis.

One of the key challenges in training Neural ODEs is accurately computing gradients. Unlike traditional networks, where backpropagation can be computed through a discrete chain of layers, Neural ODEs require solving forward and backward ODEs using numerical solvers. These solvers introduce numerical errors, which can lead to inaccurate gradients and slow convergence or even suboptimal model performance (Rodriguez et al., 2022). Moreover, ensuring the well-posedness of ODE solutions during training is nontrivial. According to the Picard–Lindelöf Theorem, solutions may not always exist or may only exist locally, which can cause training divergence or significant numerical errors (Gholami et al., 2019; Ott et al., 2020; Sander et al., 2022). Even with advanced

solvers (Zhuang et al., 2020a;b; Matsubara et al., 2021; Ko et al., 2023), it remains an *open* problem whether simple first-order methods, such as stochastic gradient descent (SGD), can reliably train Neural ODEs to convergence. While discretizing Neural ODEs as finite-depth networks offers a potential solution, it results in a deeper computation graph (Zhuang et al., 2020a;b), raising questions about whether the gradients computed in this manner truly match those of the continuous model.

Another essential challenge lies in analyzing the training dynamics of Neural ODEs. The optimization problem in training neural networks is inherently nonconvex, making theoretical analysis difficult. Recent work by Jacot et al. (2018) has shown that the training dynamics of overparameterized networks can be understood through the lens of the Neural Tangent Kernel (NTK), which converges to a deterministic limit as network width increases. This convergence has enabled researchers to establish global convergence guarantees for gradient-based methods in overparameterized regimes, provided the NTK remains *strictly positive definite (SPD)* (Du et al., 2019a; Allen-Zhu et al., 2019; Nguyen, 2021; Gao et al., 2021). The analysis of the NTK's strict positive definiteness began with Daniely et al. (2016), who introduced the concept of dual activation for two-layer networks, later extended to deeper, finite networks (Jacot et al., 2018; Du et al., 2019a). However, these results are limited to networks with discrete layers, raising the question of whether the same properties hold for continuous models like Neural ODEs.

In this paper, we address these challenges by exploring the impact of activation functions on training Neural ODEs. We show that activation function properties—specifically, smoothness and nonlinearity—play critical roles in determining the well-posedness of ODE solutions and the spectral properties of the NTK. Through our analysis, we demonstrate that smooth activation functions lead to globally unique solutions for both forward and backward ODEs, ensuring the stability of the training process. Additionally, we extend existing results on the NTK from discrete-layered neural networks to continuous models, demonstrating that the NTK for Neural ODEs is well-defined. Importantly, we find that a higher degree of nonlinearity in the activation function not only helps maintain the SPD property of the limiting NTK, but also practically speeds up Neural ODE convergence.

1. We investigate the significance of the smoothness of activation functions for the well-posedness of forward and backward ODEs in Neural ODEs. Using random matrix theory, we demonstrate the existence of globally unique solutions. Additionally, we show that no additional regularity is needed if forward and backward ODEs are combined in a weakly coupled ODE system.

2. We propose a new mathematical framework for studying continuous models from the approximation theory perspective. By using a *sequence* of finite-depth neural networks to approximate Neural ODEs, we show that key properties like activation and gradient propagation are preserved as depth approaches infinity. This allows us to apply the Moore-Osgood theorem from functional analysis to prove that the NTK of Neural ODEs is well-defined.

3. Unfortunately, the SPD property of the NTK may not hold at infinite depth, even we can show every finite-depth approximation satisfies it. To address this, we conduct a fine-grained analysis and derive an *integral form* for the limiting NTK of Neural ODEs. This form reveals that the NTK remains SPD if the activation function is non-polynomial. Leveraging this integral representation provides valuable insights into continuous models and may inspire further research.

4. We conduct a series of numerical experiments to support our theoretical findings. Beyond validating our analysis, these experiments also provide *practical* guidelines for training Neural ODEs. We show that activation function smoothness and nonlinearity accelerate convergence and improve performance (see Figure 7-8). Conversely, improper ODE scaling leads to damping from accumulated numerical errors (see Figure 3), while adaptive solvers struggle with efficiency in large-scale Neural ODEs (see Figure 14), causing instability and high computational overhead.

## 2 PRELIMINARIES

### 2.1 NEURAL ODES

In this paper, we consider a simple Neural ODE $f(\boldsymbol{x}; \boldsymbol{\theta})$ [1] defined as follows

$$f(\boldsymbol{x}; \boldsymbol{\theta}) = \frac{\sigma_v}{\sqrt{n}} \boldsymbol{v}^T \phi(\boldsymbol{h}_T), \tag{1}$$

---

[1]The general form of Neural ODEs is discussed in Appendix I.

where $\boldsymbol{h}_t \in \mathbb{R}^n$ is the hidden state that satisfies the following ordinary differential equation

$$\boldsymbol{h}_0 = \sigma_u \boldsymbol{U} \boldsymbol{x} / \sqrt{d}, \quad \text{and} \quad \dot{\boldsymbol{h}}_t = \sigma_w \boldsymbol{W} \phi(\boldsymbol{h}_t) / \sqrt{n}, \quad \forall t \in [0, T], \tag{2}$$

where $\phi$ is the activation function[2], $\boldsymbol{x} \in \mathbb{R}^d$ is input, $\boldsymbol{U} \in \mathbb{R}^{n \times d}$, $\boldsymbol{W} \in \mathbb{R}^{n \times n}$, and $\boldsymbol{v} \in \mathbb{R}^n$ are learnable parameters. These parameters, denoted by $\boldsymbol{\theta} := \mathrm{vec}(\boldsymbol{U}, \boldsymbol{W}, \boldsymbol{v})$, are randomly initialized (Glorot & Bengio, 2010; He et al., 2015) from standard Gaussian distribution:

$$\boldsymbol{U}_{ij}, \quad \boldsymbol{W}_{ij}, \quad \boldsymbol{v}_i \overset{i.i.d.}{\sim} \mathcal{N}(0, 1), \tag{3}$$

with variance hyperparameters $\sigma_u, \sigma_w, \sigma_v > 0$. Due to the continuous nature of Neural ODEs, computing gradients through standard backpropagation is not feasible. Instead, we use the adjoint method (Chen et al., 2018b) to compute gradients by solving the backward ODE:

$$\boldsymbol{\lambda}_T = \sigma_v \mathrm{diag}(\phi'(\boldsymbol{h}_t)) \boldsymbol{v} / \sqrt{n}, \quad \text{and} \quad \dot{\boldsymbol{\lambda}}_t = -\sigma_w \mathrm{diag}(\phi'(\boldsymbol{h}_t)) \boldsymbol{W}^T \boldsymbol{\lambda}_t / \sqrt{n}, \tag{4}$$

where $\boldsymbol{\lambda}_t$ is the adjoint state. When both forward and backward ODEs are well-posed, we can compute the gradients of $f_{\boldsymbol{\theta}}$ with respect to (w.r.t.) the parameters $\boldsymbol{\theta}$ as follows

$$\nabla_{\boldsymbol{v}} f(\boldsymbol{x}; \boldsymbol{\theta}) = \frac{\sigma_v}{\sqrt{n}} \phi(\boldsymbol{h}_t), \ \nabla_{\boldsymbol{W}} f(\boldsymbol{x}; \boldsymbol{\theta}) = \int_0^T \frac{\sigma_w}{\sqrt{n}} \boldsymbol{\lambda}_t \phi(\boldsymbol{h}_t)^\top dt, \ \nabla_{\boldsymbol{U}} f(\boldsymbol{x}; \boldsymbol{\theta}) = \frac{\sigma_u}{\sqrt{d}} \boldsymbol{\lambda}_0 \boldsymbol{x}^\top. \tag{5}$$

Further details on these derivations are provided in Appendix B. In Section 3, we demonstrate that if $\phi$ is Lipschitz continuous, the forward and backward ODEs have globally unique solutions. Additionally, in Section 5, we prove this well-posedness holds throughout the entire training process.

## 2.2 Neural Tangent Kernel

Given a training dataset $\{(\boldsymbol{x}_i, y_i)\}_{i=1}^N$, the objective is to learn a parameter $\boldsymbol{\theta}$ that minimizes the empirical loss:

$$L(\boldsymbol{\theta}) = \sum_{i=1}^N \frac{1}{2} (f(\boldsymbol{x}_i; \boldsymbol{\theta}) - y_i)^2 = \frac{1}{2} \|\boldsymbol{u} - \boldsymbol{y}\|^2, \tag{6}$$

where $\boldsymbol{u} = (u_1, \cdots, u_N)$ is the prediction vector with $u_i = f(\boldsymbol{x}_i; \boldsymbol{\theta})$, and $\boldsymbol{y} = (y_1, \cdots, y_N)$ is the output vector. Gradient descent with a learning rate $\eta > 0$ is used to minimize the loss:

$$\boldsymbol{\theta}^{k+1} = \boldsymbol{\theta}^k - \eta \nabla_{\boldsymbol{\theta}} L(\boldsymbol{\theta}^k). \tag{7}$$

Following (Du et al., 2019a) and (Jacot et al., 2018), under some required conditions, the evolution of the prediction vector $\boldsymbol{u}^k$ can be approximated as follows:

$$\boldsymbol{u}^{k+1} - \boldsymbol{y} \approx (\boldsymbol{I} - \eta \boldsymbol{H}^k)(\boldsymbol{u}^k - \boldsymbol{y}), \tag{8}$$

where $\boldsymbol{H}^k \in \mathbb{R}^{N \times N}$ is a Gram matrix defined through the NTK (Jacot et al., 2018):

$$K(\boldsymbol{x}, \bar{\boldsymbol{x}}; \theta) := \langle \nabla_\theta f(\boldsymbol{x}; \boldsymbol{\theta}), \nabla_\theta f(\bar{\boldsymbol{x}}; \boldsymbol{\theta}) \rangle. \tag{9}$$

The training dynamics Eq. (8) is governed by the spectral property of the NTK Gram matrix $\boldsymbol{H}^k$. If there exists a strictly positive constant $\lambda_0 > 0$ s.t. $\lambda_{\min}(\boldsymbol{H}^k) \geq \lambda_0$ for all $k$, then the residual $\boldsymbol{u}^k - \boldsymbol{y}$ decreases to zero exponentially, provided the learning rate $\eta > 0$ is sufficiently small (Allen-Zhu et al., 2019; Nguyen, 2021). As the parameters $\boldsymbol{\theta}^k$ are updated during training, the NTK $K_{\boldsymbol{\theta}}$ changes over time, making its spectral analysis challenging. Fortunately, previous research (Yang, 2020; Jacot et al., 2018) shows that the time-varying $K_{\boldsymbol{\theta}}$ converges to a deterministic limiting NTK $K_\infty$ as the network width $n \to \infty$. By leveraging this result and the concept of dual activation (Daniely et al., 2016), we can study the spectral properties of $K_{\boldsymbol{\theta}}$ during training through $K_\infty$ using perturbation theory.

However, these prior results apply only to finite-depth neural networks, while Neural ODEs are infinite-depth networks due to their continuous nature. Moreover, as prior analyses are based on induction techniques, there is no guarantee that these essential properties will also hold as depth tends to infinity. In Section 4.1 and Section 4.2, we introduce a new framework to study Neural ODEs as infinite-depth networks. We demonstrate that the smoothness of the $\phi$ is crucial to ensure these essential properties hold in Neural ODEs. Additionally, to study the spectral properties of the limiting NTK $K_\infty$, we provide a fine-grained analysis by expressing the limiting NTK of Neural ODEs in an integral form, which we conclude that the nonlinearity of activation plays a critical role in ensuring the strict positive definiteness of $K_\infty$.

---

[2]To simplify the analysis, we assume the activation threshold value for $\phi$ is at 0 and further $\phi(0) = 0$.

# 3 WELL-POSEDNESS OF NEURAL ODEs AND ITS GRADIENTS

As continuous models, Neural ODEs pose a significant challenge in accurately computing gradients. In this section, we explore the challenges associated with two methods for computing gradients: *optimize-then-discretize* and *discretize-then-optimize* (Gholami et al., 2019; Finlay et al., 2020; Onken et al., 2021). Through our exploration, we emphasize the essential role of smoothness in activation functions to guarantee the well-posedness of Neural ODEs and their gradients.

## 3.1 OPTIMIZE-THEN-DISCRETIZE METHOD

As discussed in Section 2, Neural ODEs require numerical ODE solvers to solve the forward and backward ODEs Eq. (2) and Eq. (4) to compute the gradients, employing a method known as the *optimize-then-discretize* method. When solving ODEs, ensuring their well-posedness is of primary concern. In Proposition 1, we demonstrate that if $\phi$ is Lipschitz continuous, the forward and backward ODEs have globally unique solutions, thus ensuring the well-posedness. The detailed proofs are provided in Appendix C.

**Proposition 1.** *For any given $T > 0$, if the activation function $\phi$ is Lipschitz continuous with Lipschitz constant $L_1$, then the forward ODE Eq. (2) and the backward ODE Eq. (4) have unique solutions $h_t$ and $\lambda_t$ for all $t \in [0, T]$ and $x \in \mathbb{R}^d$ almost surely over random initialization Eq. (3). In addition, $\lambda_t(x) = \partial f(x; \theta)) / \partial h_t$ is the solution to the backward ODE.*

Although Neural ODEs and their gradients are well-defined under these conditions, this does not guarantee that gradients can be computed accurately by solving the ODEs numerically. One primary issue is that the magnitudes of the hidden state $h_t$ and adjoint state $\lambda_t$ can grow exponentially over the time horizon $T$, leading to accumulated numerical errors. This issue is illustrated in Figure 3, where long-time horizon leads to damping in the early stages of training. To mitigate this problem, we suggest scaling the dynamics by setting $\sigma_w = \mathcal{O}(1/T)$, which ensures that the magnitudes of $h_t$ and $\lambda_t$ remain mild and independent of $T$. This scaling stabilizes the norms, thereby allowing numerical solvers to produce much more accurate gradient estimates.

Additionally, calculating gradients using Eq. (5) requires storing the values of $h_t$ and $\lambda_t$ at every time step $t \in [0, T]$, which can consume a significant amount of memory in practice. To address this, Chen et al. (2018b) propose solving an augmented backward ODE (defined in Appendix 36 for our setup), which combines an additional gradient state with both the backward ODE and the reversed forward ODE. This approach eliminates the need for storing intermediate states. However, since the hidden state $h_t$ is no longer constant in the augmented ODE, additional regularization conditions on the dynamics are typically required to ensure the stability of the solution. Fortunately, we demonstrate that such regularization conditions are unnecessary for Neural ODEs because the Lipschitz continuity of $\phi$ ensures that $h_t$ is well defined for all $t \in [0, T]$. Therefore, the augmented ODE approach can be used without the need for additional regularization. A detailed discussion of this result can be found in Appendix C.

## 3.2 DISCRETIZE-THEN-OPTIMIZE METHOD

Alternatively, we can discretize the ODE using Euler's method, treating the continuous ODE as a finite-depth Residual Network (ResNet) $f^L(x; \theta)$[3] with shared parameters across layers:

$$f^L(x; \theta) = \frac{\sigma_v}{\sqrt{n}} v^\top \phi(h^L(x)), \tag{10a}$$

$$h^\ell = h^{\ell-1} + \kappa \cdot \frac{\sigma_w}{\sqrt{n}} W \phi(h^{\ell-1}), \quad \forall \ell \in \{1, 2, \cdots, L\} \tag{10b}$$

$$h^0 = \frac{\sigma_u}{\sqrt{d}} U x, \tag{10c}$$

where $\kappa = T/L$ represents the time step. The gradient can then be estimated using backpropagation through the finite depth ResNet $f^L(x; \theta)$, referred to as the *discretize-then-optimize* approach.

---

[3]Here the superscript in $f_\theta^L$ indicates it has $L$ time steps, while actually $f_\theta^L$ has totally $L + 2$ layers.

As a finite-depth ResNet, the gradients of $f^L(\boldsymbol{x}; \boldsymbol{\theta})$ are always well defined. However, it remains an open question whether the gradients of the finite-depth approximation $f^L(\boldsymbol{x}; \boldsymbol{\theta})$ converge to the gradients of the continuous Neural ODE $f(\boldsymbol{x}; \boldsymbol{\theta})$ as the depth $L \to \infty$. In Proposition 2, we demonstrate that the smoothness of $\phi$ ensures this convergence. Thus, in the limit of infinite depth (or infinitesimally small time steps), both the optimize-then-discretize and discretize-then-optimize methods yield the same gradients, provided that the activation function is sufficiently smooth. The detailed proofs are provided in Appendix E.

**Proposition 2.** *Given $\boldsymbol{x} \in \mathbb{R}^d$, if the activation function $\phi$ and its derivative $\phi'$ are $L_1$- and $L_2$-Lipschitz continuous, respectively, the following inequalities hold a.s. over random initialization:*

$$\|\nabla_{\boldsymbol{\theta}} f^L(\boldsymbol{x}) - \nabla_{\boldsymbol{\theta}} f(\boldsymbol{x})\| \le CL^{-1}, \quad \forall \ell \in \{0, 1, \cdots, L\}, \tag{11}$$

*where $C > 0$ is a constant depending only on $L_1$, $L_2$, $T$, $\sigma_v$, $\sigma_w$, $\sigma_u$, and $\|\boldsymbol{x}\|$.*

To further validate our theoretical findings, we conduct experiments that compare training efficiency and gradient accuracy with and without the Lipschitz continuity of $\phi'$. These experiment results, illustrated in Figure 7, demonstrate the necessity of Lipschitz continuity for ensuring smooth gradient computation and achieving faster convergence during training.

## 4 NNGP AND NTK FOR NEURAL ODEs

Understanding how activation and gradient propagate through neural networks is crucial for analyzing their training dynamics and generalization abilities, as emphasized in previous studies (Glorot & Bengio, 2010; Poole et al., 2016; Schoenholz et al., 2017). The frameworks of Neural Network Gaussian Processes (NNGP) (Lee et al., 2018) and Neural Tangent Kernels (NTK) (Jacot et al., 2018), grounded in mean-field theory, provide powerful analytical tools to study these dynamics. In this section, we establish theoretical results that demonstrate the well-defined nature of NNGP and NTK for Neural ODEs and explore their properties with respect to information propagation.

### 4.1 NNGP: FORWARD PROPAGATION OF INPUTS

Previous work has shown that in the infinite-width limit, randomly initialized *finite-depth* neural networks converge to Gaussian processes with mean zero and recursively defined covariance functions, known as NNGP kernels (Neal, 2012; Lee et al., 2018; Daniely et al., 2016; Yang, 2019; Gao et al., 2023; Gao, 2024). When approximating Neural ODEs using a sequence of finite-depth ResNets $f_{\boldsymbol{\theta}}^L$, we can establish the NNGP for $f_{\boldsymbol{\theta}}^L$. Detailed proofs are provided in Appendix D.

**Proposition 3.** *Suppose $\phi$ is $L_1$-Lipschitz continuous. Then, as width $n \to \infty$, the finite-depth neural network $f_{\boldsymbol{\theta}}^L$ defined in Eq. (10) converges in distribution to a centered Gaussian Process with a covariance function or NNGP kernel $\Sigma^{L+1} := C^{L+1,L+1}$ defined recursively:*

$$C^{0,k}(\boldsymbol{x}, \bar{\boldsymbol{x}}) = \delta_{0,k} \frac{\sigma_u^2}{d} \boldsymbol{x}^T \bar{\boldsymbol{x}}, \qquad \forall k \in \{0, 1, \cdots, L+1\} \tag{12}$$

$$C^{\ell,k}(\boldsymbol{x}, \bar{\boldsymbol{x}}) = \sigma_w^2 \mathbb{E}\phi(u^{\ell-1})\phi(\bar{u}^{k-1}), \qquad \forall \ell, k \in \{1, 2, \cdots, L+1\} \tag{13}$$

*where $\kappa = T/L$ and $(u^\ell, \bar{u}^k)$ are centered Gaussian random variables with covariance*

$$\mathbb{E}(u^\ell \bar{u}^k) = C^{0,0}(\boldsymbol{x}, \bar{\boldsymbol{x}}) + \kappa^2 \sum_{i,j=1}^{\ell,k} C^{i,j}(\boldsymbol{x}, \bar{\boldsymbol{x}}), \quad \forall \ell, k \in \{0, 1, \cdots, L\}. \tag{14}$$

Although Proposition 3 shows that a sequence of Gaussian processes (GPs) can be derived for the sequence of $f_{\boldsymbol{\theta}}^L$, this does not necessarily mean that the Neural ODE $f_{\boldsymbol{\theta}}$ will also converge to a Gaussian Process as $L \to \infty$. The challenge lies in the difference between two convergence patterns: *infinite-width-then-depth* and *infinite-depth-then-width*. These often lead to different limits. For example, consider the simple double sequence $a_{n,\ell} := n/(\ell + n)$. This double sequence demonstrates how taking different convergence paths—first in width, then in depth, or vice versa—can yield different results. Specifically, we observe that

$$\lim_{\ell \to \infty} \left( \lim_{n \to \infty} a_{n,\ell} \right) = 1, \quad \text{while} \quad \lim_{n \to \infty} \left( \lim_{\ell \to \infty} a_{n,\ell} \right) = 0.$$

This phenomenon has been noted in several recent studies (Hayou & Yang, 2023; Yang et al., 2024; Gao et al., 2023; Gao, 2024) across various neural network architectures. Specifically, for commonly used neural networks, the two convergence patterns often do not coincide, leading to different limits for infinite-depth networks. Hence, the NNGP correspondence does not generally hold for infinite-depth neural networks. For Neural ODEs, the *infinite-depth-then-width* convergence pattern is more relevant, as Neural ODEs are equivalent to infinite-depth neural networks from the standpoint of numerical discretization. However, the continuous nature and parameter sharing in Neural ODEs present unique challenges that make previous mathematical tools inapplicable directly.

Fortunately, if the activation function $\phi$ is sufficiently smooth, we can show that these two limits *commute*, and both convergence patterns share the same limit. One crucial intermediate result involves proving that the double sequence $\langle \phi(\boldsymbol{h}^L), \phi(\bar{\boldsymbol{h}}^L) \rangle / n$ converges in depth $L$ *uniformly* in width $n$ (almost surely). This uniform convergence ensures that as depth increases, the behavior of the system remains stable regardless of width, which is crucial for showing that the limits commute and establishing the well-posedness of the NNGP for Neural ODEs. The proof relies on Euler's convergence theorem and is provided in Appendix D.

**Lemma 1.** *Let $\phi$ be $L_1$-Lipschitz continuous. For any $\boldsymbol{x}, \bar{\boldsymbol{x}} \in \mathbb{S}^{d-1}$, the double sequence $\langle \phi(\boldsymbol{h}^L), \phi(\bar{\boldsymbol{h}}^L) \rangle / n$ satisfies*

$$\left| \langle \phi(\boldsymbol{h}^L), \phi(\bar{\boldsymbol{h}}^L) \rangle - \langle \phi(\boldsymbol{h}_T), \phi(\bar{\boldsymbol{h}}_T) \rangle \right| / n \leq C L^{-1}, \tag{15}$$

*where $C > 0$ is a constant depending solely on $L_1$, $\sigma_w$, $\sigma_u$, and $T$.*

By combining Lemma 1 and Proposition 3 with Moore-Osgood theorem, as stated in Theorem 8 of Appendix A, we can establish the NNGP correspondence for the Neural ODE $f_{\boldsymbol{\theta}}$.

**Theorem 1.** *Suppose $L_1$-Lipschitz continuous $\phi$. As width $n \to \infty$, the Neural ODE $f_{\boldsymbol{\theta}}$ defined in Eq. (1) converges in distribution to a centered Gaussian process with covariance function $\Sigma^*$ defined as the limit of $\Sigma^L$ given in Proposition 3.*

**Remark 1.** *Thanks to the uniform convergence result established in Lemma 1, the covariance function $\Sigma^L$ converges to $\Sigma^*$ with a rate of $|\Sigma^L(x, \bar{x}) - \Sigma^*(\boldsymbol{x}, \bar{\boldsymbol{x}})| \sim C L^{-1}$. This polynomial rate of convergence preserves the geometry of the input space (Yang & Schoenholz, 2017). This stands in contrast to classical feedforward networks, where the input space geometry often collapses unless the variance hyperparameters are set precisely on the edge of chaos (Poole et al., 2016).*

## 4.2 NTK: BACKPROPAGATION OF GRADIENTS

While NNGP governs the forward propagation of inputs, the NTK (Jacot et al., 2018) governs the backward propagation of gradients. Understanding both is key to comprehending the full dynamics of Neural ODEs during training. As defined for Neural ODEs in Eq. (9), we can also define the NTK $K_{\boldsymbol{\theta}}^L$ for the finite-depth network $f_{\boldsymbol{\theta}}^L$ in Eq. (10) as follows:

$$K^L(\boldsymbol{x}, \bar{\boldsymbol{x}}; \boldsymbol{\theta}) := \left\langle \nabla_{\boldsymbol{\theta}} f^L(\boldsymbol{x}; \boldsymbol{\theta}), \nabla_{\boldsymbol{\theta}} f^L(\bar{\boldsymbol{x}}; \boldsymbol{\theta}) \right\rangle. \tag{16}$$

In the same infinite-width limit, as highlighted in previous works (Jacot et al., 2018; Yang, 2020), the NTK $K_{\boldsymbol{\theta}}^L$ converges to a deterministic kernel $K_{\infty}^L$ that remains constant throughout training. Notably, this deterministic limiting NTK $K_{\infty}^L$ (and $K_{\infty}$ defined in Theorem 2) governs the training dynamics of Neural ODEs under gradient descent.

Below are the results for our setup, with proofs provided in Appendix E.

**Proposition 4.** *Suppose $\phi$ is $L_1$-Lipschitz continuous. Then, as the network width $n \to \infty$, the NTK $K_{\boldsymbol{\theta}}^L$ converges almost surely to a deterministic limiting kernel: $\forall L \geq 0$*

$$K_{\infty}^L(\boldsymbol{x}, \bar{\boldsymbol{x}}) = C^{L+1, L+1}(\boldsymbol{x}, \bar{\boldsymbol{x}}) + \kappa^2 \sum_{\ell, k=1}^{L} C^{\ell, k}(\boldsymbol{x}, \bar{\boldsymbol{x}}) D^{\ell, k}(\boldsymbol{x}, \bar{\boldsymbol{x}}) + C^{0,0}(\boldsymbol{x}, \bar{\boldsymbol{x}}) D^{0,0}(\boldsymbol{x}, \bar{\boldsymbol{x}}), \tag{17}$$

*where $C^{\ell, k}$ are defined in Proposition 3 and $D^{\ell, k}$ are defined recursively:*

$$D^{L,k}(\boldsymbol{x}, \bar{\boldsymbol{x}}) = \sigma_w^2 \mathbb{E} \phi'(u^L) \phi'(\bar{u}^L), \qquad \forall k \in \{0, 1, \cdots, L\}, \tag{18}$$

$$D^{\ell,k}(\boldsymbol{x}, \bar{\boldsymbol{x}}) = \kappa^2 \sum_{i,j=L}^{\ell+1, k+1} D^{i,j}(\boldsymbol{x}, \bar{\boldsymbol{x}}) \mathbb{E}[\phi'(u^i) \phi'(\bar{u}^j)] \qquad \forall \ell, k \in \{1, 2, \cdots, L-1\}. \tag{19}$$

The same problem of different convergence patterns converging to different limits, observed in the NNGP kernel $\Sigma^*$, also arises when computing the NTK of Neural ODEs. While the Lipschitz continuity of $\phi$ enables well-posed forward propagation of inputs in Neural ODEs, *additional regularity* is required for backward propagation of gradients. Specifically, Lipschitz continuity of $\phi'$ is sufficient to ensure uniform convergence in the NTK. With $\phi$ and $\phi'$ both being Lipschitz continuous, we can obtain a uniform convergence result similar to Lemma 1.

**Lemma 2.** *If $\phi$ is $L_1$-Lipschitz continuous and $\phi'$ is $L_2$-Lipschitz continuous, then the following inequality holds almost surely:*

$$\left| K_{\boldsymbol{\theta}}^L(\boldsymbol{x}, \bar{\boldsymbol{x}}) - K_{\boldsymbol{\theta}}(\boldsymbol{x}, \bar{\boldsymbol{x}}) \right| \leq C L^{-1}, \quad \forall \boldsymbol{x}, \bar{\boldsymbol{x}} \in \mathbb{S}^{d-1}, \tag{20}$$

*where $C > 0$ is a constant dependent only on the constants $\sigma_v$, $\sigma_w$, $\sigma_u$, $L_1$, $L_2$, and $T$.*

Combining Lemma 2 with Proposition 4 and Moore-Osgood Theorem 8, we can *interchange* the limits $L$ and $n$ in the double sequence $K_{\boldsymbol{\theta}}^L(\boldsymbol{x}, \bar{\boldsymbol{x}})$ and show that the NTK $K_\theta$ of Neural ODE converges to a deterministic limiting kernel.

**Theorem 2.** *Suppose $\phi$ is $L_1$-Lipschitz continuous and $\phi'$ is $L_2$-Lipschitz continuous. As the network width $n \to \infty$, the NTK $K_{\boldsymbol{\theta}}$ converges almost surely to a deterministic limiting kernel:*

$$K_{\boldsymbol{\theta}} \to K_\infty, \quad as \ n \to \infty, \tag{21}$$

*where $K_\infty$ is the limit of the NTK $K_\infty^L$ defined in Proposition 4, as depth $L \to \infty$.*

**Remark 2.** *Using the uniform convergence from Lemma 2, we observe that $\|K_\infty^L(\boldsymbol{x}, \bar{\boldsymbol{x}}) - K_\infty(\boldsymbol{x}, \bar{\boldsymbol{x}})\| \sim C L^{-1}$. This polynomial convergence not only guarantees that gradients neither explode nor vanish as $L \to \infty$ (Yang & Schoenholz, 2017; Schoenholz et al., 2017), but also implies that the limiting NTK, $K_\infty$, has an **integral form**, as suggested by Eq. (17) and provided in Appendix E.5. This integral form provides a key insight for studying the spectral properties of the NTK $K_\infty$ directly, without relying on the inductive techniques used in previous works.*

## 5 GLOBAL CONVERGENCE ANALYSIS FOR NEURAL ODES

As discussed in Eq. (8), the dynamics of the residual $\boldsymbol{u}^k - \boldsymbol{y}$ under gradient descent can be characterized using the NTK $K_{\boldsymbol{\theta}}$. In the infinite-width limit, as shown in Theorem 2, this time-varying kernel $K_{\boldsymbol{\theta}}$ converges to a deterministic limiting kernel $K_\infty$, provided the activation function $\phi$ is sufficiently smooth. Therefore, in this section, we establish the global convergence of Neural ODEs under gradient descent by examining the spectral property of the NTK $K_{\boldsymbol{\theta}}$ and its limit $K_\infty$.

The limiting NTK $K_\infty$ is a deterministic kernel function, and its spectral properties are key to understanding global convergence. Previous studies (Jacot et al., 2018; Nguyen, 2021) have highlighted that *the strictly positive definiteness (SPD)* of the NNGP kernel $\Sigma^*$ is sufficient to guarantee the SPD property of $K_\infty$. Since $\Sigma^*$ is a component of $K_\infty$ defined in Eq. (9), demonstrating the SPD property of $\Sigma^*$ is critical for proving convergence.

However, prior analyses have relied on inductive proofs for finite-depth neural networks, which are not directly applicable to infinite-depth and continuous networks like Neural ODEs. That is because, as depth increases, information propagation can become trivial (i.e., gradients vanishing or exploding), potentially diminishing the SPD property at the infinite-depth limit (Poole et al., 2016; Schoenholz et al., 2017; Hayou & Yang, 2023). Fortunately, results in Section 4 demonstrated stable information propagation in both forward and backward directions, regardless of the choice of $\sigma_v$, $\sigma_w$, and $\sigma_u$. This allows us to retain the SPD property of the NTK of Neural ODEs as the depth $L$ approaches infinity.

Specifically, recall from Theorem 1 that we can express $\Sigma^*$ as:

$$\Sigma^*(\boldsymbol{x}, \bar{\boldsymbol{x}}) = \mathbb{E}[\phi(u)\phi(\bar{u})],$$

where $(u, \bar{u})$ are centered Gaussian random variables with covariance $S^*(x, \bar{x})$ defined by

$$S^*(\boldsymbol{x}, \bar{\boldsymbol{x}}) = \lim_{L \to \infty} C^{0,0}(\boldsymbol{x}, \bar{\boldsymbol{x}}) + \kappa^2 \sum_{\ell, k=1}^L C^{\ell, k}(\boldsymbol{x}, \bar{\boldsymbol{x}})$$

with $\kappa = T/L$. This expression $S^*$ can be interpreted as a *double integral form* whose explicit form is included in Appendix E.5. By leveraging the results from Section 4, we can derive key properties of $S^*$ in Lemma 3. These properties serve as a fundamental basis for analyzing the SPD properties of the NNGP and NTK.

**Lemma 3.** *For any $\boldsymbol{x}, \bar{\boldsymbol{x}} \in \mathbb{S}^{d-1}$, we have*

1. *$S^*(\boldsymbol{x}, \bar{\boldsymbol{x}})$ is well defined, and $0 < S^*(\boldsymbol{x}, \boldsymbol{x}) = S^*(\bar{\boldsymbol{x}}, \bar{\boldsymbol{x}}) < \infty$,*

2. *$S^*(\boldsymbol{x}, \boldsymbol{x}) \geq S^*(\boldsymbol{x}, \bar{\boldsymbol{x}})$ and the equality holds if and only if $\boldsymbol{x} = \bar{\boldsymbol{x}}$.*

Lemma 3 implies $S^*(\boldsymbol{x}, \boldsymbol{x}) = \Theta(1)$ for all $\boldsymbol{x} \in \mathbb{S}^{d-1}$. This allows us to study the SPD property of the NNGP kernel $\Sigma^*$ using its Hermitian expansion from the perspective of dual activation (Daniely et al., 2016). Detailed analysis and proofs are provided in Appendix F. Additionally, $S^*(\boldsymbol{x}, \boldsymbol{x}) > S^*(\boldsymbol{x}, \bar{\boldsymbol{x}})$ for all $\boldsymbol{x} \neq \bar{\boldsymbol{x}}$ implies that the pathology known as *the loss of input dependence*, observed in other large-depth networks such as feedforward (Poole et al., 2016), ResNet (Hayou & Yang, 2023), and RNN (Gao et al., 2023), does not occur here. This stability results from a combination of several factors, including skip connections, scaling $\kappa$, and smoothness and nonlinearity of $\phi$. With stable information propagation in Neural ODEs, we can use the nonlinearity of $\phi$ to show that the NNGP kernel $\Sigma^*$ and the limiting NTK $K_\infty$ are SPD.

**Proposition 5.** *If $\phi$ is Lipschitz, nonlinear but non-polynomial, then the NNGP kernel $\Sigma^*$ is SPD.*

**Corollary 1.** *Suppose $\phi$ and $\phi'$ are Lipschitz continuous. If $\phi$ is nonlinear but non-polynomial, then the limiting NTK $K_\infty$ is SPD.*

With these results, we can establish the global convergence of Neural ODEs under gradient descent with appropriate assumptions about the activation function $\phi$ and the training data.

**Assumption 1.** *Let $\{\boldsymbol{x}_i, y_i\}_{i=1}^N$ be a training set. Assume*

1. *Training set: $\boldsymbol{x}_i \in \mathbb{S}^{d-1}$ and $\boldsymbol{x}_i \neq \boldsymbol{x}_j$ for all $i \neq j$; $|y_i| = \mathcal{O}(1)$,*

2. *Smoothness: $\phi$ and $\phi'$ are $L_1$- and $L_2$-Lipschitz continuous, respectively,*

3. *Nonlinearity: $\phi$ is nonlinear and non-polynomial.*

Under Assumption 1, we can employ inductive proofs to show that in the overparameterized regime, the parameters $\boldsymbol{\theta}^k$ remain close to their initialization $\boldsymbol{\theta}^0$. This proximity ensures that the Neural ODE and its gradients are well-posed not only at initialization, as proved in Proposition 1, but also throughout the entire training. This consistency in parameter updates enables us to prove that the NTK $K_{\boldsymbol{\theta}}$ retains SPD during training, ensuring that the training errors of Neural ODEs consistently decrease to zero at a linear rate. Detailed analysis and proofs are provided in Appendix G.

**Theorem 3.** *Suppose Assumption 1 holds and the learning rate $\eta$ is chosen such that $0 < \eta \leq 1/\|\boldsymbol{X}\|^2$. Then for any $\delta > 0$, there exists a natural number $n_\delta$ such that for all widths $n \geq n_\delta$ the following results hold with probability at least $1 - \delta$ over random initialization Eq. (3):*

1. *The parameters $\boldsymbol{\theta}^k$ stay in a neighborhood of $\boldsymbol{\theta}^0$, i.e.,*

$$\|\boldsymbol{\theta}^k - \boldsymbol{\theta}^0\| \leq C\|\boldsymbol{X}\|\sqrt{L(\boldsymbol{\theta}_0)}/\lambda_0. \tag{22}$$

2. *The loss function $L(\boldsymbol{\theta}_k)$ consistently decreases to zero at an exponential rate, i.e.,*

$$L(\boldsymbol{\theta}_k) \leq \left(1 - \frac{\eta\boldsymbol{\lambda}_0}{16}\right)^k L(\boldsymbol{\theta}_0), \tag{23}$$

*where $\lambda_0 := \lambda_{\min}(K_\infty) > 0$, and the constant $C > 0$ only depends on $L_1$, $L_2$, $\sigma_v$, $\sigma_w$, $\sigma_u$, and $T$.*

## 6 EXPERIMENTS

In this section, we validate our theoretical findings through several experiments on Neural ODEs. We focus on the approximation errors between the continuous Neural ODE and its finite-depth ResNet approximations, the NTK behavior, and the empirical convergence properties of Neural ODEs under gradient descent. Further experimental details, including additional experiments on smooth vs. non-smooth activations and scaling for long-horizon stability, can be found in the appendix.

**Gradient and Output Approximation by Finite-Depth ResNet.** As established in Section 3, Neural ODE outputs and gradients can be approximated by a finite-depth ResNet, with an error rate that

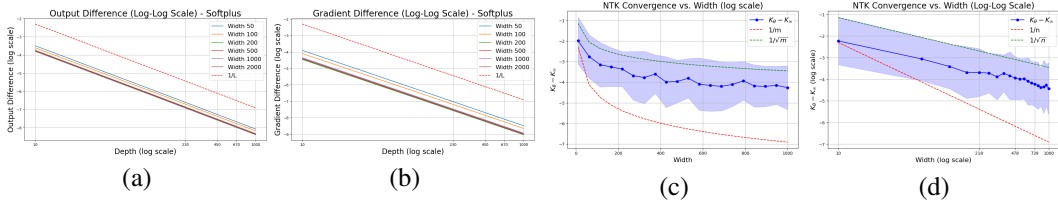

Figure 1: Analysis of Neural ODE output, gradient differences, and NTK convergence. (a) Output differences between Neural ODE and finite-depth ResNet across different widths using Softplus activation. (b) Gradient differences for Neural ODE and ResNet models under Softplus activation. (c) NTK convergence behavior across different widths, showing the NTK approximation converging to the limiting NTK as width increases. (d) NTK convergence behavior on a log-log scale, further emphasizing the rapid convergence at larger widths.

decays as 1/L, where L is the depth of the ResNet. We empirically verify this by measuring the output and gradient differences between the continuous Neural ODE and its finite-depth approximation at initialization. Both the Neural ODE and ResNet were initialized with the same random weights and evaluated on the MNIST dataset, with ResNet depths L ranging from 10 to 1,000. We used Softplus activation to ensure smoothness. Figure 1(a)-(b) demonstrates that the approximation error for both outputs and gradients decreases as 1/L, with convergence being *uniform* across different widths, consistent with our theoretical results. These findings confirm that smooth activation functions lead to well-posed ODE solutions, with accurate approximations by finite-depth networks.

**NTK Approximation Error from Finite-Depth ResNet.** As discussed in Section 4, the NTK of Neural ODEs can be approximated by the NTK of a finite-depth ResNet, with the approximation error decaying as $1/L$. This follows from the fact that the NTK is the inner product of gradients, and as shown in Proposition 2 and Figure 1(a)-(b), the gradient difference between Neural ODEs and finite-depth ResNets also decays as $1/L$. By applying the triangle inequality to the gradient differences, it is straightforward to conclude that the NTK approximation error inherits the same $1/L$ decay rate. Given this reasoning and the page limit, we skip this experiment, but refer readers to Section 4 and Theorem 3 for a detailed theoretical analysis.

**NTK Convergence to Deterministic Limiting NTK.** In Theorem 2, we prove that as the width of Neural ODEs tends to infinity, the NTK converges to a *deterministic* limiting NTK. While no theoretical convergence rate is provided, we conducted experiments to empirically investigate this convergence. We evaluated Neural ODE models with increasing widths, ranging from 10 to $1,000$, and computed the NTK for each width. These NTKs were then compared to an approximate limiting NTK derived from random matrix theory. As shown in Figure 1(c)-(d), the NTK converges to the limiting NTK as the width increases. The empirical convergence rate falls between $1/m$ and $1/\sqrt{m}$, with a tendency closer to $1/\sqrt{m}$ when plotted on a logarithmic scale. This indicates that Neural ODEs exhibit rapid convergence to their limiting NTK, validating the theoretical analysis.

**NTK's SPD and Global Convergence.** In Proposition 5 and Corollary 1, we established that the NTK of Neural ODEs is SPD when the activation function is nonlinear but not polynomial, which guarantees global convergence under gradient descent. Specifically, the NTK's smallest eigenvalue remains positive, ensuring the well-conditioning of the model during training. Additionally, we showed that the model parameters remain close to their initial values during training, further supporting the global convergence claim.

To empirically verify these results, we conducted experiments with Neural ODE models of varying widths—500, 1000, 2000, and 4000—while monitoring both the NTK's smallest eigenvalue and the distance of the model parameters from their initial values over 100 epochs. Softplus was used as the activation function to ensure smoothness and non-polynomial nonlinearity. At each epoch, we computed the smallest eigenvalue of the NTK and the Euclidean distance between the current and initial parameter values.

*The Smallest Eigenvalue of NTK:* As shown in Figure 2(a), we observed that as the width of the Neural ODE increases, the smallest eigenvalue of the NTK becomes larger. For widths greater than the number of training samples (i.e., which is 1000 in our experiments), the smallest eigenvalue remains strictly positive throughout the training process, confirming the NTK's strict positive defi-

niteness and ensuring that the model is well-conditioned for gradient descent. However, for widths smaller than the number of training samples, the smallest eigenvalue becomes negative, indicating poor conditioning at smaller widths.

*Parameter Distance:* The results also confirm that the parameter distance remains stable as training progresses, staying within a manageable bound of $\mathcal{O}(1)$, as shown in Figure 2(b). As the width increases, the parameter distance grows, but the growth remains stable and does not deviate significantly. This supports the theoretical result that the parameters do not stray far from their initialization, ensuring stable training and global convergence.

*Train and Test Loss:* Finally, as depicted in Figure 2(c)-(d), we observed that larger widths lead to faster convergence of the gradient descent. Models with larger widths (2000 and 4000) exhibited lower test losses and faster convergence rates compared to smaller widths (500 and 1000). This behavior demonstrates that larger widths allow the model to generalize better and converge more efficiently during training.

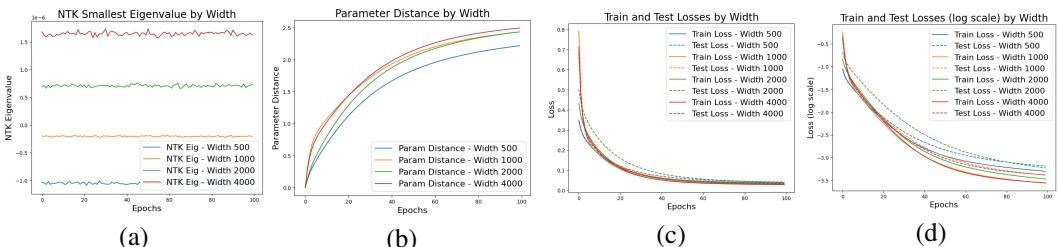

Figure 2: Empirical results of Neural ODEs with varying widths: (a) NTK smallest eigenvalue grows and stabilizes as the width increases, with negative values for widths below the training size. (b) Parameter distances stay stable and bounded within $\mathcal{O}(1)$. (c) Linear-scale train and test losses show faster convergence for larger widths. (d) Log-scale losses further confirm improved generalization for wider models.

**Additional Experimental Results.** In the appendix, we present supplementary experiments that validate and extend our findings. Without proper scaling (e.g., $\sigma_w \sim 1/T$), Neural ODEs exhibit early-stage damping during training over long-time horizons (see Figure 3). Smooth activations like Softplus converge faster than non-smooth ones like ReLU, likely due to more accurate gradient computation (see Figure 7). Additionally, while non-polynomial nonlinearity is sufficient for an SPD NTK, our experiments show that quadratic activations also yield SPD NTKs, though with slower convergence (see Figure 8). These results highlight the importance of activation functions and model design for Neural ODE performance. We also include convergence analysis on diverse datasets, such as CIFAR-10, AG News, and Daily Climate, as well as additional activations like GELU, further demonstrating the generalizability of our findings.

## 7    CONCLUSIONS

In this paper, we examined the crucial role of activation functions in the training dynamics of Neural ODEs. Our findings demonstrate that the choice of activation function significantly impacts the dynamics, stability, and global convergence of the Neural ODE models under gradient descent. Specifically, we found that using smooth activations like Softplus ensures that the forward and backward dynamics in Neural ODEs are well-posed, allowing for accurate approximation by finite-depth ResNets. As a result, the NTK of Neural ODEs converges to a deterministic limiting NTK that governs the model's training dynamics. Additionally, we demonstrated that when using nonlinear but non-polynomial activations, the NTK remains SPD, ensuring well-conditioned training and global convergence. Through extensive experiments, we verified that suitable activation functions, Neural ODEs exhibit stable parameter behavior, rapid NTK convergence, and faster optimization, particularly at larger widths. These findings highlight the importance of selecting activation functions with appropriate smoothness and nonlinearity to ensure the robustness and scalability of Neural ODEs, establishing them as a powerful approach for continuous-time deep learning.

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

## A    USEFUL MATHEMATICAL RESULTS

**Theorem 4** (Bai-Yin law, see Vershynin (2010); Bai & Yin (2008)). *Let $A$ be an $N \times n$ random matrix whose entries and independent copies of a random variable with zero mean, unit variance, and finite fourth moment. Suppose that $N$ and $n$ grow to infinity while the aspect ratio $n/N$ converges to a constant in $[0, 1]$. Then*

$$s_{\min}(A) = \sqrt{N} - \sqrt{n} + o\left(\sqrt{n}\right), \quad s_{\max}(A) = \sqrt{N} + \sqrt{n} + o\left(\sqrt{n}\right), \quad almost\ surely. \quad (24)$$

**Theorem 5** (Picard-Lindelöf theorem). *Let $f : [a, b] \times \mathbb{R}^n \to \mathbb{R}^n$ be a function. If $f$ is continuous in the first argument and Lipschitz continuous with coefficient $L$ in the second argument, then the ODE*

$$\bar{x}(t) = f(t, x(t)), \quad (25)$$

*possesses a unique solution on $[a - \varepsilon, a + \varepsilon]$ for each possible initial value $x(a) = x_0 \in \mathbb{R}^n$, where $\varepsilon < 1/L$.*

**Theorem 6** (Peano Existence Theorem). *If the function $f$ is continuous in a neighborhood of $(t_0, x_0)$, then the ODE equation 25 has at least one solution defined in a neighborhood of $t_0$.*

**Theorem 7** (Convergence for Euler's Method). *Let $x_n$ be the result of applying Euler's method to the ordinary differential equation defined as follows*

$$\dot{x} = f(x, t), \quad t \in [t_0, t_1], \quad and \quad x(0) = x_0. \quad (26)$$

*If the solution $x$ has a bounded second derivative and $f$ is $L$-Lipschitz continuous in $x$, then the global truncation error is bounded by*

$$\|x(t_n) - x_n\| \leq \frac{hM}{2L}(e^{L(t_n - t_0)} - 1), \quad (27)$$

*where $h$ is the time step, and $M$ is an upper bound on the second derivative of $x$ on the given interval.*

**Lemma 4** (Gronwall's inequality). *Let $I = [a, b]$ for an interval such that $a < b < \infty$. Let $u$, $\alpha$, $\beta$ be real-valued continuous functions that satisfies the integral inequality*

$$u(t) \leq \alpha(t) + \int_0^t \beta(s)u(s)ds, \quad \forall t \in I. \quad (28)$$

*Then*

$$u(t) \leq \alpha(t) + \int_0^t \alpha(s)\beta(s)\exp\left(\int_s^t \beta(r)dr\right), \quad \forall t \in I. \quad (29)$$

*If, in addition, $\alpha(t)$ is non-decreasing, then*

$$u(t) \leq \alpha(t)\exp\left(\int_0^t \beta(s)ds\right), \quad \forall t \in I.$$

**Theorem 8** (Moore-Osgood Theorem). *If $\lim_{n\to\infty} a_{n,m} = b_m$ uniformly in $m$, and $\lim_{m\to\infty} a_{n,m} = c_n$ for each $n$, then both $\lim_{m\to\infty} b_m$ and $\lim_{n\to\infty} c_n$ exists and are equal to the double limit, i.e.,*

$$\lim_{m\to\infty}\left(\lim_{n\to\infty} a_{n,m}\right) = \lim_{n\to\infty}\left(\lim_{m\to\infty} a_{n,m}\right) = \lim_{\substack{n\to\infty \\ m\to\infty}} a_{n,m} \quad (30)$$

## B    DERIVATION OF GRADIENT THROUGH ADJOINT METHOD

In this section, we provide the detailed derivation of the adjoint method to compute the gradients. We first recall the forward ODE as follows:

$$\dot{h}_t = \frac{\sigma_w}{\sqrt{n}}W\phi(h_t),$$

$$h_0 = \frac{\sigma_u}{\sqrt{d}}Ux.$$

To compute the gradients, we introduce the Lagrange function

$$\mathcal{L}(\tilde{h}, \theta, \lambda, \mu) = f_\theta(x) + \int_0^T \lambda_t^\top \left( \frac{\sigma_w}{\sqrt{n}} W \phi(\tilde{h}) - \dot{\tilde{h}} \right) dt + \mu^\top \left( \frac{\sigma_u}{\sqrt{d}} Ux - \tilde{h}(0)] \right)$$

where $\tilde{h}$ is an extra variable that are independent from $\theta$ and $(\lambda, \mu)$ are Lagrangian multipliers. Observe that with $\tilde{h} = h$, we have

$$\mathcal{L}(h, \theta, \lambda, \mu) = f_\theta(x), \quad \forall (\lambda, \mu).$$

Thus, the derivatives of $\mathcal{L}$ w.r.t. $\theta$ is equal to gradients of $f_\theta$ w.r.t. $\theta$.

Now, we consider a variation $(\delta h, \delta\theta)$ at point $(h, \theta)$. Then the correspondence variation of $\mathcal{L}$ is given by

$$
\begin{aligned}
\delta\mathcal{L}(h, \theta, \lambda, \mu) =& \frac{\sigma_v}{\sqrt{n}} (\delta v)^\top \phi(h(T)) + \frac{\sigma_v}{\sqrt{n}} v^\top \mathrm{diag}(\phi'(h(T))) \delta h(T) + \mu^\top \left[ \frac{\sigma_u}{\sqrt{d}} (\delta U)x - \delta h(0) \right] \\
& + \int_0^T \lambda^\top \left[ \frac{\sigma_w}{\sqrt{n}} (\delta W)\phi(h) + \frac{\sigma_w}{\sqrt{n}} W \mathrm{diag}(\phi'(h))\delta h - \delta \dot{h} \right] dt \\
=& \frac{\sigma_v}{\sqrt{n}} (\delta v)^\top \phi(h(T)) + \frac{\sigma_v}{\sqrt{n}} v^\top \mathrm{diag}(\phi'(h(T))) \delta h(T) + \mu^\top \left[ \frac{\sigma_u}{\sqrt{d}} (\delta U)x - \delta h(0) \right] \\
& - \lambda^\top \delta h|_0^T + \int_0^T \dot{\lambda}^\top \delta h \, dt + \int_0^T \lambda^\top \left[ \frac{\sigma_w}{\sqrt{n}} (\delta W)\phi(h) + \frac{\sigma_w}{\sqrt{n}} W \mathrm{diag}(\phi'(h))\delta h \right] dt \\
=& \frac{\sigma_v}{\sqrt{n}} (\delta v)^\top \phi(h(T)) + \left[ \frac{\sigma_v}{\sqrt{n}} v^\top \mathrm{diag}(\phi'(h(T))) - \lambda(T)^T \right] \delta h(T) \\
& + \mu^\top \left[ \frac{\sigma_u}{\sqrt{d}} (\delta U)x \right] + (\lambda(0) - \mu)^\top \delta h(0) \\
& + \int_0^T \left[ \dot{\lambda}^\top + \frac{\sigma_w}{\sqrt{n}} \lambda^\top W \mathrm{diag}(\phi'(h)) \right] \delta h \, dt + \int_0^T \frac{\sigma_w}{\sqrt{n}} \lambda^\top (\delta W)\phi(h) dt,
\end{aligned}
$$

where we use integration by parts in the second equality. Then we choose $(\lambda, \mu)$ such that

$$
\begin{aligned}
\mu =& \lambda(0), \\
\lambda(T) =& \frac{\sigma_v}{\sqrt{n}} \mathrm{diag}(\phi'(h(T)))v, \\
\dot{\lambda}(t) =& -\frac{\sigma_w}{\sqrt{n}} \mathrm{diag}(\phi'(h(t)))W^\top \lambda(t).
\end{aligned}
$$

Then the variation of $\mathcal{L}$ becomes

$$\delta\mathcal{L}(h, \theta, \lambda, \mu) = \frac{\sigma_v}{\sqrt{n}} \phi(h(T))^\top \delta v + \frac{\sigma_u}{\sqrt{d}} \mu^\top (\delta U)x + \int_0^T \frac{\sigma_w}{\sqrt{n}} \lambda^\top (\delta W)\phi(h) dt.$$

Thus, we obtain the gradients of $f_\theta$ as

$$
\begin{aligned}
\nabla_v f_\theta(x) =& \frac{\sigma_v}{\sqrt{n}} \phi(h(T)) \\
\nabla_W f_\theta(x) =& \int_0^T \frac{\sigma_w}{\sqrt{n}} \lambda_t \phi(h_t)^\top dt \\
\nabla_U f_\theta(x) =& \frac{\sigma_u}{\sqrt{d}} \lambda(0)x^\top.
\end{aligned}
$$

## C  WELL POSEDNESS OF NEURAL ODES AND ITS GRADIENTS

To show the existence and uniqueness, we first recall the Picard-Lindelöf theorem as follows.

## C.1 FORWARD ODE IS WELL-POSED

As we assume the activation function is Lipschitz continuous, we can immediately obtain the local result that the hidden state $h_t$ exists near the initial time.

**Lemma 5** (Local solution). *If the activation function $\phi$ is $L_1$-Lipschitz continuous, then $h_t$ uniquely exists for all $|t| \leq \varepsilon$, where $\varepsilon < 1/\sigma_w L_1$.*

*Proof.* By using Bai-Yin law 4, we know $\|W\| \sim \sqrt{n}$ a.s. Accordingly, we can show the mapping $f : x \mapsto \frac{\sigma_w}{\sqrt{n}} W \phi(x)$ is Lipschitz continuous:

$$
\begin{aligned}
\|f(x) - f(z)\| =& \|\frac{\sigma_w}{\sqrt{n}} W \phi(x) - \frac{\sigma_w}{\sqrt{n}} W \phi(z)\| \\
\leq & \sigma_w \|\phi(x) - \phi(z)\| \\
\leq & \sigma_w L_1 \|x - z\|.
\end{aligned}
$$

Hence $f$ is $\sigma_w L_1$-Lipschitz continuous a.s. As $t_0 = 0$, it follows from Picard-Lindelöf theorem that unique $h_t$ exists locally for all $|t| \leq \varepsilon$, where $\varepsilon < 1/\sigma_w L_1$. □

**Lemma 6** (Global solution). *For any given $T > 0$, if $\phi$ is $L_1$-Lipschitz continuous, then $h_t$ uniquely exists for all $|t| \leq T$.*

*Proof.* We have shown unique $h_t$ exists locally. Specifically, let $\phi_t(x)$ be the solution flow from initial condition $x$ to the solution at $t$. For any $h_0$, we chose $\varepsilon < 1/\sigma_w L_1$. Then the solution $h_1 := \phi_\varepsilon(x_0)$ is well-defined based on the local solution result. As the dynamics is the same and the Lipschitz coefficient is uniform, we have $h_2 := \phi_\varepsilon(h_1)$ is also well-defined. By repeating this process for any finite steps $N$, we have $h_N = \phi_\varepsilon(h_{N-1})$ is well-defined. Hence, as $T < \infty$, there exist $N$ such that $\varepsilon N \geq T$. Therefore, $h_t$ is well-defined for all $|t| \leq T$ and the desired result is obtained. □

Then the result for global solution simply implies that result in Proposition 1.

## C.2 BACKWARD ODE IS WELL POSED

Recall the backward ODE as follows

$$
\lambda_T = \frac{\sigma_v}{\sqrt{n}} \text{diag}(\phi'(h_T)v, \tag{31}
$$

$$
\dot{\lambda}_t = -\frac{\sigma_w}{\sqrt{n}} \text{diag}(\phi'(h_t))W^\top \lambda_t. \tag{32}
$$

Observe that if $h_t$ is well defined in $t \in [0, T]$, then the dynamics of $\lambda_t$ becomes a linear dynamics. Hence, with similar argument, we can easily show the corresponding VIP of $\lambda_t$ is well posed.

**Lemma 7.** *Given $T$, if the activation function $\phi$ is $L_1$-Lipschitz continuous, then $\lambda_t$ is uniquely determined for all $|t| \leq T$ and $\lambda_t = \partial f_\theta / \partial h_t$ is the solution.*

*Proof.* It follows Lemma 5 and 6 that $h_t$ is well defined for all $t \in [0, T]$ a.s. By Theorem 5, it suffices to show $g : x \mapsto -\frac{\sigma}{\sqrt{n}} \text{diag}[\phi'(h_t)]W^\top x$ is Lipschitz continuous:

$$
\|g(x) - g(z)\| = \|\frac{\sigma_w}{\sqrt{n}} \text{diag}[\phi'(h_t)]W^\top(x - z)\| \leq \sigma_w L_1 \|x - z\|,
$$

where we use the fact $\|W\| \sim \sqrt{n}$ a.s. by Theorem 4 and $|\phi'| \leq L_1$. Hence, the mapping $g$ is $\sigma_w L_1$ Lipschitz continuous. It follows from Theorem 5 that $\lambda_t$ uniquely exist for $t \in [T - \varepsilon, T + \varepsilon]$ for $\varepsilon < 1/\sigma_w L_1$. Then with similar argument, we can show the existence of local solution can be extended to global solution since $\phi$ is uniformly Lipschitz continuous. Therefore, $\lambda_t$ is well defined for all $t \in [0, T]$.

Additionally, we can show $\lambda(t) = \frac{\partial f_\theta(x)}{\partial h(t)}$ is a solution. Specifically, the differential of $f_\theta$ is given by

$$
df_\theta = dv^\top \phi(h(T))/\sqrt{n} = \frac{1}{\sqrt{n}} v^\top \text{diag}(\phi'(h(T)))dh(T).
$$

Then we have

$$\frac{\partial f_\theta(x)}{\partial h(T)} = \frac{1}{\sqrt{n}}\text{diag}(\phi'(h(T)))v. \tag{33}$$

Moreover, for any $\varepsilon > 0$, it follows the chain rule that

$$\frac{\partial f_\theta(x)}{\partial h(t)} = \frac{\partial h(t+\varepsilon)}{\partial h(t)}\frac{\partial f_\theta(x)}{\partial h(t+\varepsilon)}.$$

where we have

$$h(t+\varepsilon) = h(t) + \int_t^{t+\varepsilon} \frac{1}{\sqrt{n}}W\phi(h(s))ds. \tag{34}$$

Then we have

$$\begin{aligned}
\frac{d}{dt}\left(\frac{\partial f_\theta(x)}{\partial h(t)}\right) &= \lim_{\varepsilon\to 0^+} \frac{\frac{\partial f_\theta}{\partial h(t+\varepsilon)} - \frac{\partial f_\theta}{\partial h(t)}}{\varepsilon} \\
&= \lim_{\varepsilon\to 0^+} \frac{\frac{\partial f_\theta}{\partial h(t+\varepsilon)} - \frac{\partial h(t+\varepsilon)}{\partial h(t)}\frac{\partial f_\theta}{\partial h(t+\varepsilon)}}{\varepsilon} \\
&= \lim_{\varepsilon\to 0^+} \frac{\frac{\partial f_\theta}{\partial h(t+\varepsilon)} - \frac{\partial}{\partial h(t)}\left(h(t)+\frac{1}{\sqrt{n}}W\phi(h(t))\varepsilon + \mathcal{O}\left(\varepsilon^2\right)\right)\frac{\partial f_\theta}{\partial h(t+\varepsilon)}}{\varepsilon} \\
&= \lim_{\varepsilon\to 0^+} \frac{\frac{\partial f_\theta}{\partial h(t+\varepsilon)} - \left(I + \frac{\varepsilon}{\sqrt{n}}\text{diag}(\phi'(h(t)))W^\top + \mathcal{O}\left(\varepsilon^2\right)\right)\frac{\partial f_\theta}{\partial h(t+\varepsilon)}}{\varepsilon} \\
&= -\frac{1}{\sqrt{n}}\text{diag}(\phi'(h(t)))W^\top \frac{\partial f_\theta}{\partial h(t)}.
\end{aligned}$$

Thus, we can see $\partial f_\theta(x)/\partial h(t)$ is the solution to backward ODEs. □

## C.3 AUGMENTED BACKWARD ODE IS WELL POSED UNDER THE SAME REGULARITY

It can be seen from equation 5 that to compute gradient of $f_\theta$ w.r.t. $W$, we need to evaluate values of $h_t$ and $\lambda_t$ for every $t \in [0,T]$. However, Chen et al. (2018b) suggested to solve an augmented backward ODE. As a result, there is no need to store the intermediate values of $h_t$ and $\lambda_t$.

We first recall the gradients of $f_\theta$ w.r.t. $\theta$ in a vectorization form:

$$\partial_v f_\theta(x) = \frac{\sigma_v}{\sqrt{n}}\phi(h(T)) \tag{35a}$$

$$\partial_W f_\theta(x) = \int_0^T \frac{\sigma_w}{\sqrt{n}}(\phi(h_t)\otimes\lambda_t)dt \tag{35b}$$

$$\partial_U f_\theta(x) = \frac{\sigma_u}{\sqrt{d}}\left[x\otimes\lambda(0)\right]. \tag{35c}$$

The according augmented backward ODE is given by

$$\begin{bmatrix}\dot{h}_t \\ \dot{\lambda}_t \\ \dot{g}_t\end{bmatrix} = \frac{\sigma_w}{\sqrt{n}}\begin{bmatrix}W\phi(h_t) \\ -\text{diag}[\phi'(h_t)W^\top]\lambda_t \\ -\phi(h_t)\otimes\lambda_t\end{bmatrix}, \quad \forall t\in[0,T] \tag{36}$$

where $g_t \in \mathbb{R}^{n^2\times 1}$ and the initial condition is $h_T$ and $\lambda_T$ combined with $g_T = 0$.

Once this augmented backward ODE is solved, the gradients of $f_\theta(x)$ w.r.t. $W$ can be obtained by

$$\begin{aligned}
\nabla_W f_\theta(x) &= g(0) \\
&= g(T) + \int_T^0 \dot{g}_t dt \\
&= g(T) + \int_T^0 -\frac{\sigma_w}{\sqrt{n}}\phi(h_t)\otimes\lambda_t dt \\
&= \int_0^T \frac{\sigma_w}{\sqrt{n}}\left[\phi(h_t)\otimes\lambda_t\right]dt,
\end{aligned}$$

where we use the fact $g_T = 0$. Unlike $h_t$ is known in equation 4, $h_t$ is an unknown state in the augmented backward ODE equation 36. Hence, it follows from Theorem 5 that extra smoothness is generally required to ensure the well posedness such as $\phi'$ probably need to be Lipschitz continuous. However, the dynamics of $h_t$ is decoupled from the dynamics of $\lambda_t$ and $g_t$. Hence, one can solve $h_t$ first (in the backward manner), then solve the dynamics system for $\lambda_t$ and $g_t$. In this manner, $h_t$ is still an known states. Hence, one can use the same regularity condition in Proposition 2 to show existence of unique solutions for $t \in [0, T]$. Therefore, no additional smoothness is needed to solve the augmented backward ODE.

## D  NNGP CORRESPONDENCE FOR NEURAL ODEs

In this section, we establish the NNGP correspondence for Neural ODEs. It follows from the Euler method that Neural ODE can be approximated by a finite-depth neural network $f_\theta^L$ equation 10. From the asymptotic perspective, Neural ODEs is equivalent to an infinite-depth ResNet with shared parameters in its all hidden layers and a special depth-dependent scaling hyperparameter $T/L$.

### D.1  FINITE-DEPTH NEURAL NETWORKS AS GAUSSIAN PROCESSES

As the finite-depth neural network $f_\theta^L$ can be considered as an approximation to the Neural ODE $f_\theta$, we first study its signal propagation by establishing the NNGP correspondence for $f_\theta^L$.

We define vectors $g^\ell \in \mathbb{R}^n$

$$g^0(x) := \frac{\sigma_v}{\sqrt{d}} U x, \tag{37}$$

$$g^\ell(x) := \frac{\sigma_w}{\sqrt{n}} W \phi(h^{\ell-1}), \quad \forall \ell \in [1, 2, \cdots, L]. \tag{38}$$

The vectors $g^\ell$ are G-vars in Tensor program Yang (2019). Tensor program is an representation of the neural network computations that only involves linear and element-wise nonlinear operations. In the paper Yang (2019), the authors claim that a computation using G-vars is equivalent to another computation that using corresponding a list of one-dimensional Gaussian variables in the infinite-width limit, as long as the computation only involves controllable nonlinear functions. The corresponding definitions and Theorems are reformulated as follows.

**Definition 1.** *(Yang, 2019, Simplified version of Definition 5.3) A real-valued function $\psi : \mathbb{R}^k \to \mathbb{R}$ is called controllable if there exists some absolute constants $C, c > 0$ such that $|\psi(x)| \le C e^{c \sum_{i=1}^k |x_i|}$.*

**Theorem 9.** *(Yang, 2019, Theorem 5.4) Consider a NETSOR program that has **forward** computation for a given finite-depth neural network. Suppose the Gaussian random initialization and controllable activation functions for the given neural network. For any controllable $\psi : \mathbb{R}^M \to \mathbb{R}$, as width $n \to \infty$, any finite collection of G-vars $g_\alpha$ with size $M$ satisfies*

$$\frac{1}{n} \sum_{\alpha=1}^n \psi(g_\alpha^0, \ldots, g_\alpha^M) \overset{a.s.}{\to} \mathbb{E}\psi(z^0, \cdots, z^M), \tag{39}$$

*where $\{z^0, \cdots, z^M\}$ are Gaussian random variables whose mean and covariance are computed by the corresponding NETSOR Program.*

Notably, controllable functions are not necessarily smooth, although smooth functions can be easily shown to be controllable. Moreover, controllable functions, as defined in (Yang, 2019, Definition 5.3), can grow faster than exponential but remain $L^1$ and $L^2$-integrable with respect to the Gaussian measure. However, the simplified definition presented here encompasses almost most functions encountered in practice. Moreover, the vectors $g^\ell$ or G-vars are not necessary to encode the same input $x$. Hence, $g^\ell(x)$ and $g^\ell(\bar{x})$ are two different G-vars in Tensor program. However, Theorem 9 still holds for any finite collection of G-vars, even they have the different input encoded. Therefore, by utilizing Theorem 9, we can show as $n \to \infty$, the finite-depth network $f_\theta^L$ tends to a Gaussian Process weakly and the result is stated in Proposition 3 and the associated Tensor program for $f_\theta^L$ is provided in Algorithm 1.

---

**Algorithm 1** ResNet $f_\theta^L$ Forward Computation on Input $x$

---

**Input:** $Ux/\sqrt{d} : \mathsf{G}(n)$
**Input:** $W : \mathsf{A}(n, n)$
**Input:** $v : \mathsf{G}(n)$
1: $h^0 := Ux/\sqrt{d} : \mathsf{G}(n)$
2: **for** $\ell \in [L]$ **do**
3: $\quad x^\ell := \phi(h^\ell) : \mathsf{H}(n)$
4: $\quad g^\ell := Wx^{\ell-1}/\sqrt{n} : \mathsf{G}(n)$
5: $\quad h^\ell := h^{\ell-1} + \kappa \cdot g^\ell : \mathsf{G}(n)$
6: **end for**
7: $x^L = \phi(h^L) : \mathsf{H}(n)$
**Output:** $v^T x^L/\sqrt{n}$

---

In the rest of this subsection, we will provide rigours proof to show the NNGP correspondence for $f_\theta^L$ through induction. For simplicity, the proof assume only one input $x$ is given, while the result for multiple inputs is similar. Additionally, we also assume $\sigma_v = \sigma_w = \sigma_u = 1$ since their values are not significant in the proof as long as their values are strictly positive.

BASIC CASE $L = 0$

As $L = 0$, we have $f_\theta^0(x) = v^T\phi(h^0)/\sqrt{n}$. Hence, we don't have the hidden layers. Based on the random initialization equation 3, we have

$$g_k^0 \overset{i.i.d.}{\sim} := \Sigma^0(x, x)$$

Let $\mathcal{B}^0$ be the smallest $\sigma$-algebra generated by $g^0$. By condition on $\mathcal{B}^0$, we have

$$f_\theta^0 | \mathcal{B}^0 \sim \mathcal{N}(0, \|\phi^0\|^2/n),$$

where $\phi^0 := \phi(h^0)$. It follows from the law of large that

$$\sigma_v^2\|\phi^0\|^2/n = \frac{\sigma_v^2}{n}\sum_{k=1}^n |\phi(h_k^0)|^2 = \frac{\sigma_v^2}{n}\sum_{k=1}^n |\phi(g_k^0)|^2 \xrightarrow{a.s} \mathbb{E}\phi(z^0)^2 := \Sigma^1(x, x),$$

where $z^0 \sim \mathcal{N}(0, \Sigma^0(x, x))$. As the limit is deterministic, the conditional and unconditional distribution converge to the same limit. Therefore, we have

$$f_\theta^0 \to \mathcal{GP}(0, \Sigma^1),$$

where

$$\Sigma^1(x, \bar{x}) = \mathbb{E}_{z^0 \sim \Sigma^0}\phi(z^0(x))\phi(z^0(\bar{x})).$$

where we use $z^0 \sim \Sigma^0$ to denote centered Gaussian random variable(s) whose (co)variances can be computed using covariance function $\Sigma^0$.

GENERAL CASE $L$

Now consider $f_\theta^L(x) = v^T\phi(h^L)/\sqrt{n}$. Here we have $h^L = h^{L-1} + \beta g^L$ and $g^L = W\phi(h^{L-1})$, where $\beta := \frac{T}{L}$. As $W$ is used before, let $\mathcal{B}^{L-1}$ be the smallest $\sigma$-algebra generated by $\{g^0, \cdots, g^{L-1}\}$. Then we can have

$$g^\ell = W\phi(h^{\ell-1}), \quad \forall \ell \in \{1, 2, \cdots, L-1\}$$

or equivalently

$$\underbrace{\begin{bmatrix} g^1 & \cdots & g^{L-1} \end{bmatrix}}_{:=G} = W\underbrace{\begin{bmatrix} \phi^0 & \cdots & \phi^{L-2} \end{bmatrix}}_{:=\Phi}$$

where $G \in \mathbb{R}^{n \times (L-1)}$ and $\Phi \in \mathbb{R}^{n \times (L-1)}$.

We can obtain the conditional distribution of $W$ by solving the following optimization problem

$$\min_{\boldsymbol{W}} \frac{1}{2}\|\boldsymbol{W}\|_F^2, \ s.t. \ \boldsymbol{G} = \boldsymbol{W}\boldsymbol{\Phi}.$$

The Lagrange function is given by

$$L(W, V) = \frac{1}{2}\|W\|_F^2 + \langle V, G - W\Phi \rangle$$

Then

$$\nabla_W L(W, V) = W - V\Phi^T = 0 \Longrightarrow W^* = V\Phi^T.$$

As $G = W\Phi$, we have

$$G = W\Phi = V\Phi^T\Phi \Longrightarrow V = G(\Phi^T\Phi)^\dagger \Longrightarrow W^* = G(\Phi^T\Phi)^\dagger\Phi^T.$$

Thus, we have

$$W|\mathcal{B} = W^* + \tilde{W}\Pi^T = G(\Phi^T\Phi)^\dagger\Phi^T + \tilde{W}\left(I_n - \Phi\Phi^\dagger\right),$$

where $\Pi = I_n - \Phi\Phi^\dagger$, $\tilde{W}$ is $i.i.d.$copy of $W$, and $\Phi^\dagger = (\Phi^T\Phi)^\dagger\Phi^T$.

Since $g^L = W\phi(h^{L-1})$, we have the conditional distribution of $g_k^L$ as follows

$$g_k^L|\mathcal{B} \overset{independent}{\sim} \mathcal{N}(G_{k*}(\Phi^T\Phi)^\dagger\Phi^T\phi, \|\Pi^T\phi\|^2/n).$$

where $G_{k*}$ denotes the $k$-th row of matrix $G$ and $\phi = \phi^{L-1}$ for simplicity.

As Lipschitz continuous activation is controllable function, it follows from Theorem 9 and inductive hypothesis that

$$\begin{aligned}
\left\langle \phi^i, \phi^j \right\rangle/n =& \frac{1}{n}\sum_{k=1}^n \phi(h_k^i)\phi(h_k^j) \\
=& \frac{1}{n}\sum_{k=1}^n \phi(g_k^0 + \beta g_k^1 + \cdots + \beta g_k^i)\phi(g_k^0 + \beta g_k^1 + \cdots + \beta g_k^j) \\
\overset{a.s.}{\to}& \mathbb{E}\phi(z^0 + \beta z^1 + \cdots + \beta z^i)\phi(z^0 + \beta z^1 + \cdots + \beta z^j) \\
=:& \mathbb{E}\phi(u^i)\phi(u^j),
\end{aligned}$$

where we define another Gaussian random variable $u^i$ to simplify the notation:

$$u^i = z^0 + \beta z^1 + \cdots + \beta z^i.$$

Therefore, we have

$$\begin{aligned}
(\Phi^T\Phi)_{ij}/n =& \left\langle \phi^i, \phi^j \right\rangle/n \overset{a.s.}{\to} \mathbb{E}\phi(u^i)\phi(u^j), \\
(\Phi^T\phi)_i/n =& \left\langle \phi^i, \phi \right\rangle/n \overset{a.s.}{\to} \mathbb{E}\phi(u^i)\phi(u^{L-1}).
\end{aligned}$$

For $\ell \in \{0, 1, \cdots, L-1\}$, let $U^\ell = \{u^0, \cdots, u^\ell\}$ be a collection of $u^i$. We define $\Sigma(U^\ell, U^k) \in \mathbb{R}^{(\ell+1)\times(k+1)}$ as

$$\Sigma(U^\ell, U^k)_{ij} = \Sigma(u^i, u^j) = \mathbb{E}\phi(u^i)\phi(u^j), \quad \forall i \in \{0, 1, \cdots, \ell\}, j \in \{0, 1, \cdots, k\}.$$

Therefore, we have

$$(\Phi^T\Phi)^\dagger\Phi^T\phi = (\Phi^T\Phi/n)^\dagger\left(\Phi^T\phi/n\right) \to \Sigma(U^{L-2}, U^{L-2})^\dagger\Sigma(U^{L-2}, u^{L-1}).$$

Moreover, observe that

$$\begin{aligned}
\|\Pi^T\phi\|^2/n =& \frac{1}{n}\phi^T(I_n - \Phi\Phi^\dagger)\phi \\
=& \frac{1}{n}\phi^T\phi - \frac{1}{n}\phi^T\Phi(\Phi^T\Phi)^\dagger\Phi^T\phi \\
=& \phi^T\phi/n - (\phi^T\Phi/n)(\Phi^T\Phi/n)^\dagger(\Phi^T\phi/n) \\
\to& \Sigma(u^{L-1}, u^{L-1}) - \Sigma(u^{L-1}, U^{L-2})\Sigma(U^{L-2}, U^{L-2})^\dagger\Sigma(U^{L-2}, u^{L-1})
\end{aligned}$$

Therefore, for any controllable function $\psi$, it follows from Theorem 9 that

$$\frac{1}{n} \sum_{k=1}^{n} \psi(g_k^0, g_k^1, \cdots, g_k^L) \to \mathbb{E}\left[\psi(z^0, z^1, \cdots, z^L)\right],$$

where

$$\operatorname{Cov}(z^0(x), z^\ell(\bar{x})) = 0, \quad \forall \ell \geq 1$$
$$\operatorname{Cov}(z^\ell(x), z^k(\bar{x})) = \mathbb{E}\left[\phi\left(u^{\ell-1}(x)\right) \phi\left(u^{k-1}(\bar{x})\right)\right], \quad \forall \ell, k \geq 1$$

Let $\mathcal{B}^L$ be the smallest $\sigma$-algebra generated by $\{g_0, \cdots, g_L\}$. By condition on $\mathcal{B}^L$, we have

$$f_\theta^L(x) | \mathcal{B}^L \sim \mathcal{N}(0, \|\phi^L\|^2/n) \tag{40}$$

where

$$\begin{aligned}
\|\phi^L\|^2/n &= \frac{1}{n} \sum_{k=1}^{n} \phi(h_k^L)^2 \\
&= \frac{1}{n} \sum_{k=1}^{n} \left[\phi\left(g_k^0 + \beta \sum_{i=1}^{L} g_k^i\right)\right]^2 \\
&\overset{a.s.}{\to} \mathbb{E}\left[\phi\left(z^0 + \beta \sum_{i=1}^{L} z^i\right)\right]^2 = \mathbb{E}[\phi(u^L)]^2 := \Sigma^{L+1}(x, x)
\end{aligned}$$

Thus, we obtain

$$f_\theta^L \to \mathcal{GP}(0, \Sigma^{L+1})$$

where

$$\Sigma^{L+1}(x, \bar{x}) = \mathbb{E}\left[\phi\left(u^L(x)\right) \phi\left(u^L(\bar{x})\right)\right].$$

### D.2 Neural ODEs as Gaussian Processes

In this subsection, we prove Neural ODEs tends to a Gaussian process as the width $n \to \infty$. As the output parameter $v$ is independent from all previous weights, by conditioning on the previous hidden layers, the Neural ODEs becomes a Gaussian random variable with covariance $\|\phi^L(x)\|^2/n$, i.e.,

$$f_\theta(x) | \mathcal{B} \sim \mathcal{N}$$

where we denote $\phi_T(x) := \phi(h_T(x))$ to simplify the notation, $h_T$ is the exact solution from the forward ODE, and $\mathcal{B}$ is the smallest $\sigma$-algebra generated by previous hidden layers. Here we also assume $\sigma_v = \sigma_w = \sigma_u = \sigma$ as their values are not important in the proof as long as they are strictly positive.

It follows from convergence analysis of Euler's method, stated in Theorem 7, that

$$\phi^L(x) \to \phi_T(x), \quad \text{as } L \to \infty,$$

where we denote $\phi^\ell(x) := \phi(h^\ell(x))$.

Thus, the focus of analysis becomes to study the convergence of this double sequence

$$a_{n,L} := \left\langle \phi^L(x), \phi^L(\bar{x}) \right\rangle / n.$$

By leveraging the convergence result for Euler's method in Theorem 7, we can show the double sequence $a_{n,L}$ converges as $L \to \infty$ and this convergence is *uniform* in $n$ a.s.

**Lemma 8.** *If $\phi$ is $L_1$-Lipschitz continuous, then the following inequalities hold for every $x \in \mathbb{S}^{d-1}$ a.s.:*

$$\|h_t\| \leq C\sqrt{n} e^{C\sigma L_1 t}, \quad \forall t \in [0, T] \tag{41}$$

*and*

$$\|h^\ell - h(t_\ell)\| \leq \frac{A}{2B}\left(e^{Bt_\ell} - 1\right) \frac{T}{L}\sqrt{n}, \tag{42}$$

*where $A := C\sigma^2 L_1^2 e^{C\sigma L_1 T}$ and $B := C\sigma L_1$ for some absolute constant $C > 0$.*

*Proof.* Recall from Lemma 5 that the mapping $f : x \mapsto \frac{\sigma}{\sqrt{n}} W \phi(x)$ is $\sigma L_1$-Lipschitz continuous. Observe that

$$
\begin{aligned}
d(\dot{h}) &= d\sigma W \phi(h(t))/\sqrt{n} \\
&= \frac{\sigma}{\sqrt{n}} W \operatorname{diag}\left[\phi'(h(t))\right] dh(t) \\
&= \frac{\sigma}{\sqrt{n}} W \operatorname{diag}\left[\phi'(h(t))\right] \dot{h}(t) dt \\
&= \frac{\sigma}{\sqrt{n}} W \operatorname{diag}\left[\phi'(h(t))\right] \frac{\sigma}{\sqrt{n}} W \phi(h(t)) dt.
\end{aligned}
$$

Then we have

$$
\ddot{h} = \frac{d}{dt}\dot{h} = \frac{\sigma}{\sqrt{n}} W \operatorname{diag}\left[\phi'(h(t))\right] \frac{\sigma}{\sqrt{n}} W \phi(h(t))
$$

and

$$
\|\ddot{h}\| \le C^2 \sigma^2 L_1^2 \|h(t)\|
$$

where we use the fact $\|W\| \le C\sqrt{n}$ a.s. from Theorem 4 for some absolute constant $C > 0$ and $\phi$ is $L_1$-Lipschitz continuous.

Then

$$
h(t) = h(0) + \int_0^t \dot{h} ds
$$

implies

$$
\begin{aligned}
\|h(t)\| &\le \|h(0)\| + \int_0^t \|\frac{\sigma}{\sqrt{n}} W \phi(h(s))\| ds \\
&\le \|h(0)\| + \int_0^t C\sigma L_1 \|h(s)\| ds.
\end{aligned}
$$

By using the Gronwall's inequality, we have

$$
\|h(t)\| \le \|h(0)\| \exp\left(\int_0^t C\sigma L_1 ds\right) = \|h(0)\| e^{C\sigma L_1 t}
$$

Additionally, as $\|U\| \le C\sqrt{n}$ almost surely and $\|x\| = 1$, we have $\|h(0)\| \le C\sqrt{n}$, and so we obtain

$$
\|h(t)\| \le C\sqrt{n} e^{C\sigma L_1 t}, \quad \forall t \in [0, T].
$$

Therefore, we obtain

$$
\|\ddot{h}(t)\| \le C\sigma^2 L_1^2 \sqrt{n} e^{C\sigma L_1 t}, \quad \forall t \in [0, T].
$$

By the Euler's convergence theorem stated in Theorem 7, we have

$$
\|h^\ell - h(t_\ell)\| \le \frac{A}{2B}\left(e^{Bt_\ell} - 1\right)\frac{T}{L}\sqrt{n},
$$

where $A := C\sigma^2 L_1^2 e^{C\sigma L_1 T}$ and $B := C\sigma L_1$. □

**Lemma 9.** *Suppose $L_1$-Lipschitz continuous activation $\phi$ and $h_t(x)$ is the exact solution with input $x$. Given $L$, we have*

$$
\left| \frac{1}{n} \left\langle \phi(h^k(x)), \phi(h^\ell(\bar{x})) \right\rangle - \frac{1}{n} \left\langle \phi(h_{t_k}(x)), \phi(h_{t_\ell}(\bar{x})) \right\rangle \right| \le C_1 L^{-1}, \quad \forall k, \ell \in [L] \tag{43}
$$

*where $t_k = k\beta$ and $C_1 > 0$ is some constant that does not dependent on $n$ and $L$. Therefore, the double sequence $\left\langle \phi(h^k(x)), \phi(h^\ell(\bar{x})) \right\rangle / n$ converges w.r.t. $L$ and uniformly w.r.t. $n$.*

*Proof.* For simplicity, we assume the activation function is 1-Lipschitz continuous, *i.e.*, $L_1 = 1$. For $\ell \leq k \leq L$, we denote $\phi^\ell = \phi(h^\ell(x))$, $\bar{\phi}^\ell = \phi(h^\ell(\bar{x}))$, $\phi(t) = \phi(h_t(x))$, and $\bar{\phi}(t) = \phi(h_t(\bar{x}))$, where $h_t(x)$ is the exact solution to the ordinary differential equation that encodes input $x$. Then we consider

$$\left\langle \phi^k, \bar{\phi}^\ell \right\rangle / n - \left\langle \phi(k\beta), \bar{\phi}(\ell\beta) \right\rangle / n = \frac{1}{n} \left\langle \phi^k, \bar{\phi}^\ell - \bar{\phi}(\ell\beta) \right\rangle + \frac{1}{n} \left\langle \phi^k - \phi(k\beta), \bar{\phi}(\ell\beta) \right\rangle,$$

where $\beta = T/L$ is the time step.

Note that

$$\|h^{\ell+1}\| = \|h^\ell + \frac{T}{L} \frac{\sigma}{\sqrt{n}} W \phi(h^\ell)\| \leq \|h^\ell\| + C\sigma \frac{T}{L} \|h^\ell\| = (1 + C\sigma T/L)\|h^\ell\|.$$

where we use the fact that $\phi$ is 1-Lipschitz continuous and $\|W\| \leq C\sqrt{n}$ a.s. Repeat this argument $\ell$ times and we have

$$\|h^{\ell+1}\| \leq (1 + C\sigma T/L)^{\ell+1} \|h^0\|$$

Therefore, we obtain

$$\|\phi^\ell\| \leq \|h^\ell(x)\| \leq (1 + C\sigma T/L)^\ell \|h^0\| \leq e^{C\sigma T\ell/L} \|h^0\| \leq C\sqrt{n} e^{C\sigma T\ell/L},$$

where we also use $\|U\| \leq C\sqrt{n}$ a.s. and $\|x\| = 1$.

Moreover, we have

$$\|\phi^\ell - \phi(\ell\beta)\| \leq \|h^\ell - h(\ell\beta)\| \leq C_1\sqrt{n}L^{-1},$$

where $C_1 > 0$ is a constant that does not dependent on $n$ and $L$.

Therefore, we obtain

$$\left| \left\langle \phi^k, \bar{\phi}^\ell \right\rangle / n - \left\langle \phi(k\beta), \bar{\phi}(\ell\beta) \right\rangle / n \right| \leq \frac{1}{n} \cdot C_1\sqrt{n} \cdot C_1\sqrt{n}L^{-1} = C_1 L^{-1}.$$

Hence, $\left\langle \phi(h^\ell(x)), \phi(h^k(\bar{x})) \right\rangle / n$ converges w.r.t. $L$ and uniformly in $n$. $\qquad\square$

Combining Lemma 9 with Moore-Osgood theorem, stated in Theorem 8, the double sequence $a_{n,L} := \left\langle \phi(h^L(x)), \phi(h^L(\bar{x})) \right\rangle / n$ has both iterated limits that are equal to the double limit, *i.e.*,

$$\begin{aligned}
\lim_{n\to\infty} \left\langle \phi(h_T(x)), \phi(h_T(\bar{x})) \right\rangle / n &= \lim_{n\to\infty} \lim_{L\to\infty} \left\langle \phi(h^L(x)), \phi(h^L(\bar{x})) \right\rangle / n \\
&= \lim_{L\to\infty} \lim_{n\to\infty} \left\langle \phi(h^L(x)), \phi(h^L(\bar{x})) \right\rangle / n \\
&= \lim_{L\to\infty} \Sigma^{L+1}(x, \bar{x}) \\
&= \Sigma^*(x, \bar{x}).
\end{aligned}$$

As $\Sigma^*$ is a deterministic function, the conditioned and unconditioned distributions of $f_\theta(x)$ are equal in the limit: they are centered Gaussian random variables with covariance $\Sigma^*(x, x)$. This complete the proof of Theorem 1

# E NTK FOR NEURAL ODE

In this section, we derive the neural tangent kernel (NTK) for Neural ODEs and provide sufficient condition to show when the NTK is well defined for Neural ODEs. Under our exploration, the smoothness of the activation function play an significant role to study the NTK of Neural ODEs. For example, additionally smoothness is required to ensure the uniqueness and existence of the adjoint state $\lambda_t$ in the backward ODE equation 4 or augmented backward ODE equation 36.

### E.1 Convergence Analysis of Euler's method for Backward ODE

Similar to the forward ODE, we can also discretize the backward ODE as follows:

$$\tilde{\lambda}^{\ell+1} = \tilde{\lambda}^\ell - \beta \cdot \frac{\sigma_w}{\sqrt{n}} \text{diag}[\phi'(h_{t_\ell})] W^T \tilde{\lambda}^\ell, \quad \forall \ell \in [1, 2, \cdots, L] \tag{44}$$

where $\beta = T/L$ and $h_t$ is the solution from the forward ODE equation 2 and $t_\ell := \beta\ell$. Additionally, we can further discretize $h_t$ and obtain

$$\lambda^{\ell+1} = \lambda^\ell - \beta \cdot \frac{\sigma_w}{\sqrt{n}} \text{diag}[\phi'(h^\ell)] W^T \lambda^\ell, \quad \forall \ell \in [1, 2, \cdots, L]. \tag{45}$$

As $L \to \infty$ or $\beta \to \infty$, we have $h^\ell \to h_{t_\ell}$ and $\lambda^\ell \to \lambda_{t_\ell}$. By utilizing the result from convergence analysis of Euler's method equation 7, we obtain the convergence rate, that indicates this convergence is *uniform* in width $n$, if the activation function is smooth. This result serves as a fundamental result to ensure the NTK for Neural ODE is well defined and allow us to study the training dynamics of Neural ODEs under gradient-based methods.

**Lemma 10.** *If $\phi$ and $\phi'$ are $L_1$- and $L_2$-Lipschitz continuous, then the following inequalities hold for every $x \in \mathbb{S}^{d-1}$ a.s.:*

$$\|\lambda_t\| \leq C\sigma L_1 e^{C\sigma L_1 (T-t)}, \quad \forall t \in [0, T] \tag{46}$$

*and*

$$\|\lambda^\ell - \lambda_t\| \leq \frac{T}{L}\left(\frac{C_1}{C_2} e^{C_2(T-t_\ell)} - 1\right), \tag{47}$$

*where $C_1 = CL_1^2 L_2 \sigma^3 e^{C\sigma L_1 T}$, $C_2 = C\sigma L_1 + C\sigma^2 L_1 L_2 e^{C\sigma L_1 T}$ for some absolute constant $C > 0$.*

*Proof.* For the mapping $f : (\lambda, t) \mapsto -\frac{1}{\sqrt{n}}\text{diag}[\phi'(h_t)]W^T\lambda$, we consider

$$
\begin{aligned}
d\dot{\lambda} &= d\left(-\frac{\sigma}{\sqrt{n}}\text{diag}\left[\phi'(h(t))\right]W^T\lambda\right) \\
&= d\left(-\phi'(h(t)) \odot \tilde{W}^T\lambda\right) \\
&= -\left[d\phi'(h_t)\right] \odot \tilde{W}^T\lambda \\
&= -\phi''(h_t) \odot dh_t \odot \tilde{W}^T\lambda \\
&= -\phi''(h_t) \odot \tilde{W}^T\lambda \odot \dot{h}dt \\
&= -\phi''(h_t) \odot \tilde{W}^T\lambda \odot \tilde{W}\phi(h_t)dt \\
&= -\text{diag}\left(\phi''(h_t)\right)\text{diag}\left(\tilde{W}^T\lambda\right)\tilde{W}\phi(h_t)dt,
\end{aligned}
$$

where $\odot$ denotes element-wise product and we denote $\tilde{W} = \sigma W/\sqrt{n}$. Thus, we have

$$\partial_t f(\lambda, t) = -\text{diag}\left(\phi''(h_t)\right)\text{diag}\left(\tilde{W}^T\lambda\right)\tilde{W}\phi(h_t).$$

Let $\tilde{w}_k$ be the $k$-th column of $\tilde{W}$. As in this case we consider $\lambda$ as fixed, $\tilde{w}_k^T\lambda$ follows Gaussian distribution with zero mean and variance $\sigma^2\|\lambda\|^2/n$. We obtain the inequality

$$\|\partial_t f(\lambda, t)\| \leq |\phi''| \cdot \frac{\sigma}{\sqrt{n}}\|\lambda\| \cdot \|\tilde{W}\| \cdot \|\phi(h_t)\| \leq CL_1 L_2 \sigma^2 \|\lambda\| \cdot \|h_t\|/\sqrt{n},$$

where we use the assumption $|\phi''| \leq L_2$ and $C > 0$ is some absolute constant.

Observe that

$$\|\lambda_t\| \leq \|\lambda_T\| + \int_t^T \|\dot{\lambda}\|ts \leq C\sigma L_1 + \int_t^T C\sigma L_1\|\lambda_s\|ds.$$

Then it follows from the Gronwall's inequality that

$$\|\lambda_t\| \leq C\sigma L_1 \exp\left(\int_t^T C\sigma L_1 ds\right) \leq C\sigma L_1 e^{C\sigma L_1(T-t)}.$$

Combining the above bound of $\lambda_t$ with equation 41, we have

$$\|\partial_t f(\lambda, t)\| \leq C L_1^2 L_2 \sigma^3 e^{C\sigma L_1 T} := C_1.$$

Note that $C_1 > 0$ is independent from $L$ and $n$.

With argument alike in Theorem 7, we can obtain the global truncation error for $\lambda^\ell$. In Proposition 1 and Proposition 2, we have shown the uniqueness and existence of $h_t$ and $\lambda_t$ for all $t \in [0, T]$. To study the convergence of $\lambda^\ell$ to $\lambda_{t_\ell}$, it is equivalent to apply Euler's method to numerically solve $\lambda_t$ in the reverse order from $t = 0$ to $t = T$. Hence, we will assume $\lambda_0$ is know and provide the global truncation errors for $\lambda^\ell$.

Note that

$$\|\lambda^{\ell+1} - \lambda(t_{\ell+1})\| = \left\| \lambda^\ell - \beta \text{diag}[\phi'(h^\ell)]\tilde{W}^T \lambda^\ell - \left[ \lambda(t_\ell) + \beta\dot{\lambda}(t_\ell) + \frac{\beta^2}{2}\ddot{\lambda}(t_\ell) \right] \right\|$$

$$\leq \|\lambda^\ell - \lambda(t_\ell)\| + \beta \|\text{diag}[\phi'(h^\ell)]\tilde{W}^T \lambda^\ell - \text{diag}[\phi'(h(t_\ell))]\tilde{W}^T \lambda(t_\ell)\| + \frac{\beta^2}{2}C_1,$$

where $\beta = T/L$ and we use $\ddot{\lambda}(t_\ell) = \|\partial_t f(t_\ell)\| \leq C_1$. Additionally, the triangle inequality implies that

$$\|\text{diag}[\phi'(h^\ell)]\tilde{W}^T \lambda^\ell - \text{diag}[\phi'(h(t_\ell))]\tilde{W}^T \lambda(t_\ell)\|$$

$$\leq \|\text{diag}[\phi'(h^\ell)]\tilde{W}^T(\lambda^\ell - \lambda_{t_\ell})\| + \|(\text{diag}[\phi'(h^\ell)] - \text{diag}[\phi'(h(t_\ell))])\tilde{W}^T \lambda(t_\ell)\|$$

$$\leq L_1\|\tilde{W}\|\|\lambda^\ell - \lambda_{t_\ell}\| + L_2\|h^\ell - h_{t_\ell}\|\|\tilde{W}\|\|\lambda_{t_\ell}\|$$

$$\leq C_2 \left( \|\lambda^\ell - \lambda_{t_\ell}\| + \|h^\ell - h_{t_\ell}\| \right),$$

where the constant $C_2 = C\sigma L_1 + C\sigma^2 L_1 L_2 e^{C\sigma L_1 T}$. Hence, we have

$$\|\lambda^{\ell+1} - \lambda_{t_{\ell+1}}\| \leq \|\lambda^\ell - \lambda(t_\ell)\| + \beta C_2 \left( \|\lambda^\ell - \lambda_{t_\ell}\| + \|h^\ell - h_{t_\ell}\| \right) + \beta^2 C_1.$$

Denote $E^\ell = \|\lambda^\ell - \lambda_{t_\ell}\| + \|h^\ell - h_{t_\ell}\|$, then we have

$$\|\lambda^\ell - \lambda_{t_\ell}\| \leq E^\ell \leq (1 + \beta C_2)E^{\ell-1} + \beta^2 C_1.$$

By the induction, we have

$$E^\ell \leq (1 + \beta C_2)^\ell E^0 + \beta^2 C_1 \cdot \frac{(1 + \beta C_2)^\ell - 1}{(1 + \beta C_2) - 1}.$$

Since $E^0 = 0$ and $\beta = T/L$, we have

$$E^\ell \leq \frac{T}{L}\left( \frac{C_1}{C_2}e^{C_2(T-t_\ell)} - 1 \right).$$

This completes the proof. $\qquad\square$

Additionally, as we have $\lambda_t = \partial f_\theta/\partial h_t$ is the solution to the backward ODE. We have

$$\left\| \frac{\partial f_\theta}{\partial h_{t_\ell}} - \frac{\partial f_\theta^L}{\partial h^\ell} \right\| \leq C_0 L^{-1}, \quad \forall \ell \in [1, 2, \cdots, L], \tag{48}$$

where $C_0 > 0$ is some constant that is not dependent on $n$ and $L$.

## E.2 GRADIENT ALIGNMENTS

By using Lemma 8 and 10, we can show the gradients obtained from the optimize-then-discrete and discrete-then-optimize as the depth $L \to \infty$. Observe that for any $x$, we have

$$\|\nabla_{\boldsymbol{v}} f^L - \nabla_{\boldsymbol{v}} f_\theta\| = \frac{\sigma}{\sqrt{n}} \|\phi(h^L) - \phi(h(T))\| \leq \frac{\sigma}{\sqrt{n}} \cdot CL^{-1} \cdot \sqrt{n} \leq CL^{-1},$$

$$\|\nabla_W f^L - \nabla_W f_\theta\| = \left\| \int_0^T \frac{1}{\sqrt{n}} \frac{\partial f}{\partial h_t} \phi(h_t) dt - \sum_{\ell=1}^L \frac{T}{L} \frac{1}{\sqrt{n}} \frac{\partial f^L}{\partial h^\ell} \phi(h^{\ell-1}) \right\|$$

$$\leq \frac{1}{\sqrt{n}} \sum_{\ell=1}^L \int_{t_{\ell-1}}^{t_\ell} \left\| \frac{\partial f}{\partial h_t} \phi(h_t) - \frac{\partial f^L}{\partial h^\ell} \phi(h^{\ell-1}) \right\| dt$$

$$\leq \frac{1}{\sqrt{n}} \sum_{\ell=1}^L \int_{t_{\ell-1}}^{t_\ell} \left\| \frac{\partial f}{\partial h_t} - \frac{\partial f^L}{\partial h^\ell} \right\| \|h_t\| + \left\| \frac{\partial f^L}{\partial h^\ell} \right\| \|h_t - h^{\ell-1}\| dt$$

$$\leq \frac{C}{\sqrt{n}} \sum_{\ell=1}^L \int_{t_{\ell-1}}^{t_\ell} \sqrt{n} L^{-1} dt$$

$$\leq C \sum_{\ell=1}^L L^{-2} = CL^{-1},$$

$$\|\nabla_U f^L - \nabla_U f_\theta\| \leq \frac{\sigma}{\sqrt{d}} \|x\| \|\lambda^0 - \lambda_0\| \leq CL^{-1}.$$

Hence, combining the three results together prove Propostion 2.

## E.3 NTK FOR FINITE-DEPTH NEURAL NETWORKS

For Neural ODE define equation 1, its NTK is given by

$$K_\theta(x, \bar{x}) = \langle \nabla_\theta f_\theta(x), \nabla_\theta f_\theta(\bar{x}) \rangle. \tag{49}$$

As we have shown in Proposition 1 and 2, $\nabla_\theta f_\theta(x)$ is well defined for every $x \in \mathbb{S}^{d-1}$ (a.s). Hence, $K_\theta(x, \bar{x})$ is well defined for every $x, \bar{x} \in \mathbb{S}^{d-1}$. While $K_\theta$ is random and varies during the training, as observed in Jacot et al. (2018), in the infinite-width limit, it converges to an explicit deterministic kernel $K_\infty$ called *limiting NTK*. Hence, we will show that $K_\infty$ is well-defined and provides its explicit form.

Recall that we use a finite-depth neural network $f_\theta^L$ defined in equation 10 that approximates Neural ODE $f_\theta$. As a result, we can also approximate the NTK $K_\theta$ using $K_\theta^L$ defined as follows

$$K_\theta^L(x, \bar{x}) := \left\langle \nabla_\theta f_\theta^L(x), \nabla_\theta f_\theta^L(\bar{x}) \right\rangle. \tag{50}$$

We denote $K_\infty^L$ be the limit of $K_\theta^L$ as width $n \to \infty$. In this subsection, we provide the explicit form for $K_\infty^L$. For the convergence analysis, we leverage the Master Theorem introduced in (Yang, 2020, Theorem 7.2). This result is similar to Theorem 9 but it consider the backward information propagation, and it is reformed as follows.

**Theorem 10.** *(Yang, 2020, Theorem 7.2) Consider a NETSOR$^\top$ program that has both forward and backward computation for a given finite-depth neural network. Suppose the Gaussian random initialization and controllable activation functions for the given neural network. For any controllable $\psi : \mathbb{R}^M \to \mathbb{R}$, as width $n \to \infty$, any finite collection of G-vars $g_\alpha$ with size $M$ satisfies*

$$\frac{1}{n} \sum_{\alpha=1}^n \psi(g_\alpha^0, \ldots, g_\alpha^M) \stackrel{a.s.}{\to} \mathbb{E}\psi(z^0, \cdots, z^M), \tag{51}$$

*where $\{z^0, \cdots, z^M\}$ are Gaussian random variables whose mean and covariance are computed by the corresponding NETSOR$^\top$.*

---

**Algorithm 2** ResNet $f_\theta^L$ Forward and Backward Computation on Input $x$

---

**Input:** $Ux/\sqrt{d} : \mathsf{G}(n)$
**Input:** $W : \mathsf{A}(n, n)$
**Input:** $v : \mathsf{G}(n)$
 1: $h^0 := Ux/\sqrt{d} : \mathsf{G}(n)$
 2: **for** $\ell \in \{1, 2, \cdots, L\}$ **do**
 3:     $x^\ell = \phi(h^{\ell-1}) : \mathsf{H}(n)$
 4:     $g^\ell := Wx^\ell/\sqrt{n} : \mathsf{G}(n)$
 5:     $h^\ell := h^{\ell-1} + \kappa \cdot g^\ell : \mathsf{G}(n)$
 6: **end for**
 7: $x^L = \phi(h^L) : \mathsf{H}(n)$
 8: $dx^L = v/\sqrt{n} : \mathsf{G}(n)$
 9: $dh^L = dx^L \odot \phi'(h^L) : \mathsf{H}(n)$
10: **for** $\ell \in \{L, L-1, \cdots, 1\}$ **do**
11:     $dg^\ell = \kappa \cdot dh^\ell : \mathsf{H}(n)$
12:     $dx^\ell = W^\top dg^\ell/\sqrt{n} : \mathsf{G}(n)$
13:     $dh^{\ell-1} = dh^\ell + \phi'(h^\ell - 1) \odot dx^\ell : \mathsf{H}(n)$
14: **end for**
**Output:** $\|x^L\|^2/n + \sum_{\ell=1}^L \langle dg^\ell x^{\ell\top}, dg^\ell x^{\ell\top} \rangle /n + \langle dh^0 x^\top, dh^0 x^\top \rangle /d$

---

As a result, this type of Tensor program is called NESTOR$^\top$ and it includes additional $G$-vals from the backward information propagation. In our setup, to compute the gradients of $f_\theta^L$ defined in equation 10, the following new $G$-vals are introduced

$$dg^{L+1} := \frac{\sigma_v}{\sqrt{n}} \text{diag}[\phi'(h^L)]v,$$

$$dg^\ell := \frac{\sigma_w}{\sqrt{n}} \text{diag}[\phi'(h^{\ell-1})]W^T, \quad \forall[1, 2, \cdots, L].$$

and the associated NESTOR$^\top$ is given in Algorithm 2 In the rest of this subsection, we provide rigorous proof to show the convergence of $K_\theta^L$ to $K_\infty^L$, as stated in Proposition 4.

Without loss of generality, we assume $\sigma_u = \sigma_w = 1$ and $\sigma_v/\sqrt{d} = 1$. As $\theta = \text{vec}(v, W, U)$, we have

$$K_\theta^L(x, \bar{x}) = \left\langle \nabla_v f_\theta^L(x), \nabla_v f_\theta^L(\bar{x}) \right\rangle + \left\langle \nabla_W f_\theta^L(x), \nabla_W f_\theta^L(\bar{x}) \right\rangle + \left\langle \nabla_U f_\theta^L(x), \nabla_U f_\theta^L(\bar{x}) \right\rangle.$$

Hence, we will show the convergence of each term. To simplify the notation, we abbreviate $f := f_\theta^L(x)$ and $\bar{f} := f_\theta^L(\bar{x})$.

CONVERGENCE OF $\left\langle \nabla_v f, \nabla_v \bar{f} \right\rangle$

By using simple calculus, we have

$$\nabla_{\boldsymbol{v}} f = \phi(\boldsymbol{h}^L)/\sqrt{n} \tag{52}$$

$$\nabla_{\boldsymbol{h}^L} f = \boldsymbol{v} \odot \phi'(\boldsymbol{h}^L)/\sqrt{n}. \tag{53}$$

By Theorem 10, we have

$$\left\langle \nabla_{\boldsymbol{v}} f, \nabla_{\boldsymbol{v}} \bar{f} \right\rangle = \frac{1}{n}\phi(\boldsymbol{h}^L)^\top \phi(\bar{\boldsymbol{h}}^L) \overset{a.s.}{\to} \mathbb{E}\phi(u^L)\phi(\bar{u}^L) = C^{L+1,L+1}(\boldsymbol{x}, \bar{\boldsymbol{x}}),$$

where $u^\ell = z^0 + \kappa \sum_{i=1}^\ell z^i$ is a centered Gaussian random variable, $z^i$ are centered Gaussian random variables defined in Proposition 3, the convergence result follows Theorem 9.

CONVERGENCE OF $\langle \nabla_W f, \nabla_W \bar{f} \rangle$

To show the convergence, we first rewrite the forward propagation suggested by the Tensor program: for all $\ell \in \{1, 2, \cdots, L\}$

$$\boldsymbol{g}^\ell = \frac{1}{\sqrt{n}} W \boldsymbol{x}^{\ell-1}$$

$$\boldsymbol{h}^\ell = \boldsymbol{h}^{\ell-1} + \kappa \boldsymbol{g}^\ell,$$

$$\boldsymbol{x}^\ell = \phi(\boldsymbol{h}^\ell).$$

By using the chain rule, we obtain

$$\nabla_{\boldsymbol{W}} f = \frac{1}{\sqrt{n}} \sum_{\ell=1}^{L} (\nabla_{\boldsymbol{g}^\ell} f) \cdot (\boldsymbol{x}^{\ell-1})^\top \tag{54}$$

Then, the quantity can be written as follows:

$$\langle \nabla_{\boldsymbol{W}} f, \nabla_{\boldsymbol{W}} \bar{f} \rangle = \sum_{\ell,k=1}^{L} \langle d\boldsymbol{g}^\ell, d\bar{\boldsymbol{g}}^k \rangle \cdot \langle \boldsymbol{x}^{\ell-1}, \bar{\boldsymbol{x}}^{k-1} \rangle / n$$

where $d\boldsymbol{z}$ is denoted the gradient of $f$ w.r.t. a vector $\boldsymbol{z}$ occurred in the forward propagation.

It follows from Theorem 9 that

$$\frac{1}{n} \langle \boldsymbol{x}^{\ell-1}, \bar{\boldsymbol{x}}^{k-1} \rangle = \frac{1}{n} \langle \phi(\boldsymbol{h}^{\ell-1}), \phi(\bar{\boldsymbol{h}}^{k-1}) \rangle \overset{a.s.}{\to} C^{\ell,k}(\boldsymbol{x}, \bar{\boldsymbol{x}}). \tag{55}$$

Moreover, we have

$$d\boldsymbol{x}^{\ell-1} = \frac{1}{\sqrt{n}} \boldsymbol{W}^\top d\boldsymbol{g}^\ell$$

and

$$d\boldsymbol{g}^\ell = \kappa d\boldsymbol{h}^\ell = \kappa(d\boldsymbol{h}^{\ell+1} + d\boldsymbol{x}^\ell \odot \phi'(\boldsymbol{h}^\ell)) = d\boldsymbol{g}^{\ell+1} + \kappa \left[ d\boldsymbol{x}^\ell \odot \phi'(\boldsymbol{h}^\ell) \right]$$

Repeat this recursive relation, and we obtain

$$d\boldsymbol{g}^\ell = \kappa \sum_{i=\ell}^{L} d\boldsymbol{x}^i \odot \phi'(\boldsymbol{h}^i) \tag{56}$$

By Theorem 10 or Yang (2020), it is equivalent to consider the coordinates in $d\boldsymbol{x}^{\ell-1}$ are asymptotically *i.i.d.* following some centered Gaussian random variables which satisfies:

$$\mathbb{E}[Z^{d\boldsymbol{x}^{\ell-1}} Z^{d\boldsymbol{x}^{k-1}}] = \kappa^2 \mathbb{E} \left[ \sum_{i,j=L}^{\ell,k} Z^{d\boldsymbol{x}^i} \bar{Z}^{d\boldsymbol{x}^j} \phi'(u^i) \phi'(\bar{u}^j) \right]$$

$$= \kappa^2 \sum_{i,j=L}^{\ell,k} \mathbb{E} \left[ Z^{d\boldsymbol{x}^i} \bar{Z}^{d\boldsymbol{x}^j} \right] \mathbb{E}[\phi'(u^i) \phi'(\bar{u}^j)]$$

Hence, we obtain

$$D^{\ell,k}(\boldsymbol{x}, \bar{\boldsymbol{x}}) = \kappa^2 \sum_{i,j=L}^{\ell+1,k+1} D^{i,j}(\boldsymbol{x}, \bar{\boldsymbol{x}}) \mathbb{E}[\phi'(u^i) \phi'(\bar{u}^j)] \tag{57}$$

As a result, we have

$$\langle \nabla_W f, \nabla_W \bar{f} \rangle \overset{a.s.}{\longrightarrow} \kappa^2 \sum_{\ell,k=1}^{L} C^{\ell,k}(x, \bar{x}) D^{\ell,k}(x, \bar{x}) \tag{58}$$

CONVERGENCE OF $\langle \nabla_U f, \nabla_U \bar{f} \rangle$

As $h^0 = Ux$, $h_i^0 = \sum_{j=1}^d U_{ij} x_j$ implies

$$\partial h_k^0 / \partial U_{ij} = \delta_{k,i} x_j.$$

Observe that

$$
\begin{aligned}
\langle \nabla_U f, \nabla_U \bar{f} \rangle &= \sum_{i,j} \frac{\partial f}{\partial U_{ij}} \frac{\partial \bar{f}}{U_{ij}} \\
&= \sum_{ij} \left( \sum_\alpha \frac{\partial h_\alpha^0}{\partial U_{ij}} \frac{\partial f}{\partial h_\alpha^0} \right) \left( \sum_\beta \frac{\partial \bar{h}_\beta^0}{\partial U_{ij}} \frac{\partial \bar{f}}{\partial \bar{h}_\beta^0} \right) \\
&= \sum_{\alpha,\beta} \frac{\partial f}{\partial h_\alpha^0} \frac{\partial \bar{f}}{\partial \bar{h}_\beta^0} \sum_{i,j} \frac{\partial h_\alpha^0}{\partial U_{ij}} \frac{\partial \bar{h}_\beta^0}{\partial U_{ij}} \\
&= \sum_{\alpha,\beta} \frac{\partial f}{\partial h_\alpha^0} \frac{\partial \bar{f}}{\partial \bar{h}_\beta^0} \sum_{i,j} \delta_{\alpha,i} x_j \delta_{\beta,i} \bar{x}_j \\
&= \sum_{\alpha,\beta} \frac{\partial f}{\partial h_\alpha^0} \frac{\partial \bar{f}}{\partial \bar{h}_\beta^0} \cdot \delta_{\alpha,\beta} x^T \bar{x} \\
&= \sum_\alpha \frac{\partial f}{\partial h_\alpha^0} \frac{\partial \bar{f}}{\partial \bar{h}_\alpha^0} \cdot x^T \bar{x} \\
&\overset{a.s.}{\to} D^{0,0}(x,\bar{x}) C^{0,0}(x,\bar{x}),
\end{aligned}
$$

where $C^{0,0}(x,\bar{x}) = x^T \bar{x}$.

Putting everything together yields

$$
\langle \nabla_\theta f, \nabla_\theta \bar{f} \rangle = \langle \nabla_v f, \nabla_v \bar{f} \rangle + \langle \nabla_W f, \nabla_W \bar{f} \rangle + \langle \nabla_U f, \nabla_U \bar{f} \rangle
$$

$$
\overset{a.s.}{\longrightarrow} C^{L+1,L+1}(x,\bar{x}) + \sum_{\ell,k=1}^L C^{\ell,k}(x,\bar{x}) D^{\ell,k}(x,\bar{x}) + C^{0,0}(x,\bar{x}) D^{0,0}(x,\bar{x})
$$

Hence, we obtain $K_\theta^L(x,\bar{x})$ converges a.s. to $K_\infty^L(x,\bar{x})$ defined as follows

$$
K_\infty^L(x,\bar{x}) = C^{L+1,L+1}(x,\bar{x}) + \sum_{\ell,k=1}^L C^{\ell,k}(x,\bar{x}) D^{\ell,k}(x,\bar{x}) + C^{0,0}(x,\bar{x}) D^{0,0}(x,\bar{x}).
$$

### E.4 NTK FOR NEURAL ODES

In the previous subsection, we have shown the NTK $K_\theta^L$ converges to a deterministic limiting NTK $K_\infty^L$ as the width $n \to \infty$. In this subsection, in the same limit, we will show the NTK $K_\theta$ of Neural ODE $f_\theta$ defined in equation 1 converges to the limiting NTK $K_\infty$.

Similar to the NNGP kernel $\Sigma^*$, the NTK $K_\infty$ can be considered as the limit of a double sequence:

$$
K_\infty(x,\bar{x}) = \lim_{n \to \infty} \langle \nabla_\theta f_\theta, \nabla_\theta \bar{f}_\theta \rangle = \lim_{n \to \infty} \lim_{L \to \infty} \langle \nabla_\theta f_\theta^L, \nabla_\theta \bar{f}_\theta^L \rangle
$$

We have shown $\lim_{n \to \infty} \langle \nabla_\theta f_\theta^L, \nabla_\theta \bar{f}_\theta^L \rangle = K_\infty^L(x,\bar{x})$ in the previous subsection. Hence, the convergence of $K_\infty$ is equivalent to show the two indices, *i.e.*, depth and width, are interchangeable. Fortunately, if the activation function $\phi$ is sufficiently smooth, the two indices are indeed swappable and so the NTK $K_\infty$ is well defined.

Based on Moore-Osgood Theorem stated in Theorem 8, a double sequence has well defined iterated limits that are equal to the double limit if the double sequence converges in one index and uniformly in the other. Hence, we will show the NTK $K_\theta^L$ as the double sequence converges in depth $L$ and uniformly with respect to the width $n$.

*Proof.* Without loss of generality, we will assume $\sigma_v = \sigma_w = 1$ and $\sigma_u/\sqrt{d} = 1$. Observe that

$$K_\theta(x, \bar{x}) = \langle \nabla_v f_\theta(x), \nabla_v f_\theta(\bar{x}) \rangle + \langle \nabla_W f_\theta(x), \nabla_W f_\theta(\bar{x}) \rangle + \langle \nabla_U f_\theta(x), \nabla_U f_\theta(\bar{x}) \rangle.$$

Hence, the rest proof is to establish the convergence rate for each term in the summation.

Note that

$$
\begin{aligned}
&\left| \langle \nabla_v f^L(x), \nabla_v f^L(\bar{x}) \rangle - \langle \nabla_v f_\theta(x), \nabla_v f_\theta(\bar{x}) \rangle \right| \\
=& \left| \frac{1}{n} \langle \phi(h^L(x)), \phi(h^L(\bar{x})) \rangle - \frac{1}{n} \langle \phi(h(x,T)), \phi(h(\bar{x},T)) \rangle \right| \\
=& \frac{1}{n} \langle \phi(h^L(x)), \phi(h^L(\bar{x})) - \phi(h(\bar{x},T)) \rangle + \frac{1}{n} \langle \phi(h^L(x)) - \phi(h(x,T)), \phi(h(\bar{x},T)) \rangle \\
\leq& \frac{L_1^2}{n} \|h^L(x)\| \|h^L(\bar{x}) - h(\bar{x},T)\| + \frac{L_1^2}{n} \|h^L(x) - h(x,T)\| \|h(\bar{x},T)\| \\
\leq& \frac{1}{n} C\sqrt{n} \cdot \sqrt{n} L^{-1} \\
=& CL^{-1},
\end{aligned}
$$

where we use Lipschitz continuous of $\phi$ and Lemma 8.

Next, we can first show $\|\nabla_W f\|$ and $\|\nabla_W f^L\|$ are upper bounded by some constants as long as $T < \infty$. Observe that

$$
\begin{aligned}
\|\nabla_W f(x)\| =& \left\| \int_0^T \frac{1}{\sqrt{n}} \lambda_t \phi(h_t) dt \right\| \\
\leq& \left\| \int_0^T \frac{1}{\sqrt{n}} \cdot e^{C\sigma(T-t)} \cdot \sqrt{n} e^{C\sigma t} dt \right\| \\
\leq& C\sigma T e^{C\sigma T},
\end{aligned}
$$

where we use Lemma 8 and 10.

Similarly, we have

$$
\begin{aligned}
\|\nabla_W f^L(x)\| =& \left\| \sum_{\ell=1}^L \frac{T}{L} \frac{1}{\sqrt{n}} \frac{\partial f^L}{\partial h^\ell} \phi(h^{\ell-1}) \right\| \\
\leq& \frac{T}{L} \sum_{\ell=1}^L \frac{1}{\sqrt{n}} \left\| \frac{\partial f^L}{\partial h^\ell} \right\| \|h^{\ell-1}\| \\
\leq& \frac{T}{L} \sum_{\ell=1}^L \frac{1}{\sqrt{n}} \cdot (1 + \sigma T/L)^{L-\ell} \cdot (1 + \sigma T/L)^{\ell-1} \cdot C\sigma\sqrt{n} \\
\leq& C\sigma T e^{\sigma T},
\end{aligned}
$$

where we have the facts

$$\|h^\ell\| \leq (1 + \sigma T/L)^\ell \|h^0\|, \tag{59}$$

$$\left\| \frac{\partial f^L}{\partial h^\ell} \right\| \leq (1 + \sigma T/L)^{L-\ell} \|\partial f^L/\partial h^L\|, \tag{60}$$

for all $\ell \in \{0, 1, \cdots, L\}$.

Additionally, it follows from Lemma 8 and 10 that

$$
\|\nabla_W f^L(x) - \nabla_W f_\theta(x)\|
$$

$$
= \left\| \int_0^T \frac{1}{\sqrt{n}} \frac{\partial f}{\partial h_t} \phi(h_t) dt - \sum_{\ell=1}^L \frac{T}{L} \frac{1}{\sqrt{n}} \frac{\partial f^L}{\partial h^\ell} \phi(h^{\ell-1}) \right\|
$$

$$
\leq \frac{1}{\sqrt{n}} \sum_{\ell=1}^L \int_{t_{\ell-1}}^{t_\ell} \left\| \frac{\partial f}{\partial h_t} \phi(h_t) - \frac{\partial f^L}{\partial h^\ell} \phi(h^{\ell-1}) \right\| dt
$$

$$
\leq \frac{1}{\sqrt{n}} \sum_{\ell=1}^L \int_{t_{\ell-1}}^{t_\ell} \left\| \frac{\partial f}{\partial h_t} - \frac{\partial f^L}{\partial h^\ell} \right\| \|h_t\| + \left\| \frac{\partial f^L}{\partial h^\ell} \right\| \|h_t - h^{\ell-1}\| dt
$$

$$
\leq \frac{C}{\sqrt{n}} \sum_{\ell=1}^L \int_{t_{\ell-1}}^{t_\ell} \sqrt{n} L^{-1} dt
$$

$$
\leq C \sum_{\ell=1}^L L^{-2} = CL^{-1}.
$$

Hence, we obtain

$$
\left\langle \nabla_W f^L(x), \nabla_W f^L(\bar{x}) \right\rangle - \left\langle \nabla_W f_\theta(x), \nabla_W f_\theta(\bar{x}) \right\rangle
$$

$$
\leq \left\langle \nabla_W f^L(x), \nabla_W f^L(\bar{x}) - \nabla_W f_\theta(\bar{x}) \right\rangle + \left\langle \nabla_W f^L(x) - \nabla_W f_\theta(x), \nabla_W f_\theta(\bar{x}) \right\rangle
$$

$$
\leq \|\nabla_W f^L(x)\| \cdot \|\nabla_W f^L(\bar{x}) - \nabla_W f_\theta(\bar{x})\| + \|\nabla_W f^L(x) - \nabla_W f_\theta(x)\| \|\nabla_W f_\theta(\bar{x})\|
$$

$$
\leq CL^{-1},
$$

or equivalently

$$
\left| \left\langle \nabla_W f^L(x), \nabla_W f^L(\bar{x}) \right\rangle - \left\langle \nabla_W f_\theta(x), \nabla_W f_\theta(\bar{x}) \right\rangle \right| \leq CL^{-1}. \tag{61}
$$

Next, observe that

$$
\left\langle \nabla_U f^L(x), \nabla_U f^L(\bar{x}) \right\rangle - \left\langle \nabla_U f_\theta(x), \nabla_U f_\theta(\bar{x}) \right\rangle
$$

$$
= \langle x, \bar{x} \rangle \left\langle \frac{\partial f^L(x)}{\partial h^0(x)}, \frac{\partial f^L(\bar{x})}{\partial h^0(\bar{x})} \right\rangle - \langle x, \bar{x} \rangle \left\langle \frac{\partial f_\theta(x)}{\partial h(x,0)}, \frac{\partial f_\theta(\bar{x})}{\partial h(\bar{x},0)} \right\rangle.
$$

Then we have

$$
\left\langle \frac{\partial f^L(x)}{\partial h^0(x)}, \frac{\partial f^L(\bar{x})}{\partial h^0(\bar{x})} \right\rangle - \left\langle \frac{\partial f_\theta(x)}{\partial h(x,0)}, \frac{\partial f_\theta(\bar{x})}{\partial h(\bar{x},0)} \right\rangle
$$

$$
\leq \left\langle \frac{\partial f^L(x)}{\partial h^0(x)}, \frac{\partial f^L(\bar{x})}{\partial h^0(\bar{x})} - \frac{\partial f_\theta(\bar{x})}{\partial h(\bar{x},0)} \right\rangle + \left\langle \frac{\partial f^L(x)}{\partial h^0(x)} - \frac{\partial f_\theta(x)}{\partial h(x,0)}, \frac{\partial f_\theta(\bar{x})}{\partial h(0,\bar{x})} \right\rangle
$$

$$
\leq \left\| \frac{\partial f^L(x)}{\partial h^0(x)} \right\| \cdot \left\| \frac{\partial f^L(\bar{x})}{\partial h^0(\bar{x})} - \frac{\partial f_\theta(\bar{x})}{\partial h(\bar{x},0)} \right\| + \left\| \frac{\partial f^L(x)}{\partial h^0(x)} - \frac{\partial f_\theta(x)}{\partial h(x,0)} \right\| \cdot \left\| \frac{\partial f_\theta(\bar{x})}{\partial h(0,\bar{x})} \right\|
$$

$$
\leq CL^{-1},
$$

where we use the Lipschitz smoothness of $\phi'$ and Lemma 10. Therefore, we have

$$
\left| \left\langle \nabla_U f^L(x), \nabla_U f^L(\bar{x}) \right\rangle - \left\langle \nabla_U f_\theta(x), \nabla_U f_\theta(\bar{x}) \right\rangle \right| \leq CL^{-1}.
$$

Then putting everything together yields

$$
\left| \left\langle \nabla_\theta f^L(x), \nabla_\theta f^L(\bar{x}) \right\rangle - \left\langle \nabla_\theta f_\theta(x), \nabla_\theta f_\theta(\bar{x}) \right\rangle \right| \leq CL^{-1}. \tag{62}
$$

Therefore, it converges uniformly in $L$ and uniformly in $n$. □

Combining Lemma 2 with Proposition 4 and Moore-Osgood Theorem 8, we can switch $L$ and $n$ in the double sequence $K_\theta(x, \bar{x})$ and obtain the desired result

$$
\begin{aligned}
K_\infty(x, \bar{x}) &= \lim_{n \to \infty} K_\theta(x, \bar{x}) \\
&= \lim_{n \to \infty} \left\langle \nabla_\theta f_\theta, \nabla_\theta \bar{f}_\theta \right\rangle \\
&= \lim_{n \to \infty} \lim_{L \to \infty} \left\langle \nabla_\theta f_\theta^L, \nabla_\theta \bar{f}_\theta^L \right\rangle \\
&= \lim_{L \to \infty} \lim_{n \to \infty} \left\langle \nabla_\theta f_\theta^L, \nabla_\theta \bar{f}_\theta^L \right\rangle \\
&= \lim_{L \to \infty} K_\infty^L(x, \bar{x}).
\end{aligned}
$$

### E.5 INTEGRAL FORM OF NNGP AND NTK

In this subsection, we provide the explicit form of the NNGP and NTK of Neural ODEs as the limits of $\Sigma^L$ and $K_\infty^L$. It follows from Proposition 5 and Lemma 8 that

$$
\Sigma^{0,t}(\boldsymbol{x}, \bar{x}) = \delta_{0,t} \frac{\sigma_u^2}{d} \boldsymbol{x}^\top \bar{\boldsymbol{x}}, \quad \forall t \in [0, T] \tag{63}
$$

$$
\Sigma^{t,s}(\boldsymbol{x}, \bar{\boldsymbol{x}}) = \sigma_w^2 \mathbb{E} \phi(u_t) \phi(\bar{u}_s), \quad \forall t, s \in [0, T], \tag{64}
$$

where $(u_t, \bar{u}_s)$ are centered Gaussian random variables with covariance

$$
\mathbb{E}(u_t, \bar{u}_s) = \Sigma^{0,0} + \int_0^t \int_0^s \Sigma^{t',s'}(\boldsymbol{x}, \bar{\boldsymbol{x}}) dt' ds'. \tag{65}
$$

Hence, the NNGP kernel of Neural ODE is given by

$$
\Sigma^*(\boldsymbol{x}, \bar{\boldsymbol{x}}) = \Sigma^{T,T}(\boldsymbol{v}, \bar{x}) = \sigma_v^2 \mathbb{E} \phi(u_T) \phi(\bar{u}_T) \tag{66}
$$

For the NTK of Neural ODEs, we have

$$
K^{t,s}(\boldsymbol{x}, \bar{\boldsymbol{x}}) = \int_t^T \int_s^T K^{t',s'}(\boldsymbol{x}, \bar{\boldsymbol{x}}) \dot{\Sigma}^{t',s'}(\boldsymbol{x}, \bar{\boldsymbol{x}}) dt' ds', \tag{67}
$$

where $\dot{\Sigma}^{t',s'}(\boldsymbol{x}, \bar{\boldsymbol{x}}) := \mathbb{E} \phi'(u_{t'}) \phi'(\bar{u}_{s'})$. As a result, the NTK of Neural ODE is given by

$$
K_\infty(\boldsymbol{x}, \bar{\boldsymbol{x}}) = \Sigma^*(\boldsymbol{x}, \bar{\boldsymbol{x}}) + \int_0^T \int_0^T \Sigma^{t,s}(\boldsymbol{x}, \bar{\boldsymbol{x}}) K^{t,s}(\boldsymbol{x}, \bar{\boldsymbol{x}}) dt ds + \Sigma^{0,0}(\boldsymbol{x}, \bar{\boldsymbol{x}}) K^{0,0}(\boldsymbol{x}, \bar{\boldsymbol{x}}). \tag{68}
$$

## F STRICT POSITIVE DEFINITENESS OF NEURAL ODE'S NTK

In this subsection, we will prove the NTK $K_\infty$ of Neural ODEs are strictly positive definite. We first recall the definition of strict positive definite for a kernel function.

**Definition 2.** *A kernel function $k : \mathbb{X} \times \mathbb{X} \to \mathbb{R}$ is strictly positive definite (SPD) if, for any finite set of distinct points $x_1, \cdots, x_N \in \mathbb{X}$, the symmetric matrix $K = [k(x_i, x_j)]_{i,j=1}^N$ is strictly positive definite, i.e., $c^\top K c > 0$ for all nonzero vector c.*

Recall that

$$
K_\theta(x, \bar{x}) = \left\langle \nabla_v f_\theta(x), \nabla_v f_\theta(\bar{x}) \right\rangle + \left\langle \nabla_W f_\theta(x), \nabla_W f_\theta(\bar{x}) \right\rangle + \left\langle \nabla_U f_\theta(x), \nabla_U f_\theta(\bar{x}) \right\rangle.
$$

In Theorem 2, we have shown that $K_\theta(x, \bar{x}) \to K_\infty(x, \bar{x})$ as $n \to \infty$, provided $\phi$ is sufficient smooth, and

$$
\left\langle \nabla_v f_\theta(x), \nabla_v f_\theta(\bar{x}) \right\rangle \to \Sigma^*(x, \bar{x}).
$$

Hence, to show $K_\infty$ is SPD, it is sufficient to show $\Sigma^*$ is SPD.

Moreover, it follows from Theorem 1 that $\lim_{L \to \infty} \Sigma^L(x, \bar{x}) = \Sigma^*(x, \bar{x})$. We first show $\Sigma^L$ is SPD.

### F.1 Dual Activation and SPD of Finite-Depth Network's NNGP Kernel

We first provide the result for finite-depth network $f_\theta^L$ defined by 10, where the depth $L < \infty$.

**Proposition 6.** *Suppose $\phi$ is $L_1$-Lipschitz continuous. If $\phi$ is non-polynomial nonlinear, then $\Sigma^L$ is SPD on $\mathbb{S}^{d-1}$ for $1 \le L < \infty$.*

The proof is based on the concept of *dual activation* and *Hermitian expansion*. Here a brief introduction is provided as follows. For details, we refer readers to Appendices from (Gao et al., 2021; Daniely et al., 2016).

Let $x \sim \mathcal{N}(0,1)$ and $f : \mathbb{R} \to \mathbb{R}$ be a real-valued function. We can define an inner product using expectation:

$$\langle f, g \rangle := \mathbb{E}_{x \sim \mathcal{N}(0,1)} f(x)g(x).$$

Thus, we can further define a Hilbert space of functions $\mathcal{H}$, that is, $f \in \mathcal{H}$ if and only if

$$\|f\|^2 = \langle f, f \rangle = \mathbb{E}_{x \sim \mathcal{N}(0,1)} |f(x)|^2 < \infty.$$

Apply Gram-Schmidt process to the polynomial functions $\{1, x, x^2, \cdots, \}$ w.r.t. to the inner product we defined before, and we obtain $\{h_n\}$ the *(normalized) Hermite polynomials* that is an **orthonormal basis** to the Hilbert space $\mathcal{H}$:

$$h_n(x) = (-1)^n e^{\frac{x^2}{2}} \frac{d^n}{dx^n} e^{-\frac{x^2}{2}},$$

The **dual activation** $\hat{\phi} : [-1, 1] \to \mathbb{R}$ of an activation function $\phi$ is defined by

$$\hat{\phi}(\rho) := \mathbb{E}_{(u,v) \sim \mathcal{N}_\rho} \phi(u)\phi(v).$$

where $\mathcal{N}_\rho$ is multidimensional Gaussian distribution with mean 0 and covariance matrix $\begin{bmatrix} 1 & \rho \\ \rho & 1 \end{bmatrix}$.

Then the **dual kernel** $K_\phi$ is defined over the unit sphere $\mathbb{S}^{d-1}$: for every pair $x, \bar{x} \in \mathbb{S}^{d-1}$, the dual kernel $K_\phi : \mathbb{S}^{d-1} \times \mathbb{S}^{d-1} \to \mathbb{R}$ is defined by

$$K_\phi(x, \bar{x}) := \hat{\phi}(x^T \bar{x}).$$

If a function $\phi \in \mathcal{H}$, we not only can obtain an expansion of $\phi$ by using the orthonormal basis of Hermitian polynomials but also an expansion to the dual activation $\hat{\phi}$ by using the same Hermitian coefficients. As a consequence, the corresponding dual kernel $K_\phi$ can be shown to be strict positive definite by using the Hermitian expansion.

**Lemma 11.** *(Daniely et al., 2016, Lemma 12) If $\phi \in \mathcal{H}$, then the **Hermitian expansion** is given by*

$$\phi(x) = \sum_{n=0}^{\infty} a_n h_n(x), \tag{69}$$

$$\hat{\phi}(\rho) = \sum_{n=0}^{\infty} a_n^2 \rho^n. \tag{70}$$

*where $a_n := \langle h_n, \phi \rangle$ is the **Hermite coefficients**.*

**Theorem 11.** *(Jacot et al., 2018, Theorem 3)(Gneiting, 2013, Theorem 1) For a function $f : [-1, 1] \to \mathbb{R}$ with $f(\rho) = \sum_{n=0}^{\infty} b_n \rho^n$, the kernel $K_f : S^{d-1} \times S^{d-1} \to \mathbb{R}$ defined by*

$$K_f(x, \bar{x}) := f(x^T \bar{x})$$

*is **strictly positive define** for any $d \ge 1$ if and only if the coefficients $b_n > 0$ for infinitely many even and odd integer $n$.*

Now, with these results, we are ready to prove the SPD of $\Sigma^L$.

**Lemma 12.** *If $\phi$ is nonlinear and non-polynomial, then $\Sigma^1$ is SPD.*

*Proof.* We first show $\Sigma^1$ is SPD. As $\Sigma^0(x, \bar{x}) = \frac{\sigma_u^2}{d} \langle x, \bar{x} \rangle$ and we have

$$\Sigma^1(x, \bar{x}) = \sigma_w^2 \mathbb{E}_{(u,v) \sim \mathcal{N}(0, G^0)} [\phi(u)\phi(v)],$$

where

$$G^0 = \frac{\sigma_u^2}{d} \begin{bmatrix} 1 & \langle x, \bar{x} \rangle \\ \langle \bar{x}, x \rangle & 1 \end{bmatrix}.$$

By the notion of dual activation, we have

$$\Sigma^1(x, \bar{x}) = \sigma_w^2 \hat{\mu}(x^T \bar{x}),$$

where $\mu(x) := \phi(\sigma_u x / \sqrt{d})$.

Clearly, $\mu$ is Lipschitz continuous since $\phi$ is. Then $\mu \in \mathcal{H}$ and let the expansion of $\mu$ in Hermite polynomials $\{h_n\}_{n=0}^\infty$ to be given as $\mu = \sum_{n=0}^\infty a_n h_n$, where $a_n = \langle \mu, h_n \rangle$ are the Hermitian coefficients. Then we can write $\hat{\mu}$ as $\hat{\mu}(\rho) = \sum_{n=0}^\infty a_n^2 \rho^n$ and we have

$$\Sigma^1(x, \bar{x}) = \sigma_w^2 \hat{\mu}(x^T \bar{x}) = \sigma_w^2 \sum_{n=0}^\infty a_n^2 (x^T \bar{x})^n.$$

Note that $\mu$ is non-polynomial if and only if $\phi$ is non-polynomial. As we assume $\phi$ is non-polynomial, we have $\mu$ is non-polynomial, hence there are infinitely many number of nonzero $a_n$ in the expansion. That indicates $b_n := a_n^2 > 0$ for infinitely many even and odd numbers. As $\sigma_w^2 > 0$, we have $\Sigma^1$ is strictly positive definite. $\qquad\square$

Next, we can show if $\Sigma^L$ is SPD, then $\Sigma^{L+1}$ is also SPD for all $L \geq 1$.

**Lemma 13.** *Suppose nonlinear non-polynomial $\phi$. Given $L < \infty$, then*

1. *$\mathbb{E}[u^\ell \bar{u}^\ell] = C^{0,0}(x, \bar{x}) + \kappa^2 \sum_{i,j=1}^\ell C^{i,j}(x, \bar{x})$ is SPD for all $\ell \in \{1, 2, \cdots, L+1\}$,*

2. *$\Sigma^L$ is also SPD.*

*Proof.* As we are working with finite-depth network $f_\theta^L$, it is fine to assume $\kappa = 1$ to simplify the notations. Then $\Sigma^\ell$ and $\Sigma^L$ have the recurrent relation, stated in Proposition 3, and so as $C^{\ell,k}$ and $C^{L,K}$. By Theorem 12, we have $C^{1,1}$ is SPD. Additionally, we have

$$C^{1,\ell}(x, \bar{x}) = \mathbb{E}\phi(u^0)\phi(\bar{u}^1) = \mathbb{E}\phi(u^0)\phi(\bar{u}^0) = C^{1,1},$$

where we use the fact $\mathbb{E}[z^0 \bar{z}^\ell] = \delta_{0,\ell} C^{0,0}(x, \bar{x})$. Thus, $C^{1,\ell}$ is SPD for all $\ell$. Recall that $\mathbb{E}[u^\ell \bar{u}^k] = C^{0,0}(x, \bar{x}) + \sum_{i=1}^\ell \sum_{j=1}^k C^{i,j}(x, \bar{x})$. Using this relation, we can write

$$\mathbb{E}[u^\ell \bar{u}^\ell] = C^{0,0}(x, \bar{x}) + C^{1,1}(x, \bar{x}) + 2\sum_{i=2}^\ell C^{1,i}(x, \bar{x}) + \sum_{i,j=2}^\ell C^{i,j}(x, \bar{x}).$$

As $C^{1,i}$ is SPD for all $i$, the symmetry of $C^{i,j}$ implies $\mathbb{E}[u^\ell \bar{u}^\ell]$ is SPD.

Now, assume the contrary, *i.e.*, $\Sigma^{\ell+1} = C^{\ell+1,\ell+1}$ is not SPD. Then there exists distinct $\{x_1, \cdots, x_N\}$ and nonzero $a \in \mathbb{R}^N$ such that

$$0 = \sum_{i,j=1}^N a_i a_j C^{\ell+1,\ell+1}(x_i, x_j) = \sum_{i,j} a_i a_j \mathbb{E}[\phi(u_i^\ell)\phi(u_j^\ell)] = \mathbb{E}\left[\sum_{i=1}^N a_i \phi(u_i^\ell)\right]^2.$$

We must have $\sum_i a_i \phi(u_i^\ell) = 0$. As we already show $u^\ell := (u_1^\ell, \cdots, u_N^\ell) \in \mathbb{R}^N$ is a non-degenerate Gaussian random variables, nonlinearity of $\phi$ implies $a = 0$, which contradicts $a \neq 0$. Hence, $\Sigma^{\ell+1} = C^{\ell+1,\ell+1}$ is SPD. $\qquad\square$

### F.2 STRICT POSITIVE DEFINITENESS OF NEURAL ODE'S NNGP KERNEL

Observe that the previous result uses induction to show $\Sigma^L$ is SPD. However, the strict positive definiteness of $\Sigma^L$ might diminish as $L \to$. To address this, we conduct a fine-grained analysis of the properties of $\Sigma^L$ and demonstrate that these properties persist when $L \to \infty$. Consequently, $\Sigma^*$ retains these essential properties, which are crucial for proving that $\Sigma^*$ is SPD.

Recall from Theorem 1 that

$$\Sigma^*(x, \bar{x}) = \mathbb{E}[\phi(u^*)\phi(\bar{u}^*)],$$

where $(u^*, \bar{u}^*)$ are centered Gaussian random variables with covariance $S^*(x, \bar{x})$ defined as the limit of $S^L(x, \bar{x})$, i.e.,

$$S^L(x, \bar{x}) = C^{0,0}(x, \bar{x}) + \kappa^2 \sum_{\ell,k=1}^{L} C^{\ell,k}(x, \bar{x}) \to S^*(x, \bar{x}), \quad \text{as } L \to \infty. \tag{71}$$

Based on the proof of Theorem 1, $S^*$ is well defined. Some essential properties of $S^L$ and $S^*$ are given as follows.

**Lemma 14.** *Suppose $L < \infty$. For any $x, \bar{x} \in \mathbb{S}^{d-1}$, we have*

1. $S^L(x, x) = S^L(\bar{x}, \bar{x})$

2. $S^L(x, x) \geq S^L(x, \bar{x})$ *and the equality holds if and only if $x = \bar{x}$*

*Proof.* As $L < \infty$, we can assume $\kappa = 1$ for simplicity. To prove the result, we make the inductive hypothesis that $C^{\ell,k}(x, x) = C^{\ell,k}(\bar{x}, \bar{x})$ for all $\ell, k \leq L$. Then observe that

$$S^{L+1}(x, \bar{x}) = S^L(x, \bar{x}) + 2 \sum_{\ell=1}^{L} C^{\ell,L+1}(x, \bar{x}) + C^{L+1,L+1}(x, \bar{x}).$$

Using the inductive hypothesis, for any $\ell \in \{1, 2, \cdots, L+1\}$ we have

$$C^{\ell,L+1}(x, x) = \mathbb{E}\phi(u^{\ell-1})\phi(u^L) = \mathbb{E}\phi(\bar{u}^{\ell-1})\phi(\bar{u}^L) = C^{\ell,L+1}(\bar{x}, \bar{x}),$$

where $(u^{\ell-1}, u^L)$ are centered Gaussian random variables with covariance

$$\mathbb{E}[u^{\ell-1}u^L] = C^{0,0}(x, x) + \sum_{i,j=1}^{\ell-1,L} C^{i,j}(x, x) = C^{0,0}(\bar{x}, \bar{x}) + \sum_{i,j=1}^{\ell-1,L} C^{i,j}(\bar{x}, \bar{x}) = \mathbb{E}[\bar{u}^{\ell-1}\bar{u}^L].$$

This shows $S^{L+1}(x, x) = S^{L+1}(\bar{x}, \bar{x})$ and also $C^{\ell,k}(x, x) = C^{\ell,k}(\bar{x}, \bar{x})$ for all $\ell, k \leq L+1$.

Next, using $C^{\ell,k}(x, x) = C^{\ell,k}(\bar{x}, \bar{x})$, we have

$$S^L(x, x) - S^L(x, \bar{x}) = \frac{1}{2}\|x - \bar{x}\|^2 + \frac{1}{2}\mathbb{E}\left|g^L(x) - g^L(\bar{x})\right|^2,$$

where the function $g^L(x) := \kappa \sum_{\ell=1}^{L} \phi(u^\ell)$. This indicates $S^L(x, x) \geq S^L(x, \bar{x})$ and the equality holds if and only if $\bar{x} = x$. $\qquad\square$

**Corollary 2.** *For any $x, \bar{x} \in \mathbb{S}^{d-1}$, we have*

1. $0 < S^*(x, x) = S^*(\bar{x}, \bar{x}) < \infty$

2. $S^*(x, x) \geq S^*(x, \bar{x})$ *and the equality holds if and only if $x = \bar{x}$.*

*Proof.* Observe that

$$S^*(x, \bar{x}) = x^T\bar{x} + \mathbb{E}\left[g(x)g(\bar{x})\right],$$

where $g(x) := \lim_{L \to \infty} g^L(x)$ for $g^L(x) = L^{-1}\sum_{\ell=1}^{L}\phi(u^\ell)$. By Lemma 17, we obtain $\left|g^L(x)\right| = \mathcal{O}(1)$ uniform in $L$. Hence, $g(x)$ is well defined and $|g(x)| = \mathcal{O}(1)$. Therefore, we obtain $S^*(x, \bar{x}) = \Theta(1)$. Additionally, it follows from the relation of $S^L(x, x) - S^L(x, \bar{x})$ that

$$S^*(x, x) - S^*(x, \bar{x}) = \frac{1}{2}\|x - \bar{x}\|^2 + \frac{1}{2}\mathbb{E}|g(x) - g(\bar{x})|^2,$$

which allows us to obtain the second result. $\qquad\square$

Now, we are ready to prove the SPD of $\Sigma^*$.

**Lemma 15.** *If $\phi$ is nonlinear and non-polynomial, then $\Sigma^*$ is SPD.*

*Proof.* As $\phi$ is Lipschitz continuous, we can use Hermitian expansion to rewrite $\Sigma^*$:

$$\Sigma^*(x, \bar{x}) = E_{(u,\bar{u}) \sim S^*(x,\bar{x})}[\phi(u)\phi(\bar{u})] = \sum_{n=0}^{\infty} a_n^2 [S^*(x,\bar{x})/S_0]^n,$$

where we use $(u, \bar{u}) \sim S^*(x, \bar{x})$ to denote centered Gaussian random variables with covariance computed using kernel $S^*(x, \bar{x})$, $a_n$ is the Hermitian coefficients of function $\psi(u) := \phi(\sqrt{S_0}u)$ with $S_0 := S^*(x, x)$, and we also use the facts $S_0 = S^*(x, x) = S^*(\bar{x}, \bar{x})$ for all $x, \bar{x}$ and $S_0 = \Theta(1)$ from Corollary 2.

Suppose we are given any finite distinct $\{x_i\}_{i=1}^N$ from $\mathbb{S}^{d-1}$ and nonzero $c \in \mathbb{R}^N$. Observe that

$$\sum_{i,j=1}^N c_i c_j \Sigma^*(x_i, x_j) = \sum_{n=0}^{\infty} a_n^2 S_0^{-n} \sum_{i,j=1}^N c_i c_j [S^*(x_i, x_j)]^n$$

$$= \sum_{n=0}^{\infty} a_n^2 S_0^{-n} \sum_{i,j=1}^N c_i c_j \left[ x_i^T x_j + \mathbb{E}g(x_i)g(x_j) \right]^n,$$

where we use $S^*(x, \bar{x}) = x^T \bar{x} + \mathbb{E}g(x)g(\bar{x})$. By using fundamental properties for positive definite matrices from linear algebra, we have

$$\sum_{i,j=1}^N c_i c_j \left[ x_i^T x_j + \mathbb{E}g(x_i)g(x_j) \right]^n = c^T (XX^T + \mathbb{E}g(X)g(X)^T)^{\odot n} c$$

$$\geq c^T (XX^T)^{\odot n} c = \sum_{i,j=1}^N c_i c_j \left[ x_i^T x_j \right]^n,$$

where $\odot$ is Hadamard product. Then we obtain

$$\sum_{i,j=1}^N c_i c_j \Sigma^*(x_i, x_j) \geq \sum_{n=0}^{\infty} a_n^2 S_0^{-n} \sum_{i,j=1}^N c_i c_j \left( x_i^T x_j \right)^n$$

$$= \sum_{i,j=1}^N c_i c_j \sum_{n=0}^{\infty} a_n^2 (x_i^T x_j / S_0)^n$$

$$= \sum_{i,j=1}^N c_i c_j \mathbb{E}_{(u,\bar{u}) \sim x_i^T x_j / S_0}[\psi(u)\psi(\bar{u})]$$

$$= \sum_{i,j=1}^N c_i c_j \mathbb{E}_{(u,\bar{u}) \sim x_i^T x_j}[\psi(u/\sqrt{S_0})\psi(\bar{u}/\sqrt{S_0})]$$

$$= \sum_{i,j=1}^N c_i c_j \mathbb{E}_{(u,\bar{u}) \sim x_i^T x_j}[\phi(u)\phi(\bar{u})]$$

$$= \sum_{i,j=1}^N c_i c_j \Sigma^1(x_i, x_j),$$

where we use the definitions of Hermitian coefficients $a_n$ and $\psi$. By Lemma 12, $\Sigma^1$ is SPD and so $\Sigma^*$ is also SPD. $\square$

As a corollary result, we have the NTK of Neural ODE is also SPD.

**Corollary 3.** *Suppose $\phi$ and $\phi'$ are nonlinear Lipschitz continuous. If $\phi$ is non-polynomial, then the NTK $K_\infty$ of Neural ODE is SPD.*

**Theorem 12.** *Let $\{x_i, y_i\}_{i=1}^{N}$ be a training set. Assume*

1. *$x_i \in \mathbb{S}^{d-1}, |y_i| \leq 1$, and $x_i \neq x_j$ for all $i \neq j$.*

2. *the activation $\phi$ is $L_1$-Lipschitz nonlinear continuous, but non-polynomial,*

3. *its derivative $\phi'$ are $L_2$-Lipschitz nonlinear continuous,*

4. *and we choose the learning rate $\eta \leq 1/\|X\|^2$.*

*For any $\delta > 0$, there exists a natural number $n_\delta$ such that for all $n \geq n_\delta$ the parameter $\theta^k$ stays in a neighborhood of $\theta^0$, i.e.,*

$$\|\theta^k - \theta^0\| \leq C\|X\|\|u_0 - y\|/\lambda_0, \tag{72}$$

*and the loss function $L(\theta_k)$ consistently decrease to zero at an exponential rate, i.e.,*

$$L(\theta_k) \leq \left(1 - \frac{\eta\lambda_0}{16}\right)^k L(\theta_0), \tag{73}$$

*where $C > 0$ is some constant only depends on $L_1$, $L_2$, $\sigma_v$, $\sigma_w$, $\sigma_u$, and $T$.*

*Proof.* Given a distinct $\{x_i\}_{i=1}^{N}$, we consider the limiting NTK matrix $H^\infty \in \mathbb{R}^{N \times N}$ defined as $H_{ij}^\infty = K_\infty(x_i, x_j)$. As $\phi$ is non-polynomial, we have $\lambda_0 := \lambda_{\min}\{H^\infty\} > 0$. Let $\theta_0$ denote the parameters at initialization and $H(0) \in \mathbb{R}^{N \times N}$ be the corresponding NTK computed by $\theta_0$ at initialization.

By Theorem 2, we have $H(0)$ converges a.s. to $H^\infty$, as the width $n \to \infty$. Then for any $\delta_0 > 0$, there exists a natural number $n_0$ such that with probability at least $(1-\delta_0)$ over random initialization $\lambda_{\min}\{H(0)\} \geq \lambda_0/2$ for all $n \geq n_0$. By Lemma 19, there exists another natural number $n_1$ such that with probability at least $(1-\delta_0)$, the initial residual $\|u_0 - y\| \leq \sigma_* \sqrt{2N \log N/\delta}$ for all $n \geq n_1$. Therefore, for any $\delta > 0$, we choose $\delta_0 = \delta/2$, and it follows from Lemma 16 that, with probability at least $(1 - \delta)$ over random initialization, we have

$$\|v^k - v^0\|, \quad \|W^k - W^0\|, \quad \|U^k - U^0\| \leq C\|X\|\|u_0 - y\|/\lambda_0,$$

and

$$\|u^k - y\| \leq \left(1 - \frac{\eta\lambda_0}{16}\right)^k \|u^0 - y\|,$$

for all

$$n \geq \max\left\{n_0, n_1, C_0 N^3 \log(N/\delta)/\lambda_0^3\right\}.$$

$\square$

## G  GLOBAL CONVERGENCE OF NEURAL ODES

In this section, we provide the convergence analysis of Neural ODEs defined equation 1 under gradient descent.

As we use square loss, the loss function is given by

$$L(\theta) := \sum_{i=1}^{N} \frac{1}{2}(f_\theta(x_i) - y_i)^2. \tag{74}$$

By using the vectorization form equation 35 and chain rule, the gradients are given by

$$\frac{\partial L(\theta)}{\partial v} = \sum_{i=1}^{N} \frac{\sigma_v}{\sqrt{n}} \phi(h_T(x_i))(f_\theta(x_i) - y_i), \tag{75}$$

$$\frac{\partial L(\theta)}{\partial W} = \sum_{i=1}^{N} \left[\int_0^T \frac{\sigma_w}{\sqrt{n}} \phi(h_t(x_i)) \otimes \lambda_t(x_i) dt\right] (f_\theta(x_i) - y_i), \tag{76}$$

$$\frac{\partial L(\theta)}{\partial U} = \sum_{i=1}^{N} \frac{\sigma_u}{\sqrt{d}} [x_i \otimes \lambda_0(x_i)] (f_\theta(x_i) - y_i). \tag{77}$$

Consider the gradient descent

$$\theta^{k+1} = \theta^k - \eta \frac{\partial L(\theta^k)}{\partial \theta}. \tag{78}$$

Assume the inductive hypothesis: For all $i \leq k$, there exist some constants $\alpha_v, \alpha_w, \alpha_u > 0$ such that

1. $\|v_i\|, \|W_i\|, \|U_i\| \leq C\sqrt{n}$,
2. $\|u^i - y\| \leq (1 - \eta\alpha_0^2)^i \|u^0 - y\|$,

where $C > 0$ is a constant and $\alpha_0 := \sigma_{\min}\left(\frac{\sigma_v}{\sqrt{n}} \Phi^0\right)$.

Without loss generality, we assume $\sigma_v = 1$, $\sigma_w = \sigma$, $\sigma_u/\sqrt{d} = 1$ and $L_1 = L_2 = 1$.

Observe that

$$\|\frac{\partial f_\theta}{\partial v}\| = \|\frac{1}{\sqrt{n}}\phi(h_T)\| \leq \frac{1}{\sqrt{n}}\|U\|\|x\|e^{\sigma T\|W\|/\sqrt{n}}.$$

Note that

$$\begin{aligned}
\|\frac{\partial f_\theta}{\partial W}\| &\leq \frac{\sigma}{\sqrt{n}} \int_0^T \|\phi(h_t)\|\|\lambda_t\| dt \\
&\leq \frac{\sigma}{\sqrt{n}} \int_0^T \|U\|\|x\|e^{\sigma t\|W\|/\sqrt{n}} \cdot \frac{\|v\|}{\sqrt{n}} e^{\sigma(T-t)\|W\|/\sqrt{n}} dt \\
&= (\sigma T)\frac{\|U\|}{\sqrt{n}}\frac{\|v\|}{\sqrt{n}}\|x\|e^{\sigma T\|W\|/\sqrt{n}}.
\end{aligned}$$

Observe that

$$\|\frac{\partial f_\theta}{\partial U}\| \leq \|x\|\|\lambda_0\| \leq \|x\| \cdot \frac{\|v\|}{\sqrt{n}} \exp\left\{\sigma T\|W\|/\sqrt{n}\right\}$$

By using the inductive hypothesis, we obtain

$$\|\frac{\partial f_\theta}{\partial v}\| \leq Ce^{C\sigma T}\|x\|, \tag{79}$$

$$\|\frac{\partial f_\theta}{\partial W}\| \leq (\sigma T)Ce^{C\sigma T}\|x\|, \tag{80}$$

$$\|\frac{\partial f_\theta}{\partial U}\| \leq Ce^{C\sigma T}\|x\|. \tag{81}$$

Then we obtain

$$\begin{aligned}
\|v^{k+1} - v^0\| &\leq \eta \sum_{i=0}^k \|\frac{\partial L(\theta^i)}{\partial v}\| \\
&\leq \eta \sum_{i=0}^k Ce^{C\sigma T}\|X\|\|u^i - y\| \\
&\leq \eta Ce^{C\sigma T}\|X\| \sum_{i=0}^k (1 - \eta\alpha_0^2)^i \|u^0 - y\| \\
&\leq Ce^{C\sigma T}\|X\|\|u^0 - y\|/\alpha_0^2
\end{aligned}$$

**Note** that the RHS is an constant after initialization. If we assume $\|x\| = 1$ and $|y| = 1$, then we need to ensure

$$Ce^{C\sigma T}\|X\|\|u^0 - y\|/\alpha_0^2 \leq C\sqrt{n}. \tag{82}$$

And as a result, we have
$$\|v^{k+1}\| \le \|v^{k+1} - v^0\| + \|v^0\| \le C\sqrt{n}.$$
Similarly, we have
$$\|W^{k+1} - W^0\| \le \eta \sum_{i=0}^{k} \|\frac{\partial L(\theta^i)}{\partial W}\|$$
$$\le \eta \sum_{i=0}^{k} (\sigma T)Ce^{C\sigma T}\|X\|\|u^i - y\|$$
$$\le \eta(\sigma T)Ce^{C\sigma T}\|X\| \sum_{i=0}^{k} (1 - \eta\alpha_0^2)\|u^0 - y\|$$
$$\le (\sigma T)Ce^{C\sigma T}\|X\|\|u^0 - y\|/\alpha_0^2.$$
Then we need to ensure
$$(\sigma T)Ce^{C\sigma T}\|X\|\|u^0 - y\|/\alpha_0^2 \le C\sqrt{n}. \tag{83}$$
Then we obtain
$$\|W^{k+1}\| \le \|W^{k+1} - W^0\| + \|W^0\| \le C\sqrt{n}.$$
Observe that
$$\|U^{k+1} - U^0\| \le \eta \sum_{i=0}^{k} \|\frac{\partial L(\theta^i)}{\partial U}\|$$
$$\le \eta \sum_{i=0}^{k} Ce^{C\sigma T}\|X\|\|u^i - y\|$$
$$\le \eta Ce^{C\sigma T}\|X\| \sum_{i=0}^{k} (1 - \eta\alpha_0^2)^i\|u^0 - y\|$$
$$\le Ce^{C\sigma T}\|X\|\|u^0 - y\|/\alpha_0^2.$$
Hence, we obtain
$$\|U^{k+1}\| \le \|U^{k+1} - U^0\| + \|U^0\| \le C\sqrt{n}.$$
Next, observe that
$$u^{k+1} - y = u^{k+1} - u^k + (u^k - y)$$
$$= \left(\frac{\partial \tilde{u}}{\partial \theta}\right)^\top (\theta^{k+1} - \theta^k) + (u^k - y)$$
$$= \left(\frac{\partial \tilde{u}}{\partial \theta}\right)^\top \left(-\eta\frac{\partial u^k}{\partial \theta}\right)(u^k - y) + (u^k - y)$$
$$= \left[I - \eta\left(\frac{\partial \tilde{u}}{\partial \theta}\right)^\top \left(\frac{\partial u^k}{\partial \theta}\right)\right](u^k - y)$$
$$= \left[I - \eta\left(\frac{\partial u^k}{\partial \theta}\right)^\top \left(\frac{\partial u^k}{\partial \theta}\right)\right](u^k - y) + \eta\left(\frac{\partial u^k}{\partial \theta} - \frac{\partial \tilde{u}}{\partial \theta}\right)^\top \left(\frac{\partial u^k}{\partial \theta}\right)(u^k - y)$$
where $\tilde{u} = u(\tilde{\theta})$ and $\tilde{\theta}$ is an interpolation in between $\theta^k$ and $\theta^{k+1}$.

Note that
$$\|\frac{\partial f}{\partial v} - \frac{\partial \bar{f}}{\partial v}\| = \|\frac{1}{\sqrt{n}}\phi(h_T) - \frac{1}{\sqrt{n}}\phi(\bar{h}_T)\|$$
$$\le \frac{1}{\sqrt{n}}\|h_T - \bar{h}_T\|$$
$$\le \frac{C}{\sqrt{n}}\|\theta - \bar{\theta}\|e^{C\sigma T}\|x\|$$

where we use the Lemma and the inductive hypotheses.

Similarly, note that

$$\|\frac{\partial f}{\partial W} - \frac{\partial \bar{f}}{\partial W}\| \leq \frac{\sigma}{\sqrt{n}} \| \int_0^T \phi(h_t) \otimes \lambda_t - \phi(\bar{h}_t) \otimes \bar{\lambda}_t dt \|$$

$$\leq \frac{\sigma}{\sqrt{n}} \int_0^T \Big( \|h_t - \bar{h}_t\| \|\lambda_t\| + \|\bar{h}_t\| \|\lambda_t - \bar{\lambda}_t\| \Big) dt$$

$$\leq C \frac{\sigma}{\sqrt{n}} \int_0^T \|\theta - \bar{\theta}\| e^{C\sigma t} \|x\| \cdot e^{C\sigma(T-t)} dt$$

$$\leq (\sigma T) \frac{C}{\sqrt{n}} \|\theta - \bar{\theta}\| e^{C\sigma T} \|x\|.$$

and

$$\|\frac{\partial f}{\partial U} - \frac{\partial \bar{f}}{\partial U}\| \leq \|x\| \|\lambda_0 - \bar{\lambda}_0\| \leq \frac{C}{\sqrt{n}} \|\theta - \bar{\theta}\| e^{C\sigma T} \|x\|.$$

Hence, we have

$$\|\frac{\partial f}{\partial \theta} - \frac{\partial \bar{f}}{\partial \theta}\| = \|\frac{\partial f}{\partial v} - \frac{\partial \bar{f}}{\partial v}\| + \|\frac{\partial f}{\partial W} - \frac{\partial \bar{f}}{\partial W}\| + \|\frac{\partial f}{\partial U} - \frac{\partial \bar{f}}{\partial U}\| \leq (\sigma T) \frac{C}{\sqrt{n}} \|\theta - \bar{\theta}\| e^{C\sigma T} \|x\|.$$

Then

$$\|\frac{\partial u^k}{\partial \theta} - \frac{\partial \tilde{u}}{\partial \theta}\| \leq (\sigma T) \frac{C}{\sqrt{n}} \|\theta^k - \tilde{\theta}\| e^{C\sigma T} \|X\| \leq (\sigma T) \frac{C}{\sqrt{n}} \|\theta^k - \theta^{k+1}\| e^{C\sigma T} \|X\|,$$

where we use the fact $\tilde{\theta} = \alpha \theta^k + (1 - \alpha)\theta^{k+1}$ for some $\alpha \in [0, 1]$.

Observe that

$$\|\theta^{k+1} - \theta^k\| = \eta \|\frac{\partial L(\theta^k)}{\partial \theta}\| = \eta \| \Big( \frac{\partial u^k}{\partial \theta} \Big)^\top (u^k - y) \| \leq \eta (\sigma T) C e^{C\sigma T} \|X\| \|u^k - y\|.$$

Hence, we obtain

$$\|\frac{\partial u^k}{\partial \theta} - \frac{\partial \tilde{u}}{\partial \theta}\| \leq \eta (\sigma T)^2 \frac{C}{\sqrt{n}} e^{C\sigma T} \|X\|^2 \|u^k - y\|,$$

and

$$\|\frac{\partial u^k}{\partial \theta} - \frac{\partial u^0}{\partial \theta}\| \leq (\sigma T) \frac{C}{\sqrt{n}} \|\theta^k - \theta^0\| e^{C\sigma T} \|X\|$$

$$\leq (\sigma T) \frac{C}{\sqrt{n}} e^{C\sigma T} \|X\| \sum_{i=0}^{k-1} \|\theta^{i+1} - \theta^i\|$$

$$\leq \eta (\sigma T)^2 \frac{C}{\sqrt{n}} e^{C\sigma T} \|X\|^2 \sum_{i=0}^{k-1} \|u^i - y\|$$

$$\leq \eta (\sigma T)^2 \frac{C}{\sqrt{n}} e^{C\sigma T} \|X\|^2 \sum_{i=0}^{k-1} (1 - \eta \alpha_0^2)^i \|u^0 - y\|$$

$$\leq (\sigma T)^2 \frac{C}{\sqrt{n}} e^{C\sigma T} \|X\|^2 \|u^0 - y\|/\alpha_0^2$$

$$\leq \alpha_0/2,$$

where we use the assumption

$$\sqrt{n} \geq C(\sigma T)^2 e^{C\sigma T} \|X\|^2 \|u^0 - y\|/\alpha_0^3. \tag{84}$$

It follows from the Weyl's inequality that

$$\sigma_{\min}\left(\frac{\partial u^k}{\partial \theta}\right) \geq \sigma_{\min}\left(\frac{\partial u^0}{\partial \theta}\right) - \|\frac{\partial u^k}{\partial \theta} - \frac{\partial u^0}{\partial \theta}\| \geq \alpha_0/2.$$

and so

$$\lambda_{\min}\left[\left(\frac{\partial u^k}{\partial \theta}\right)^T \left(\frac{\partial u^k}{\partial \theta}\right)\right] \geq \alpha_0^2/4.$$

Therefore, we obtain

$$\|u^{k+1} - y\| \leq \left[1 - \eta\alpha_0^2/4\right]\|u^k - y\| + \eta^2(\sigma T)^3 \frac{C}{\sqrt{n}}e^{C\sigma T}\|X\|^3\|u^k - y\|^2$$

$$\leq \left[1 - \eta\alpha_0^2/4 + \eta^2(\sigma T)^3 \frac{C}{\sqrt{n}}e^{C\sigma T}\|X\|^3\|u^0 - y\|\right]\|u^k - y\|$$

$$= \left[1 - \eta\left(\alpha_0^2/4 - \eta(\sigma T)^3 \frac{C}{\sqrt{n}}e^{C\sigma T}\|X\|^3\|u^0 - y\|\right)\right]\|u^k - y\|$$

$$\leq \left[1 - \eta\alpha_0^2/8\right]\|u^k - y\|,$$

where we assume

$$\sqrt{n} \geq 8C\eta(\sigma T)^3 e^{C\sigma T}\|X\|^3\|u^0 - y\|/\alpha_0^2.$$

This finishes proving Lemma 16.

**Lemma 16.** *Assume $\phi$ and $\phi'$ are $L_1$- and $L_2$-Lipschitz continuous and $\lambda_0 := \lambda_{\min}(K_{\theta_0}) > 0$. Suppose we choose the width $n = \Omega(\|X\|^4\|u^0 - y\|^2/\lambda_0^3)$ and the learning rate $\eta \leq \frac{1}{\|X\|^2}$. Then the parameters $\theta^k$ stays in the neighborhood of $\theta^0$, i.e.,*

$$\|v^k - v^0\|, \quad \|W^k - W^0\|, \quad \|U^k - U^0\| \leq C\|X\|\|u_0 - y\|/\lambda_0, \tag{85}$$

*and the residual $\|u_k - y\|$ consistently decreases, i.e.,*

$$\|u^k - y\| \leq \left(1 - \frac{\eta\lambda_0}{8}\right)^k \|u^0 - y\|, \tag{86}$$

*where $C > 0$ is some constant only depends on $L_1$, $L_2$, $\sigma_v$, $\sigma_w$, $\sigma_u$, and $T$.*

**Lemma 17.** *Given $\theta$, we have*

$$\|h_t\| \leq \|U\|\|x\| \exp\left\{\frac{\sigma t}{\sqrt{n}}\|W\|\right\}, \tag{87}$$

$$\|\lambda_t\| \leq \frac{\|v\|}{\sqrt{n}} \exp\left\{\frac{\sigma(T - t)}{\sqrt{n}}\|W\|\right\}, \tag{88}$$

*for all $t \in [0, T]$*

*Proof.* Observe that

$$h_t = h_0 + \int_0^t \frac{\sigma}{\sqrt{n}}W\phi(h_s)ds$$

and so

$$\|h_t\| \leq \|h_0\| + \frac{\sigma}{\sqrt{n}}\|W\|\int_0^t \|h_s\|ds$$

Then it follows from the Gronwall's inequality that

$$\|h_t\| \leq \|U\|\|x\| \exp\left\{\frac{\sigma t}{\sqrt{n}}\|W\|\right\}, \quad \forall t \in [0, T]. \tag{89}$$

Similarly, we have

$$\lambda_t = \lambda_T + \int_t^T -\frac{\sigma}{\sqrt{n}}\text{diag}[\phi'(h_t)]W^\top \lambda_s ds$$

implies

$$\|\lambda_t\| \leq \|\lambda_T\| + \frac{\sigma}{\sqrt{n}}L_1\|W\|\int_t^T \|\lambda_s\|ds.$$

By the Gronwall's inequality, we obtain

$$\|\lambda_t\| \leq \|\lambda_T\|\exp\left\{\int_t^T \sigma\|W\|/\sqrt{n}ds\right\}$$
$$\leq \|\lambda_T\|\exp\left\{\sigma\|W\|/\sqrt{n}(T-t)\right\}.$$

By $\lambda_T = \frac{1}{\sqrt{n}}\text{diag}[\phi'(h_T)]v$, we obtain the final result.

$\square$

**Lemma 18.** *Given $\theta, \bar{\theta}$, we have*

$$\|h_t - \bar{h}_t\| \leq \|\theta - \bar{\theta}\|\frac{\|U\|}{\|W\|}e^{\sigma t(\|W\|+\|\bar{W}\|)/\sqrt{n}}\|x\| \tag{90}$$

$$\|\lambda_t - \bar{\lambda}_t\| \leq \|\theta - \bar{\theta}\|\frac{\|v\|}{\|W\|}e^{\sigma(T-t)(\|W\|+\|\bar{W}\|)/\sqrt{n}}/\sqrt{n} \tag{91}$$

*for all $t \in [0, T]$*

*Proof.* Observer that

$$h_t - \bar{h}_t = (h_0 - \bar{h}_0) + \frac{\sigma}{\sqrt{n}}\int_0^t \left[W\phi(h_s) - \bar{W}\phi(\bar{h}_s)\right]ds$$

Then we have

$$\|h_t - \bar{h}_t\| \leq \|h_0 - \bar{h}_0\| + \frac{\sigma}{\sqrt{n}}\int_0^t \left[\|W - \bar{W}\|\|h_s\| + \|\bar{W}\|\|h_s - \bar{h}_s\|\right]ds$$

$$\leq \|h_0 - \bar{h}_0\| + \frac{\sigma}{\sqrt{n}}\|W - \bar{W}\|\int_0^t \|Ux\|\exp\left\{\sigma s\|W\|/\sqrt{n}\right\}ds + \frac{\sigma}{\sqrt{n}}\|\bar{W}\|\int_0^t \|h_s - \bar{h}_s\|ds$$

Using the bound of $\|h_s\|$, we have

$$\frac{\sigma}{\sqrt{n}}\|Ux\|\|W - \bar{W}\|\int_0^t \exp\left\{\sigma s\|W\|/\sqrt{n}\right\}ds$$

$$= \frac{\sigma}{\sqrt{n}}\|Ux\|\|W - \bar{W}\| \cdot \left(\frac{\sigma}{\sqrt{n}}\|W\|\right)^{-1}\left(e^{\sigma t\|W\|/\sqrt{n}} - 1\right)$$

$$= \frac{\|U\|}{\|W\|}\|W - \bar{W}\|\left(e^{\sigma t\|W\|/\sqrt{n}} - 1\right)\|x\|.$$

Then by Grownwall's inequality, we obtain

$$\|h_t - \bar{h}_t\| \leq \left(\|h_0 - \bar{h}_0\| + \|W - \bar{W}\|\frac{\|U\|}{\|W\|}\left(e^{\sigma\|W\|t/\sqrt{n}} - 1\right)\|x\|\right)e^{\sigma\|\bar{W}\|t/\sqrt{n}}$$

$$\leq \left(\|U - \bar{U}\| + \|W - \bar{W}\|\right)\frac{\|U\|}{\|W\|}e^{\sigma t(\|W\|+\|\bar{W}\|)/\sqrt{n}}\|x\|.$$

Then we obtain the result.

Observe that

$$\lambda_t - \bar{\lambda}_t = (\lambda_T - \bar{\lambda}_T) + \frac{\sigma}{\sqrt{n}}\int_t^T \text{diag}[\phi'(h_s)]W^\top \lambda_s - \text{diag}[\phi'(\bar{h}_s)]\bar{W}^\top \bar{\lambda}_s ds.$$

Then we obtain

$$\|\lambda_t - \bar{\lambda}_t\| \le \|\lambda_T - \bar{\lambda}_T\| + \frac{\sigma}{\sqrt{n}} \int_t^T \left( \|W - \bar{W}\|\|\lambda_s\| + \|\bar{W}\|\|\lambda_s - \bar{\lambda}_s\| \right) ds$$

By using the bound of $\|\lambda_s\|$, we obtain

$$\frac{\sigma}{\sqrt{n}} \|W - \bar{W}\| \frac{\|v\|}{\sqrt{n}} \int_t^T \exp\left\{ \frac{\sigma(T-s)}{\sqrt{n}} \|W\| \right\} ds$$

$$\le \frac{\sigma}{\sqrt{n}} \|W - \bar{W}\| \frac{\|v\|}{\sqrt{n}} \left( \frac{\sigma}{\sqrt{n}} \|W\| \right)^{-1} \left( e^{\sigma(T-t)\|W\|/\sqrt{n}} - 1 \right)$$

$$= \frac{1}{\sqrt{n}} \|W - \bar{W}\| \frac{\|v\|}{\|W\|} \left( e^{\sigma(T-t)\|W\|/\sqrt{n}} - 1 \right).$$

Then by Grownwall's inequality, we have

$$\|\lambda_t - \bar{\lambda}_t\| \le \left( \|\lambda_T - \bar{\lambda}_T\| + \frac{1}{\sqrt{n}} \|W - \bar{W}\| \frac{\|v\|}{\|W\|} \left( e^{\sigma(T-t)\|W\|/\sqrt{n}} - 1 \right) \right) e^{\sigma(T-t)\|\bar{W}\|/\sqrt{n}}$$

$$\le \frac{1}{\sqrt{n}} \left( \|v - \bar{v}\| + \|W - \bar{W}\| \right) \frac{\|v\|}{\|W\|} e^{\sigma(T-t)(\|W\|+\|\bar{W}\|)/\sqrt{n}}$$

$$\square$$

**Lemma 19.** *Given $\delta > 0$, there exists a natural number $n_\delta$ such that for all $n \ge n_\delta$, with probability at least $1 - \delta$ over random initialization, we have*

$$\|u\| \le \sigma \sqrt{2N \log(N/\delta)}, \tag{92}$$

*where $\sigma^2 := \Sigma^*(x,x)$ for $x \in \mathbb{S}^{d-1}$.*

*Proof.* Fix $x$, denote $u := f_\theta(x) = v^T \phi(h_T(x))/\sqrt{n}$. By Theorem 1, we have $u$ converges in distribution to a centered Gaussian random variable with variance $\sigma^2 := \Sigma^*(x,x)$. Hence, given $\delta > 0$, we have there exists $n_\delta$ such that $n \ge n_\delta$ implies

$$|P(u \ge \varepsilon) - P(z \ge \varepsilon)| \le \delta/2,$$

where $z \sim \mathcal{N}(0, \sigma^2)$. Then we have

$$P(u \ge \varepsilon) \le \delta/2 + P(z \ge \varepsilon) \le \delta/2 + e^{-\varepsilon^2/2\sigma^2} \le \delta,$$

where the last inequality is due to $\varepsilon := \sigma\sqrt{2\log(2/\delta)}$. Similarly, we obtain two two-tailed bound, *i.e.*,

$$P(|u| \ge \varepsilon) \le \delta.$$

Now, denote $u = f_\theta(X) \in \mathbb{R}^N$ as a vector. We have

$$P(\|u\| \ge \varepsilon_0) = P(\|u\|^2 \ge \varepsilon_0^2) = P(\sum_{i=1}^N |u_i|^2 \ge \varepsilon_0^2)$$

$$\le \sum_{i=1}^N P(|u_i|^2 \ge \varepsilon_0^2/N) = \sum_{i=1}^N P(|u_i| \ge \varepsilon_0/\sqrt{N})$$

$$\le \delta,$$

where we use the fact $P(\sum_{i=1}^N x_i \ge \varepsilon) \le \sum_{i=1}^N P(x_i \ge \varepsilon/N)$ and $\varepsilon_0 := \sigma\sqrt{2N \log(N/\delta)}$. $\square$

## H  ADDITIONAL EXPERIMENTS

In this appendix, we provide supplementary experiments that complement the results in the main paper. These experiments explore the impact of different activation functions, scaling for long time horizons, and the behavior of Neural ODEs when approximated by Gaussian processes. Additionally, we examine the behavior of the NTK when using polynomial activations.

## H.1 Scaling for Long-Time Horizons

As discussed in Proposition 1 and Proposition 2, smooth activations ensure that the forward and backward dynamics of Neural ODEs have globally unique solutions. However, extending the time range or working with long-time horizons in the dynamics can introduce difficulties for numerical solvers, leading to higher numerical errors. To understand how Neural ODEs behave over extended time horizons, we investigated their behavior at initialization as the time horizon increases, focusing on how output magnitudes and variance are affected. The objective was to understand how extending the time horizon impacts the model's outputs and the subsequent training process.

At initialization, as the time horizon $T$ increases, the output magnitudes grow larger, resulting in increased variance, as shown in Figure 3(a). This increased variance negatively impacts the training of Neural ODEs with gradient descent, leading to damping in the early stages of training as illustrated in Figure 3(b). We observed that increasing the width of neural networks reduced the output variance, as shown in Figure 3(a). Additionally, for long-time horizons, where $T$ is large, we suggest scaling the dynamics by setting the weight variance $\sigma_w \sim 1/T$. This approach effectively mitigates the growth in output magnitudes, as shown in Figure 3(c), and reduces early-stage damping during training, as illustrated in Figure 3(d).

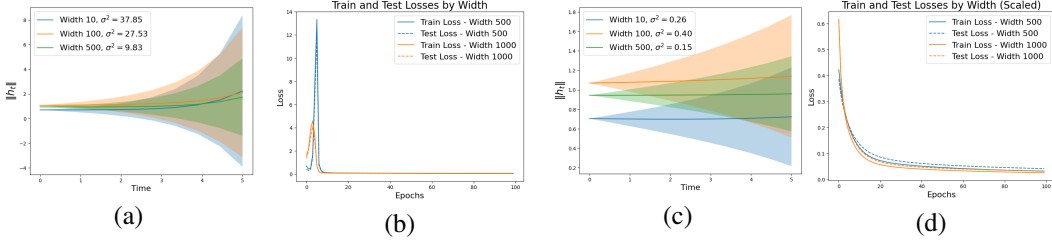

Figure 3: Effects of increasing time horizons on Neural ODE outputs and training. (a) Output variance increases as the time horizon $T$ becomes large at initialization. (b) This leads to damping during the early stages of training with gradient descent. (c) Scaling the dynamics by setting the weight variance $\sigma_w \sim 1/T$ reduces the output variance. (d) This scaling also mitigates early-stage damping, improving training stability.

## H.2 Gaussian Process Approximation

In Section 4, we established that Neural ODEs tend toward a Gaussian Process (GP) as their width increases, as demonstrated in Theorem 1. The associated NNGP kernel of this Gaussian process is non-degenerate, as stated in Lemma 5. To empirically verify these theoretical findings, we conducted a series of experiments.

First, we fixed an input $x$ and initialized 10,000 random Neural ODEs. We then plotted the output histograms for various network widths and fitted the distributions with a Gaussian model, as shown in Figure 4. Additionally, we ran statistical tests to confirm whether the output distributions followed a Gaussian distribution. The Kolmogorov-Smirnov (KS) test statistics and p-values indicated that as long as the width exceeds 100, the outputs closely follow a Gaussian distribution.

Next, we analyzed the independence of the output neurons by plotting pairwise outputs across two coordinates. According to Theorem 1, the output neurons should become independent as the width increases. Figure 5 confirms this: while the diagonal plots show Gaussian bell shapes, the off-diagonal plots resemble random ball shapes, indicating that the neurons are uncorrelated and, therefore, independent as the width increases.

Finally, we investigated whether Neural ODEs preserve the structure of input data at the output. We constructed a matrix $X$ of 10 samples and calculated the input covariance matrix $XX^\top$. Then, we initialized 10,000 Neural ODEs with random weights and evaluated them on the input $X$, computing the output covariance matrix. As shown in Figure 6, the output covariance matrix retained the correlation patterns of the input matrix but with reduced magnitudes, indicating that Neural ODEs act as structure-preserving smoothers, reducing the spread of the data while maintaining its underlying relationships.

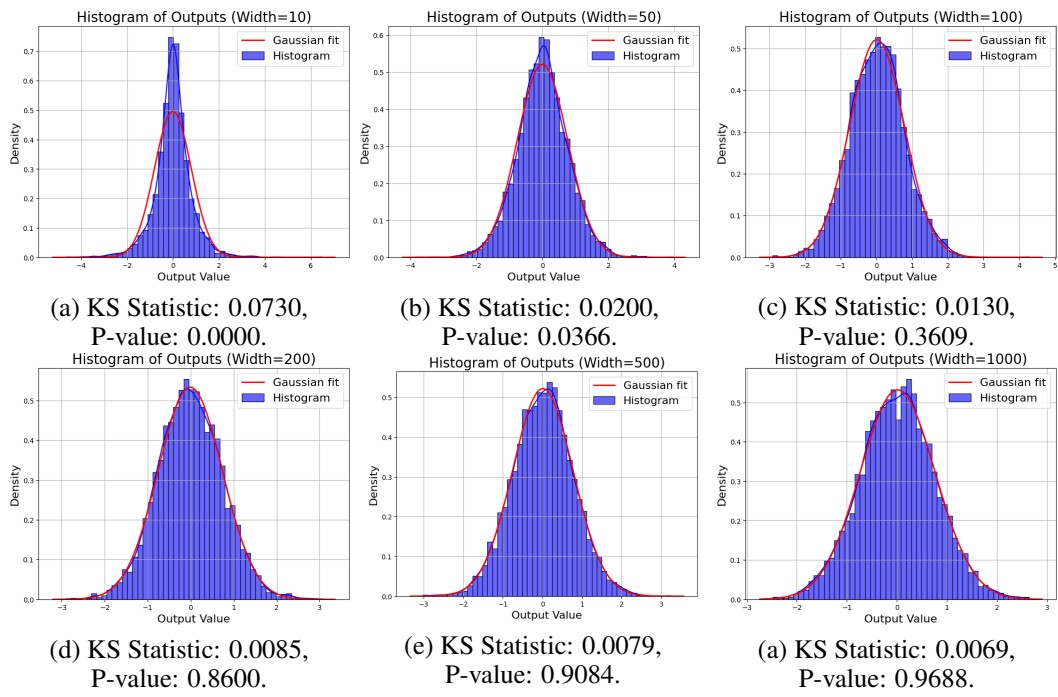

Figure 4: Gaussian fit of the sample distribution from 10,000 randomly initialized Neural ODEs across widths 10, 50, 100, 200, 500, and 1000. The corresponding KS statistics and p-values are displayed, showing improved Gaussian fit as width increases.

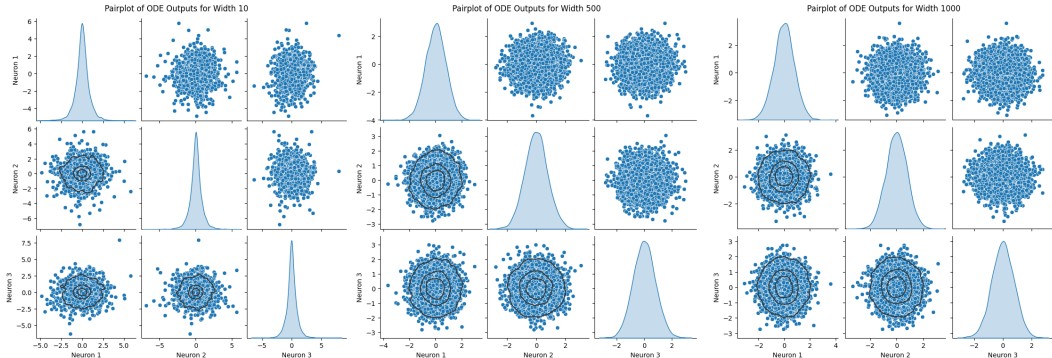

Figure 5: Pairplots of output neurons given the same input data, showing that output neurons become independent as network width increases.

Additionally, we computed the smallest eigenvalues of the output covariance matrix. By sampling 500 examples from the MNIST training set and computing the covariance matrix for 10,000 independently initialized Neural ODEs, we found that the smallest eigenvalues became strictly positive when the width exceeded the number of samples. This result indicates that the NNGP and NTK kernels are strictly positive definite, aligning with our theoretical findings.

### H.3    SMOOTH VS. NON-SMOOTH ACTIVATIONS

To compare the performance of smooth and non-smooth activation functions, we evaluated both Softplus (smooth) and ReLU (non-smooth) across different widths, measuring their training and test losses, parameter distances, and NTK least eigenvalues. The results highlight several key differences between these activations.

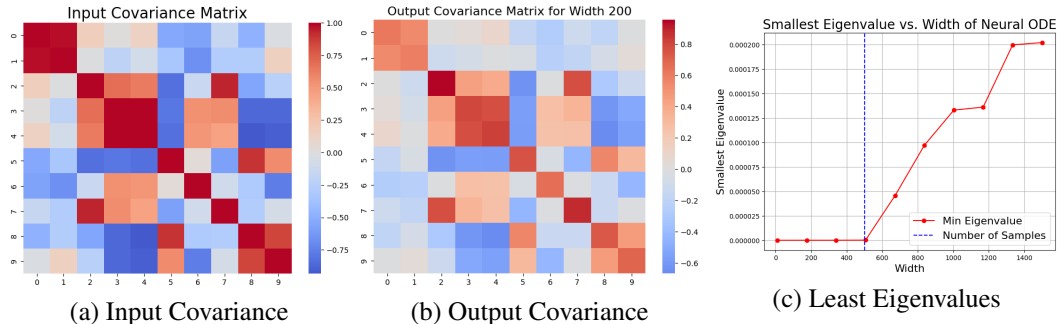

(a) Input Covariance      (b) Output Covariance      (c) Least Eigenvalues

Figure 6: Comparison of input and output covariance matrices. (a) Input covariance matrix. (b) Output covariance matrix from Neural ODEs, showing similar structure but reduced variance. (c) Least eigenvalues of the covariance matrices, confirming positive definiteness as width increases.

**Training and Test Loss Behavior:** As illustrated in Figure 7(a)-(b), the log-scale plots show that Softplus consistently converges faster than ReLU. ReLU experiences slower convergence, particularly during the early stages of training, while Softplus benefits from smoother and faster optimization, especially at larger widths.

**Parameter Distance:** We also measured the distance between the model parameters and their initial values throughout training. As shown in Figure 7(c), parameters remained relatively closer to their initialization for Softplus, while the parameter distance for ReLU was significantly higher. This suggests that Softplus's smoother nature results in more stable parameter updates during training, contributing to its faster convergence.

**NTK Least Eigenvalues:** Regarding the NTK least eigenvalues, the results shown in Figure 7(d) indicates both activations exhibited strictly positive eigenvalues, with ReLU's slightly larger than Softplus's. However, despite this, Softplus converged more rapidly based on both training and test loss results. We hypothesize that Softplus's smoothness allows for more accurate gradient computation by the numerical solver, leading to more efficient optimization compared to ReLU, which may suffer from less precise gradient calculations due to its non-smooth nature.

In summary, Softplus not only converges faster in terms of loss, but it also leads to more stable parameter behavior during training, despite the slightly smaller NTK least eigenvalues compared to ReLU. These findings suggest that the smoother nature of Softplus provides significant advantages for Neural ODE training.

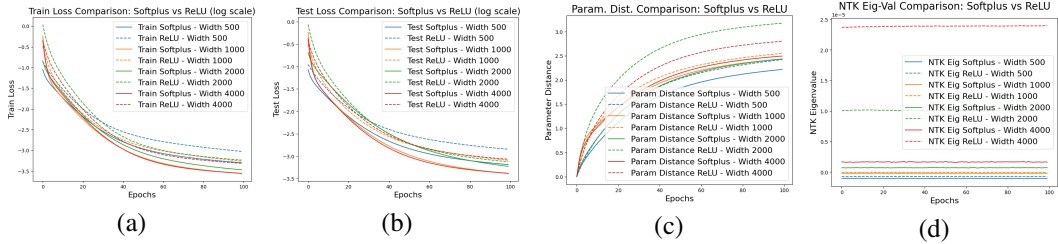

(a)      (b)      (c)      (d)

Figure 7: Comparison of Smooth (Softplus) and Non-Smooth (ReLU) Activation Functions across Neural ODE Training. (a) Log-scale plot of training loss shows faster convergence for Softplus compared to ReLU, especially at larger widths. (b) Log-scale plot of test loss similarly demonstrates faster convergence for Softplus, with ReLU exhibiting slower progress in the early stages. (c) Parameter distance from initialization, showing that Softplus keeps parameters closer to their initial values, suggesting more stable updates. (d) NTK least eigenvalues for both activations, where ReLU's eigenvalues are slightly larger, though Softplus achieves faster overall convergence.

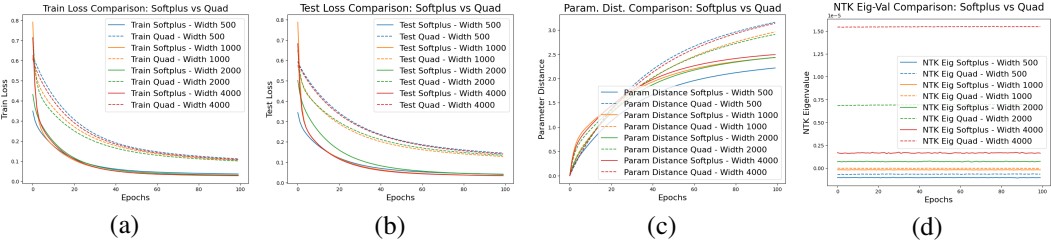

Figure 8: Comparison of Neural ODEs with Softplus (non-polynomial) and Quadratic (polynomial) activation functions. (a) Training loss for Softplus and Quadratic activations, showing slower convergence for the Quadratic case. (b) Test loss comparison, further illustrating the slower convergence for Quadratic activations. (c) Parameter differences from initialization, showing that Quadratic activations lead to slightly larger parameter drift compared to Softplus. (d) NTK least eigenvalues, where both activations show strictly positive eigenvalues, with Quadratic's being slightly larger than Softplus's.

## H.4  POLYNOMIAL ACTIVATIONS FOR NTK AND GLOBAL CONVERGENCE

In this experiment, we tested quadratic activation functions to assess their impact on NTK behavior and convergence. While previous results suggested that nonlinearity but non-polynomiality is a sufficient condition for the strict positive definiteness (SPD) of the NTK, our experiments reveal that it is not a necessary condition.

We observed that the NTK of Neural ODEs using quadratic activation is also strictly positive definite, with the smallest eigenvalue slightly higher than that of Softplus, as shown in Figure 8(d). Despite this, the quadratic Neural ODE converged much more slowly than Softplus, as illustrated in the training and test losses (Figure 8(a)-(b)).

In terms of parameter behavior (Figure 8(c)), the parameter differences for the quadratic activation were slightly larger than those for Softplus, meaning the parameters drifted further from their initial values. However, these differences remained within the same order of magnitude, indicating that the model still satisfies the conditions for global convergence, even though it does not meet the sufficient condition of being non-polynomial.

In summary, while quadratic activation functions result in strictly positive definite NTKs similar to non-polynomial activations, they lead to slower convergence and slightly less stable parameter behavior compared to smoother activations like Softplus. This suggests that while non-polynomiality is not strictly necessary for SPD and convergence, smoother activations may offer practical benefits for faster and more stable training.

## H.5  CONVERGENCE ANALYSIS ON DIVERSE DATASETS

In the main paper, we focused on the convergence properties of Neural ODEs using different activation functions on the MNIST dataset. To ensure that these findings generalize across different types of data and tasks, we extended our experiments to three additional datasets: CIFAR-10 (image classification), AG News (text classification), and Daily Climate (time series forecasting). This section details the performance of three key activation functions—Softplus, ReLU, and GELU—on these datasets, highlighting their effects on convergence speed, stability, and generalization.

For each dataset, we trained Neural ODE models with different widths (i.e., 500, 1000, 2000, 3000) using Softplus, ReLU, and GELU activations. We monitored the training loss and test loss, comparing how different activations influence convergence behavior across datasets. The optimizer used was gradient descent with a learning rate of 0.1, and models were trained for 100 epochs.

For CIFAR-10, the results showed minimal differences between the activation functions.

- Softplus, ReLU, and GELU all exhibited similar convergence patterns, with larger widths leading to faster convergence across the board.

- Larger widths consistently resulted in lower training and test losses, but the specific choice of activation did not have a significant impact on the overall performance or convergence speed.

These results suggest that for CIFAR-10, the activation function choice is less critical, particularly when the network is sufficiently wide, *i.e.*, see Figure 9.

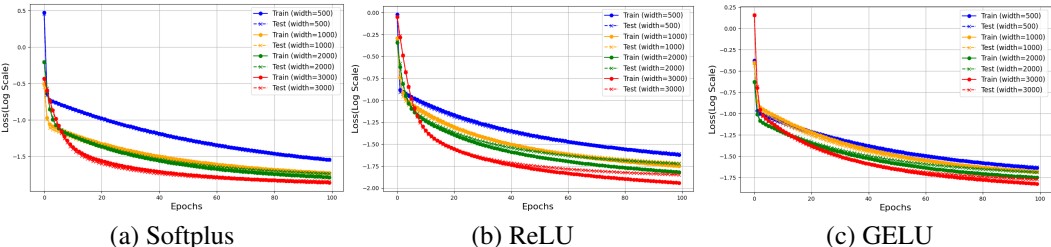

| (a) Softplus | (b) ReLU | (c) GELU |

Figure 9: Training and test loss behavior for CIFAR-10 across different activations: (a) Softplus, (b) ReLU, and (c) GELU. All activations show similar convergence patterns, with larger widths leading to faster convergence.

For AG News, we observed distinct convergence patterns across the activation functions:

- Softplus converged the fastest, followed by ReLU, with GELU converging the slowest. Despite GELU being a smooth activation, its derivative differs significantly compared to the other activations, which may explain the slower convergence rate.

- All three activations shared the same trend: larger widths led to faster convergence and lower test losses. However, the differences between activation functions were more pronounced at smaller widths, where GELU lagged behind (Figure 10).

This suggests that while GELU's smoothness offers theoretical benefits, in practice, its derivative may cause slower optimization dynamics, particularly for text-based tasks like AG News.

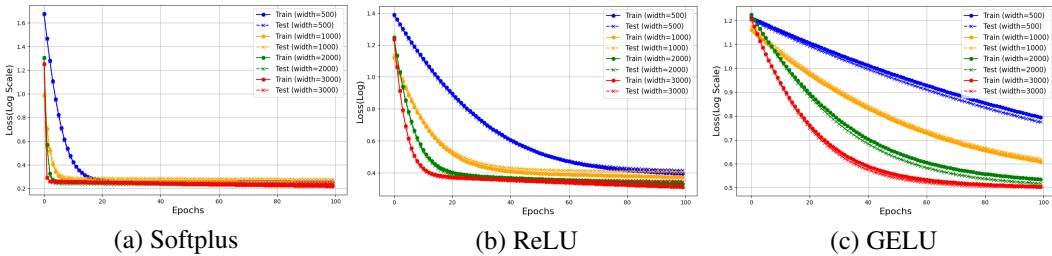

| (a) Softplus | (b) ReLU | (c) GELU |

Figure 10: Training and test loss behavior for AG News across different activations: (a) Softplus, (b) ReLU, and (c) GELU. Softplus converges fastest, while GELU lags due to its derivative behavior.

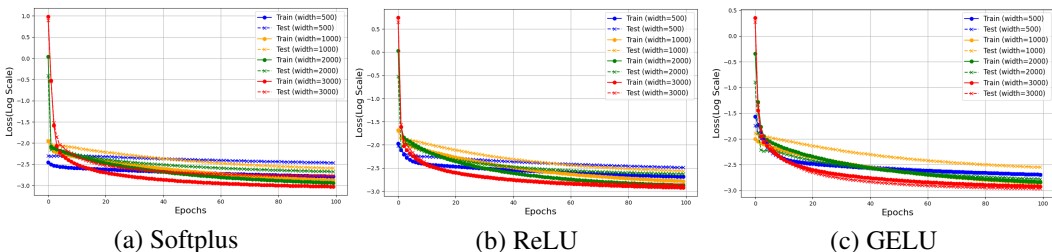

| (a) Softplus | (b) ReLU | (c) GELU |

Figure 11: Training and test loss behavior for Daily Climate time series forecasting: (a) Softplus, (b) ReLU, and (c) GELU. All activations show similar convergence, with larger widths leading to faster results.

## H.6 NON-SMOOTH ACTIVATION FUNCTIONS: COMPARING "DISCRETIZE-THEN-OPTIMIZE" AND "OPTIMIZE-THEN-DISCRETIZE"

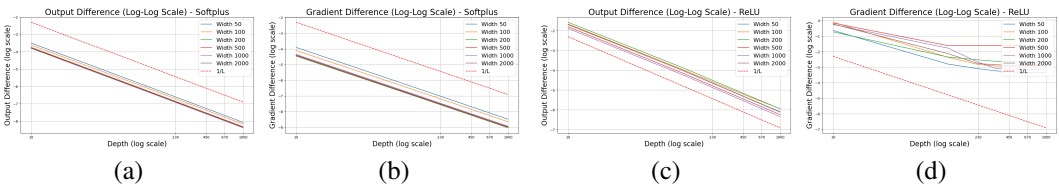

Figure 12: Comparison of Neural ODEs and ResNets with Softplus (smooth) and ReLU (non-smooth) activations. (a, b) Softplus: Output and gradient differences in log-log scale, both showing $1/L$ convergence. (c, d) ReLU: Output difference shows $1/L$ convergence, while gradient difference remains constant as depth $L$ increases.

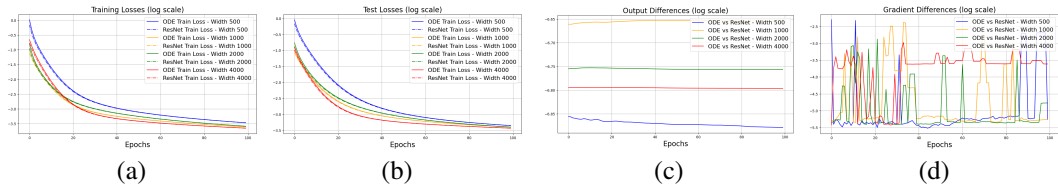

Figure 13: Training comparison of Neural ODEs and ResNets with ReLU (non-smooth) activation. (a) Training loss decreases consistently and aligns closely for both Neural ODEs and ResNets. (b) Test loss shows a similar trend, consistently decreasing for both models. (c) Output difference remains consistently small throughout training. (d) Gradient difference oscillates during training.

In this experiment, we investigate the impact of non-smooth activation functions, specifically ReLU, on the performance of Neural ODEs and their ResNet approximations under the "Discretize-Then-Optimize" and "Optimize-Then-Discretize" frameworks. While the output differences between the two frameworks decrease as the depth $L$ increases, our results reveal that the backward gradients fail to converge due to the non-smooth nature of ReLU's derivative.

**Smooth Activation Functions (Softplus).** For smooth activation functions like Softplus, both the output difference and gradient difference between the two frameworks decrease at a rate of $1/L$ as the depth $L$ increases. This behavior aligns with Proposition 2 and is illustrated in Figure 12(a)-(b).

**Non-Smooth Activation Functions (ReLU).** In contrast, for ReLU, the output difference still decreases at a rate of $1/L$, as shown in Figure 12(c). However, the gradient difference fails to converge, as illustrated in Figure 12(d). Initially, the gradient difference reduces as depth increases, but it eventually stagnates at a fixed error level. Increasing the network width does not resolve this issue. Notably, the largest gradient difference is observed at width 500, whereas smaller errors are achieved for both smaller and larger widths, such as width 200 and 1000. These results confirm that the lack of a continuous derivative in ReLU introduces inconsistencies in gradient computations between the two frameworks.

**Training Dynamics.** Despite this mismatch in gradient computation, we did not observe significant differences in the training dynamics between Neural ODEs and ResNets. We trained Neural ODEs and their finite-depth ResNet approximations (fixed at depth 200, as further depth increases did not reduce errors, as shown in Figure 12(d)) on a subset of MNIST. As illustrated in Figure 13, both models exhibit similar training and test losses. While the output differences remain consistently small during training, the gradient differences oscillate, as shown in Figure 13. ResNets, as finite-depth networks, are known to exhibit global convergence guarantees under gradient descent in overparameterized regimes (Du et al., 2019a), so their convergence is unsurprising. What is unexpected, however, is the near-identical training dynamics between Neural ODEs and ResNets despite the gradient mismatch caused by ReLU's non-smoothness. Our hypothesis is that while gradient differences oscillate during training, they remain within small magnitudes because MNIST is a simple dataset and ReLU's derivative is almost continuous everywhere except at the origin. This partial

smoothness may mitigate the adverse effects of the gradient mismatch. However, we anticipate that in realistic applications involving more complex datasets, these differences could lead to divergent training trajectories and dynamics for Neural ODEs and ResNets using non-smooth activations.

## H.7 Sensitivity of "Optimize-then-discretize" to ODE Solvers

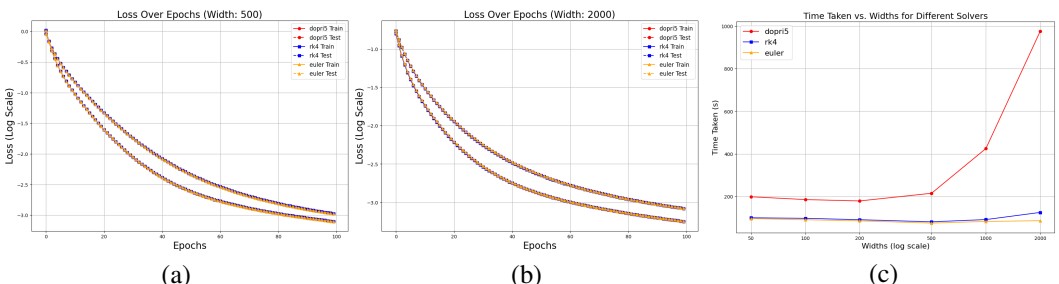

(a)                                      (b)                                      (c)

Figure 14: Sensitivity of "Optimize-then-discretize" to ODE solvers. (a) Training and test losses decrease consistently for all three solvers at width 500. (b) Training and test losses decrease consistently for all three solvers at width 2000. (c) Time taken by each ODE solver across different widths, highlighting the scalability advantage of fixed-step solvers.

In this subsection, we investigate the impact of different numerical ODE solvers on the accuracy of gradient computation and overall training dynamics in the "Optimize-then-discretize" framework. The solvers considered in our experiments are Euler, rk4, and dopri5.

As illustrated in Figure 14(a)-(b), the choice of ODE solver does not significantly affect the accuracy of gradient computation or the overall training dynamics in our specific setting. This is consistent with the theoretical guarantees established in Proposition 1, where we demonstrated that the ODE dynamics in Equations Eq. (2) and Eq. (4) possess globally unique solutions under the smoothness conditions on activation functions. Given the relatively simple nature of the system studied, the numerical errors introduced by the solvers appear to be negligible in this context. However, this observation may not generalize to more complex systems or practical applications where numerical errors can be influenced by other factors such as stiffness or stability in the dynamics, which are beyond the scope of this paper.

An interesting observation from our experiments is the computational efficiency of the solvers. While adaptive solvers like dopri5 provide high accuracy, they require significantly more computation time as the neural network width increases. In contrast, fixed-step methods such as Euler and rk4 scale more efficiently with width, making them preferable in scenarios where computational cost is a concern. This is illustrated in Figure 14(c), where we compare the time taken by the solvers across different widths.

## I Discussion on General Dynamic Form in Neural ODEs

In this section, we discuss extending our results from the specific form equation 1 and equation 2 to a more general dynamic formulation. Specifically, we first consider a generalized nonlinear transformation:

$$\dot{\boldsymbol{h}}_t = \frac{\sigma_w}{\sqrt{n}} \boldsymbol{W} \boldsymbol{f}(\boldsymbol{h}_t, t), \quad \forall t \in [0, T],$$

where the original nonlinear activation function $\phi$ in equation 2 is replaced by a general nonlinear mapping $\boldsymbol{f} : \mathbb{R}^n \times \mathbb{R} \to \mathbb{R}^n$, defined as $\boldsymbol{f} : (\boldsymbol{h}, t) \mapsto \boldsymbol{f}(\boldsymbol{h}, t)$. This generalization introduces explicit time dependence, transforming the system from an *autonomous* to a *non-autonomous* system. Non-autonomous systems are prevalent in applications such as diffusion models (Song et al., 2020) and physics-informed neural networks (PINNs) (Sholokhov et al., 2023). The function $\boldsymbol{f}$ can represent another shallow neural network or more complex operations, such as convolution layers (LeCun et al., 1998), gating mechanisms (Hochreiter, 1997), attention mechanisms (Vaswani, 2017), or batch normalization (Ioffe, 2015).

For this generalized form, the backward dynamics take the form:

$$\dot{\boldsymbol{\lambda}}_t = -\frac{\sigma_w}{\sqrt{n}} \boldsymbol{J}(\boldsymbol{h}_t, t) \boldsymbol{W}^\top \boldsymbol{\lambda}_t,$$

where $\boldsymbol{J} = \partial \boldsymbol{f}/\partial \boldsymbol{h} \in \mathbb{R}^{n \times n}$ is the Jacobian matrix of $\boldsymbol{f}$ with respect to $\boldsymbol{h}$. By Theorem 5, the forward ODE has unique global solutions if $\boldsymbol{f}$ is continuous in $t$ and Lipschitz continuous in $\boldsymbol{h}$, with a Lipschitz constant independent of $t$. This generalizes the continuity requirement for the activation function $\phi$ to $\boldsymbol{f}$. Additionally, if the Jacobian matrix $\boldsymbol{J}$ is globally bounded, the backward ODE also admits a unique global solution. Since $\boldsymbol{f}$ is Lipschitz continuous in $\boldsymbol{h}$, the boundedness of $\boldsymbol{J}$ is naturally satisfied (on a compact set). Therefore, appropriate smoothness conditions on $\boldsymbol{f}$ ensure well-posed forward and backward dynamics with unique solutions.

Using Euler's method, we discretize the forward and backward dynamics as follows:

$$\boldsymbol{h}^{\ell+1} = \boldsymbol{h}^\ell + \kappa \cdot \frac{\sigma_w}{\sqrt{n}} \boldsymbol{W} \boldsymbol{f}(\boldsymbol{h}^\ell, t_{n_\ell}),$$

$$\boldsymbol{\lambda}^{\ell+1} = \boldsymbol{\lambda}^\ell - \kappa \cdot \frac{\sigma_w}{\sqrt{n}} \boldsymbol{J}(\boldsymbol{h}^\ell, t_{n_\ell}) \boldsymbol{W}^\top \boldsymbol{\lambda}^\ell,$$

where $\kappa = T/L$. Ensuring convergence of $(\boldsymbol{h}^\ell, \boldsymbol{\lambda}^\ell)$ to $(\boldsymbol{h}_t, \boldsymbol{\lambda}_t)$ is critical to aligning the gradients obtained from the "discretize-then-optimize" and "optimize-then-discretize" methods. As discussed in Proposition 2, additional smoothness of the backward ODE is required for gradient equivalence. By Theorem 7, the mapping $t \mapsto \boldsymbol{J}(\boldsymbol{h}_t, t)$ must be continuous in $t$, which implies that $\boldsymbol{J}$ is Lipschitz continuous in $\boldsymbol{h}$ with a Lipschitz constant independent of $t$. The smoothness of $\boldsymbol{J}$ with respect to $\boldsymbol{h}$ can be guaranteed by imposing second-order regularity conditions on $\boldsymbol{f}$. Specifically, bounding the Jacobian tensor $\partial \boldsymbol{J}/\partial \boldsymbol{h}$ under suitable norms, such as the operator norm or Frobenius norm, ensures the required regularity. Although $\partial \boldsymbol{J}/\partial \boldsymbol{h}$ represents a higher-order tensor, these regularity conditions allow the gradient consistency results from Proposition 2 to extend seamlessly to this generalized formulation.

Theorem 7 provides not only convergence guarantees but also a uniform convergence rate under globally uniform smoothness conditions. Consequently, by Theorem 8, the iterated limits in Lemma 1 and Lemma 2 converge to the same double limit. As a result, the NNGP and NTK of the generalized Neural ODE remain well-defined. If the limiting NNGP or NTK is strictly positive definite (SPD), global convergence under gradient descent can also be established.

Finally, we discuss extending the dynamics to a post-activation formulation:

$$\dot{\boldsymbol{h}}_t = \boldsymbol{f}\left(\frac{\sigma_w}{\sqrt{n}} \boldsymbol{W} \boldsymbol{h}_t, t\right),$$

where the linear transformation $\boldsymbol{h} \mapsto \frac{\sigma_w}{\sqrt{n}} \boldsymbol{W} \boldsymbol{h}$ is applied before the nonlinear mapping $\boldsymbol{f}$. The analysis remains analogous because the linear transformation is globally 1-Lipschitz continuous under Theorem 4. However, we focus primarily on the pre-activation form, as it consistently achieves superior empirical performance compared to the post-activation formulation (He et al., 2016b).

## J    RELATED WORKS

Neural Ordinary Differential Equations (Neural ODEs) (Chen et al., 2018b) introduced a continuous-depth framework for modeling dynamics by replacing discrete-layer transformations with parameterized differential equations. This innovative framework has since inspired extensive research, leading to both theoretical advancements and practical applications.

Neural ODEs are distinguished by their continuous-time representation and memory efficiency through parameter sharing, setting them apart from traditional architectures like ResNet (He et al., 2016a). Building on this foundation, several extensions have been proposed to address more complex systems. Notable examples include Neural Stochastic Differential Equations (SDEs) for stochastic dynamics (Li et al., 2020), Neural Partial Differential Equations (PDEs) for spatiotemporal systems (Sirignano & Spiliopoulos, 2018; Raissi et al., 2019), Neural Controlled Differential Equations (CDEs) for irregular time-series data (Kidger et al., 2020), and Neural Variational and Hamiltonian Systems for capturing conserved quantities in physical dynamics (Greydanus et al.,

2019). These advanced formulations have broadened the applicability of Neural ODEs to diverse domains, such as time-series modeling (Rubanova et al., 2019), computer vision (Chen et al., 2018b; Park et al., 2021), and reinforcement learning (Du et al., 2020). In generative modeling, Neural SDEs underpin approaches like FFJORD (Grathwohl et al., 2018), score-based methods (Song et al., 2020), and diffusion models (Ho et al., 2020). Similarly, in physics-informed machine learning, Neural PDEs and Physics-Informed Neural Networks (PINNs) have proven critical for solving physical systems while incorporating domain-specific knowledge (Sholokhov et al., 2023; Karniadakis et al., 2021; Raissi et al., 2019). However, while these features offer flexibility and efficiency, they also introduce significant challenges during training.

A key challenge in training Neural ODEs lies in gradient computation. The original adjoint method introduced by Chen et al. (2018b) computes gradients with minimal memory overhead. However, this approach can suffer from numerical instabilities, as observed in Gholaminejad et al. (2019). To address these issues, advanced methods have been developed. For instance, Zhuang et al. (2020a) integrates adjoint techniques with checkpointing to balance memory usage and computational cost, while Matsubara et al. (2021) employs symplectic integrators to preserve ODE structure, ensuring stability in long-time horizons and oscillatory systems. Finlay et al. (2020) regularizes the Jacobian norm of the dynamics to improve stability and generalization. Ko et al. (2023) introduces a homotopy-based approach, starting with simplified dynamics and gradually transitioning to target dynamics. These methods generally follow an "optimize-then-discretize" approach, where (augmented) backward ODEs are solved numerically to compute gradients. Conversely, the "discretize-then-optimize" approach, which discretizes the forward ODE into a finite-depth network for gradient computation via backpropagation, has been explored by Massaroli et al. (2020). However, as noted in Zhuang et al. (2020a;b), this method often results in deeper computational graphs, raising concerns about gradient accuracy.

To address the challenge in gradient computation, several theoretical studies have been conducted, focusing on well-posedness and stability. For instance, Gholaminejad et al. (2019) highlighted significant numerical instabilities when using ReLU activations in Neural ODEs. Meanwhile, Rodriguez et al. (2022) investigated the stability of Neural ODEs through a Lyapunov framework derived from control theory. Despite these advancements, none of these works address when and how the "discretize-then-optimize" and "optimize-then-discretize" methods can yield equivalent gradients. Moreover, the question of whether simple first-order optimization methods, such as stochastic gradient descent, can reliably train Neural ODEs to convergence remains unexplored.

Another essential challenge lies in analyzing the training dynamics of Neural ODEs due to the inherent nonconvexity of neural network optimization. A significant breakthrough in this area came from the Neural Tangent Kernel (NTK) framework introduced by Jacot et al. (2018), which demonstrated that the NTK governs the training dynamics of feedforward networks (FFNs) under gradient descent and converges to a deterministic limit as network width increases. This convergence facilitates global convergence guarantees for gradient-based optimization in overparameterized regimes, provided the NTK remains strictly positive definite (SPD) (Du et al., 2019a; Allen-Zhu et al., 2019; Nguyen, 2021). The strict positive definiteness of the NTK has been extensively studied, beginning with dual activation analysis for two-layer networks (Daniely et al., 2016) and later extended to finite-depth FFNs (Jacot et al., 2018; Du et al., 2019a). Recent work has further applied NTK theory to diverse architectures, including convolutional neural networks (CNNs) (Arora et al., 2019), recurrent neural networks (RNNs) (Yang, 2020), transformers (Hron et al., 2020), physics-informed neural networks (PINNs) (Wang et al., 2022), and graph neural networks (GNNs) (Du et al., 2019b). NTK analysis has also been explored for various optimization methods, such as stochastic gradient descent (SGD) (Zou et al., 2020) and adaptive gradient algorithms (Chen et al., 2018a). A few recent works start studying large-depth neural networks Gao & Gao (2022b;a); Gao (2024). However, applying NTK theory to continuous-depth models like Neural ODEs and determining whether similar SPD and convergence properties hold remains an open and active area of research.

Despite significant advancements, challenges persist in understanding the training dynamics of Neural ODEs and ensuring gradient consistency between the "discretize-then-optimize" and "optimize-then-discretize" approaches. Our work addresses these gaps by:

1. **Gradient Equivalence**: Establishing conditions under which the gradients computed by the two methods are equivalent, as demonstrated in Proposition 1 and Proposition 2, emphasizing the role of smooth activations.

2. **NTK Analysis**: Providing rigorous conditions for the well-definedness of the Neural ODE NTK, demonstrating its strict positive definiteness (SPD) under suitable activation function properties, as stated in Theorem 2 and Corollary 1.

3. **Global Convergence**: Extending global convergence guarantees for gradient descent in overparameterized Neural ODEs, bridging the gap between discrete and continuous-depth models, as outlined in Theorem 3.

