# OpenReview forum: "Global Convergence in Neural ODEs: Impact of Activation Functions"
_ICLR.cc/2025/Conference — ICLR 2025 Oral_

### Official Review · Reviewer_Hphg · 2024-11-02

**Soundness:** 4
**Presentation:** 4
**Contribution:** 3
**Rating:** 8
**Confidence:** 5

**Summary:**

This paper studies the convergence of gradient descent for training neural ODEs from the neural tangent kernel perspectives, with experimental results validating the theoretical claims.

**Strengths:**

(1) The paper extends NTK theory from general regression problems with ReLU nets to Neural ODEs with ResNet architectures. The analysis and results are novel for Neural ODEs.

(2) The empirical results validate the theorems.

**Weaknesses:**

(1) Some prior work has extended NTK theory to analyze CNNs (for image classification tasks), PINNs, Transformers, and various gradient-based algorithms beyond vanilla gradient descent. However, related references are missing.

(2) SGD is more commonly used in practice, making a convergence analysis of SGD particularly valuable and interesting.

(3) The results are specific to ODEs of the form given in Equation (2). Theorem 2 does not hold for PINNs if the underlying PDE is nonlinear. How would Theorem 2 apply if the ODE were not limited to this special case?

(4) This paper claims that the nonlinear and non-polynomial nature of the activation function is crucial for convergence. However, I believe the convergence results (Theorem 3) may also hold with polynomial activation functions, provided that Condition (1) of Assumption 1 is further strengthened to ensure linear independence.

(5) From the title, the topic of the paper is exploring the smoothness of activation functions in neural ODEs. However, after reading your paper, your main contribution is developing the global convergence of GD in training neural ODEs. The title is quite strange...

**Questions:**

See weakness.

---

> ### Author Response · Authors · 2024-11-19
>
> Thank you for your detailed review and for acknowledging the novelty of our analysis and results, particularly in extending NTK theory to Neural ODEs and ResNet architectures. We are also grateful for your recognition of the soundness of our results, the excellent presentation, and the significant contributions of our work. Below, we address your specific comments and weaknesses.
>
> > Weakness (1): Missing Related References
>
> We sincerely thank the reviewer for pointing out these important prior results on NTK theory. In the revised manuscript, we have incorporated references to NTK analyses for various architectures and methods, including CNNs, RNNs, PINNs, attention mechanisms, GNNs, and optimization techniques such as SGD and adaptive gradient descent.
>
> To further contextualize our contributions within the broader NTK literature, we have also included a detailed **Related Works section**. However, due to page limit, we moved this comprehensive discussion to the appendix. We believe this approach balances the need to present our main theoretical contributions in the primary text while properly acknowledging foundational work in the field. We greatly appreciate this suggestion, which has helped us enhance the clarity and completeness of our manuscript.
>
> > Weakness (2): Convergence Analysis for SGD
>
> We agree with the reviewer on the importance of analyzing SGD, as it is widely used in practice. While our study focuses on gradient descent (GD) as a foundational step, this is significant since no theoretical convergence analysis for Neural ODEs under any gradient-based method has been established. Given that SGD is an approximation of GD using mini-batches, our results provide a crucial basis for studying SGD and other advanced optimizers. We recognize the practical value of such analyses and consider extending our work to SGD as a natural and important direction for future research.
>
> > Weakness (3): Generalization Beyond Specific ODE Forms
>
> Thank you for raising this important point regarding the applicability of our theoretical results, such as Theorem 2, to more general ODEs or PINNs with nonlinear PDEs. While our current analysis assumes the specific ODE form in Eq. (2), we believe the results have the potential for extension to broader setups. To ensure such generalization, it is crucial to establish that the forward and backward dynamics remain **well-defined**, i.e., they have globally unique solutions and exhibit **uniform convergence** of the Euler method.
>
> To address this, we have added a new **Discussion Section** in the appendix of the revised version, where we explore the extension of our results to generalized ODE forms. In particular, we examine the pre-activation form:
> $$
> \begin{align*}
>     \dot{h} = \frac{\sigma_w}{\sqrt{n}} W f(h, t),\quad \forall t \in [0, T],
> \end{align*}
> $$
> where $f:\mathbb{R}^{n}\times \mathbb{R}\rightarrow \mathbb{R}^n$ is a general nonlinear mapping that replace the activation function
> $\phi$ in Eqution (2). Here, the explicit time-dependence transforms the system from *autonomous* to *non-autonomous*, a framework widely used in PINNs and diffusion models. This general nonlinear mapping  *f*  encompasses components like shallow neural networks, layer normalization, attention, or gating mechanisms.
>
> For the generalized form, the backward dynamics become:
>
> $$
> \dot{\lambda} = -\frac{\sigma_w}{\sqrt{n}} J(h, t) W^\top \lambda,
> $$
>
> where $J=\partial f/\partial h\in \mathbb{R}^{n\times n}$ is the Jacobian of $ f $ w.r.t. $ h $. By Picard-Lindelöf Theorem and Euler's convergence theorem, the regularity conditions we require for activation functions $\phi$ in this paper must extend to the nonlinear map $f$. For example, ensuring the backward dynamics have a globally unique solution necessitates that $J$ is globally bounded. Furthermore, gradient alignment between the “optimize-then-discretize” and “discretize-then-optimize” methods additionally requires  $J$  to be Lipschitz continuous in $h$, under suitable norms such as Frobenius or operator norms.
>
> In the Discussion Section, we also examine the post-activation form:
> $$
> \begin{align*}
>     \dot{h} = f\left(\frac{\sigma_w}{\sqrt{n}} W h, t\right),
> \end{align*}
> $$
> where the linear transformation $ h \mapsto \frac{\sigma_w}{\sqrt{n}} W h $ is applied before the nonlinear mapping $ f $. The analysis remains analogous because the linear transformation is globally $ 1 $-Lipschitz continuous under Bai-Yin law from random matrix theory. However, we focus on the pre-activation form due to its superior empirical performance [1] compared to the post-activation formulation.
>
> We hope this discussion addresses the broader applicability of our theoretical framework and highlights potential extensions to more general ODEs and PINN setups. Thank you for encouraging us to elaborate on this important aspect of our work.
>
> [1] Identity mappings in deep residual networks, ECCV2016

---

> > ### Author Response · Authors · 2024-11-19
> >
> > > Weakness (4): Convergence with Polynomial Activations
> >
> > Your insight into convergence with polynomial activation functions is greatly appreciated. Our experiments (Appendix H.4) demonstrate that quadratic activations can still satisfy the SPD property of the NNGP or NTK, indicating that non-polynomial nonlinearity is a sufficient but not necessary condition. However, as shown in Figure 8, we observe slower convergence with quadratic activation compared to non-polynomial activations like Softplus and ReLU. Thus, while non-polynomial activations are not strictly required, we recommend them to achieve faster convergence. We also agree that strengthening Condition (1) of Assumption 1 to enforce linear independence could facilitate convergence in such cases. We have updated the discussion in the paper to incorporate this perspective and provide further clarification. Thank you for highlighting this important point.
> >
> > > Weakness (5): Title and Focus
> >
> > Thank you for highlighting the potential misalignment between the title and the paper’s focus. While our main contribution is the global convergence analysis of gradient descent (GD) for training Neural ODEs, the smoothness and nonlinearity of activation functions are integral to both our theoretical findings and empirical studies. Due to space constraints, these empirical studies have been detailed in Appendix H.1–H.7.
> >
> > For instance, as shown in Figure 12 (Appendix), nonsmooth activations like ReLU fail to align gradients between the “discretize-then-optimize” and “optimize-then-discretize” frameworks, suggesting continuous differentiability is likely necessary for gradient alignment. Additionally, Figure 8 (Appendix) shows that quadratic activations satisfy the SPD property of NTK, indicating non-polynomial nonlinearity is sufficient but not necessary. However, non-polynomial activations like Softplus achieve significantly faster convergence, underscoring their practical advantages.
> >
> > To better reflect this balance, we are open to revising the title in the final version. For instance, a potential revision could be: “Global Convergence of Gradient Descent for Neural ODEs: Impact of Activation Functions on Training Dynamics.” We hope this adjustment more accurately reflects the paper’s content and scope.

---

> ### Comment · Reviewer_Hphg · 2024-11-22
>
> Thanks for the rebuttal. I am fully satisfied with your responses and the updated version of the paper (including the title modification). I am pleased to raise my scores to 8.

---

> > ### Author Response · Authors · 2024-11-22
> >
> > Thank you for your feedback and support! We’re glad our revisions addressed your concerns and truly appreciate the raised score!

---

### Official Review · Reviewer_Zcba · 2024-11-03

**Soundness:** 3
**Presentation:** 3
**Contribution:** 3
**Rating:** 8
**Confidence:** 4

**Summary:**

The authors studied activation functions in the training dynamics of neural ODEs. They showed that the choice of activation function has significant impacts on the dynamic and asymptotic behavior. Additionally, they propose a new mathematical framework to study continuous models from the perspective of approximation theory. Finally, the authors conduct experiments to empirically validate their theoretical results.

**Strengths:**

The paper is well-written. The ideas presented are novel and represent a significant contribution in the context of neural ODEs. The main strength is the theoretical foundation of the work.

**Weaknesses:**

The proofs are generally well-written and clear, but since they are in the appendix, the flow of the paper's main narrative can sometimes be interrupted.

**Questions:**

No questions.

---

> ### Author Response · Authors · 2024-11-19
>
> Thank you very much for your thoughtful review and high evaluation of our work. We deeply appreciate your recognition of the novelty and significance of our contributions, as well as your positive assessment of the theoretical foundation and empirical validation. Your feedback is incredibly encouraging and motivates us to continue refining our work.

---

> > ### Comment · Reviewer_Zcba · 2024-11-25
> > **Response**
> >
> > Thank you for the reply! My concerns are addressed.

---

> > > ### Author Response · Authors · 2024-11-25
> > >
> > > Thank you for your feedback! We’re glad we could address your concerns.

---

### Official Review · Reviewer_E13D · 2024-11-04

**Soundness:** 3
**Presentation:** 3
**Contribution:** 3
**Rating:** 8
**Confidence:** 3

**Summary:**

This paper investigates the role of activation functions in the training dynamics of Neural ODEs, particularly focusing on the convergence and stability properties of two key methods: "Optimize-then-discretize" and "Discretize-then-optimize." The authors establish theoretical guarantees for the convergence of both methods under certain conditions on the activation functions, and they provide numerical experiments to support their findings.

**Strengths:**

Theoretical Analysis: The paper offers a rigorous mathematical analysis of the well-posedness of Neural ODE solutions and their gradients, focusing on the impact of activation function smoothness. The authors clearly demonstrate the importance of Lipschitz continuity for both the activation function and its derivative to ensure stable and accurate gradient computation.

NTK Analysis: The paper extends the concept of the Neural Tangent Kernel (NTK) to the continuous setting of Neural ODEs. This is a valuable contribution, as it provides insights into the training dynamics of these models. The authors show that the NTK converges to a deterministic limit under certain conditions, which helps explain the convergence properties of gradient descent in Neural ODE training.

Global Convergence Proof: The authors establish the global convergence of Neural ODEs under gradient descent in the overparameterized regime, under assumptions about the activation function's nonlinearity and non-polynomial nature. This theoretical result provides a strong foundation for understanding the optimization behavior of Neural ODEs.

Empirical Validation: The paper presents several numerical experiments to validate the theoretical findings. These experiments demonstrate the convergence of Neural ODE outputs and gradients to their finite-depth ResNet approximations, the behavior of the NTK, and the convergence properties of Neural ODEs under gradient descent.

**Weaknesses:**

Lack of Demonstration for "Discretize-then-optimize" Convergence Issue: While the paper theoretically analyzes the convergence problem of "Discretize-then-optimize" with non-smooth activation functions, it does not provide a dedicated numerical experiment to showcase this issue. This would be valuable for empirically confirming the theoretical claims and for understanding the practical implications of using non-smooth activations with this method.

The lack of a numerical demonstration of the "Discretize-then-optimize" convergence problem weakens the overall impact of the paper. Without empirical evidence, the theoretical claims regarding the convergence issue remain unsubstantiated, and the practical implications of using non-smooth activation functions with this method remain unclear.

**Questions:**

1.
Could you provide a numerical experiment that specifically highlights the convergence problem of the "Discretize-then-optimize" method when using an activation function that does not meet the Lipschitz continuity requirements? This would strengthen the paper by providing empirical support for your theoretical analysis.

2.
How does the choice of numerical ODE solver in the "Optimize-then-discretize" method impact the accuracy of the gradient computation and the overall training dynamics? Exploring the sensitivity of the method to different solvers would be valuable.

3.
What are the practical implications of your findings for the choice of activation functions in real-world applications of Neural ODEs? Providing guidance on activation function selection based on the desired trade-off between theoretical guarantees and empirical performance would be beneficial.

Addressing these points would significantly improve the paper and provide a more comprehensive understanding of the impact of activation functions on Neural ODE training.

---

> ### Author Response · Authors · 2024-11-19
>
> We sincerely thank the reviewer for the detailed feedback and for acknowledging the theoretical rigor and contributions of our work, including the NTK analysis, global convergence proof, and empirical validation. Below, we address each of the points raised.
>
> > Lack of Demonstration for “Discretize-then-optimize” Convergence Issue
>
> We appreciate your observation regarding the need for a dedicated numerical experiment to demonstrate the convergence issue of the “Discretize-then-optimize” method with *non-smooth* activation functions. To address this, we have conducted additional experiments, as detailed in *Appendix H.6* of the revised manuscript. Specifically, we compare the output and gradient differences between the “Discretize-then-optimize” and “Optimize-then-discretize” frameworks using ReLU (non-smooth) and Softplus (smooth) activations.
>
> Our findings, illustrated in Figure 12, show that while the output differences reduce with increasing depth L, the gradient differences for ReLU fail to converge, confirming our theoretical claims. These results underscore the importance of continuous differentiability for gradient alignment, as discussed in the main text.
>
> Interestingly, despite the gradient mismatch, we observed no significant differences in training dynamics between Neural ODEs and their finite-depth ResNet approximations. In experiments where both models were trained (ResNets fixed at depth 200, as further depth increases did not reduce errors, as shown in Figure 12(d)) on a subset of MNIST, their training and test losses were nearly identical, as illustrated in Figure 13. While output differences remained consistently small, gradient differences oscillated during training.
>
> ResNets, as finite-depth networks, are known to achieve global convergence under gradient-based methods in overparameterized regimes [1], making their behavior expected. However, the near-identical training dynamics of Neural ODEs and ResNets, despite the gradient mismatch caused by ReLU’s non-smoothness, were surprising. We hypothesize that the small magnitude of gradient differences, attributed to MNIST’s simplicity and ReLU’s near-continuous derivative (except at the origin), mitigates the adverse effects of the mismatch. Nevertheless, we anticipate that with more complex datasets, these gradient discrepancies could lead to divergent training trajectories.
>
> > Sensitivity to Numerical ODE Solvers
>
> Your suggestion to explore the sensitivity of the “Optimize-then-discretize” method to different numerical ODE solvers is highly valuable. In response, we have included a new experiment comparing solvers such as Euler, RK4, and Dopri5 in Appendix H.7. Our results show that while all solvers maintain consistent accuracy due to the well-posedness of the ODE dynamics (as proven in Proposition 1), the computational time varies significantly. Adaptive solvers like Dopri5 are slower and scale poorly with increasing neural network width, whereas fixed-step solvers like Euler and RK4 are more scalable. These findings highlight the trade-off between solver choice and computational efficiency in practice.
>
> > Practical Implications for Activation Function Selection
>
> We agree that providing practical guidelines for activation function selection is important. Our results suggest the following:
> - Smooth and Non-Polynomial Activations: Smooth activations (e.g., Softplus) ensure gradient alignment and faster convergence, making them preferable for applications requiring high accuracy and robust optimization.
> - Non-Smooth Activations: Activations like ReLU may lead to convergence issues in gradient alignment, as demonstrated in Figure 12. Thus, they are less suitable for scenarios requiring precise gradient computations.
> - Polynomial Activations: While our experiments in Figure 8 (Appendix H.4) show that polynomial activations (e.g., quadratic) maintain the SPD property of NTK, their slower convergence suggests they may be less effective in practice.
>
> [1] Gradient descent finds global minima of deep neural networks. ICML 2019

---

> > ### Comment · Reviewer_E13D · 2024-11-25
> >
> > Thank you for the detailed response and the new appendix. I'm happy with the revision. Raised the score to 8.

---

> > > ### Author Response · Authors · 2024-11-25
> > >
> > > Thank you for your response and updated score! We’re glad you are satisfied with the revisions!

---

### Official Review · Reviewer_C5fC · 2024-11-04

**Soundness:** 4
**Presentation:** 4
**Contribution:** 4
**Rating:** 8
**Confidence:** 3

**Summary:**

This paper presents several theoretical results about Neural ODEs, built around sufficient regularity of activations:
- It proves that "optimize then discretize" setting, built upon the use of classical ODE solvers, is well-posed under the assumption of Lipschitz-continuous activations (proposition 1).
- It proves that "discretize-then-optimize" setting, built as an Euler scheme discretization of the ODE by growing the depth of a residual neural network, is also well posed in the sense that the solution of the discretization scheme converges to the actual solution of the ODE in the infinite depth limit, if the activations and there derivatives are lipschitz-continuous (proposition 2). It has to be noted that authors gives explicit convergence rates.
- It shows that under the "discretize-then-optimize" setting with fixed depth L, the associated residual network converges in infinite width limit to a fixed kernel with explicit form, under the assumption of Lipschitz continuity of activations (proposition 3).
- It shows that under the assumption of Lipschitz continuity of activations, the infinite depth and width limits commute, implying that the neural ODE converges to a Gaussian process with covariance given as the infinite depth limit of the covariance of proposition 3 (theorem 1).
- Similar results are shown for the convergence of the NTK (proposition 4 and theorem 2)
- In section 5, authors show that the resulting limiting covariance of the gaussian process as well as NTK are Semi-positive definite (lemma 3, proposition 5, corrolary 1), resulting in derivation of explicit convergences rates (theorem 3).
Finally empirical evidences of the theoretical results are presented in section 6.

**Strengths:**

- precise, rigorously demonstrated theoretical results
- clear presentation
- extension of fundamental NTK results to neural ODEs, using classical ODE analytical tools (Picard theorem, Groenwald theorem etc).

**Weaknesses:**

- Typo : the index of the last h in equation 1 should probably be T instead of t
- Figure 1 : a,b,c should rather be in log-log scale to better appreciate the slope comparison with 1/L.

**Questions:**

I don't think it's quite right (or at least I don't understand exactly why, but I'm not a specialist in Neural ODEs) that the title of the paper focuses on the impact of activations and their regularities, whereas most of your results use these as hypotheses for theorem proving. In my opinion, you should assume a more theoretical title (or present more results comparing activations, but it seems to me that this is a slightly different job than what you present, which is amply sufficient from a theoretical point of view).

---

> ### Author Response · Authors · 2024-11-19
>
> We appreciate your thoughtful comments and your acknowledgment of the paper’s strengths, including precise and rigorous results, clear presentation, and the extension of NTK theory to Neural ODEs using classical analytical tools. Your feedback provides valuable insights to refine our work further. Below, we address your points systematically.
>
> > Typographical Error in Equation 1.
>
> Thank you for pointing out the typo in the index of  $h_t$  in Equation 1. You are correct that the last  $h$  should have the subscript  $T$  instead of  $t$. We have corrected this in the revised manuscript.
>
> > Log-Log Scale for Figure 1
>
> We really appreciate your suggestion to plot Figure 1 using a log-log scale to better highlight the  1/L  convergence rates. This is an excellent idea, and we have updated the figure accordingly to improve clarity and better emphasize the convergence rates. Please see the update in the revised version.
>
> > Title Suggestion and Focus on Activation Impact
>
> We understand your concern about the alignment between the title and the paper’s primary focus. The current title may overemphasize the impact of activation functions, potentially overshadowing the core theoretical contributions. To address this, we are open to revising the title in the final version to better reflect the paper’s focus and avoid any confusion.
>
> That said, the impact of activation functions is an integral part of our study. Due to space constraints, we included comprehensive experimental analyses on activation functions in the appendix. For instance, as shown in Figure 12 (Appendix), when the activation function is nonsmooth (e.g., ReLU), the gradient differences between the “discretize-then-optimize” and “optimize-then-discretize” frameworks fail to converge as depth increases. This suggests that continuous differentiability is not just sufficient but likely necessary for gradient alignment.
>
> Additionally, we investigated the SPD property of NNGP and NTK under polynomial activation functions, such as quadratic activation, in Figure 8 (Appendix). Our experiments show that quadratic activations still ensure SPD. Hence, the non-polynomial nonlinearity seems to be a sufficient but not necessary condition. Moreover, we also observed that the convergence is significantly slower compared to non-polynomial nonlinear activations like Softplus. This highlights the practical advantage of non-polynomial activations in accelerating training.
>
> In summary, while the theoretical contributions form the core of the main paper, the experimental results offer valuable insights into the implications of activation regularity. These findings complement the theoretical results and address broader questions about activation design. Thank you for highlighting this balance, and we will ensure the final version reflects this alignment more clearly.

---

> > ### Comment · Reviewer_C5fC · 2024-12-02
> >
> > Thank you for your reply and modifications.
> > I keep thinking this is a very good and interesting work !

---

### Author Response · Authors · 2024-11-19

We sincerely thank all reviewers for their thoughtful and constructive comments, which have significantly enhanced the quality of our work. Based on your invaluable feedback, we conducted additional experiments and expanded our discussions to address your suggestions. These updates are primarily included in the appendix, ensuring the core contributions of the paper remain clear while providing deeper insights into the discussed topics.

Specifically, we added the following sections:
- Appendix H.6 (Non-Smooth Activation Functions): This section presents experiments comparing the “Discretize-then-optimize” and “Optimize-then-discretize” frameworks for non-smooth activations, such as ReLU. We provide empirical evidence supporting our theoretical claims about gradient alignment challenges.
- Appendix H.7 (Sensitivity of “Optimize-then-discretize” to ODE Solvers): This section investigates the impact of different numerical solvers on training dynamics and gradient computation accuracy, providing practical guidance for solver selection in Neural ODE applications.
- Appendix I (Discussion on General Dynamic Form in Neural ODEs): Here, we extend the theoretical results to more general dynamic forms, including non-autonomous systems and broader nonlinear mappings, discussing the necessary conditions for the applicability of our results.
- Appendix J (Related Works): We provide a comprehensive review of related works, situating our contributions within the broader context of Neural ODEs and NTK theory. This section acknowledges foundational studies and highlights how our work builds on and extends existing research.

We are grateful for your thoughtful suggestions, which guided us to clarify and enrich our contributions further. These additions not only strengthen the theoretical and empirical aspects of our work but also provide practical insights for the community. Thank you for helping us improve the paper’s quality and impact.

---

### Meta-Review · Area_Chair_rcup · 2024-12-19

**Metareview:**

This paper studies the impact of activation functions on Neural ODE training. It shows that smoothness of the activations are essential for stabilizing training and guaranteeing a unique solution. In addition, the paper extends NTK theory to continuous-layered models, showing a well-defined NTK for Neural ODEs. Crucially, higher nonlinearity in the activation function not only preserves desirable NTK properties, but also accelerates convergence in practice. In essence, the choice of activation function significantly impacts Neural ODE stability and training efficiency.

The paper received positive reviews, all of which appreciated its theoretical rigor, clarity of presentation, and the significance of its findings, resulting in a final rating of 8 across the board. Reviewer C5fC noted a few typos and offered suggestions for improvement. Reviewer E13D highlighted the lack of demonstration for the "discretize-then-optimize" convergence issue, prompting the authors to conduct further experiments. This reviewer also inquired about the sensitivity of the "optimize-then-discretize" method to numerical ODE solvers, to which the authors responded with additional experiments comparing different solvers. Consequently, E13D raised their score. Reviewer Hphg inquired about convergence with polynomial activations and generalization beyond specific ODE forms, while also pointing out some missing references. The authors either provided clarification or revised the manuscript to address these points, leading the reviewer to increase their initial rating to 8.

In agreement with the reviewers, I find the paper to be well-written and the presented results on the effect of activation functions on neural ODE stability and training efficiency to be of interest to the community and recommend accept.

**Additional Comments On Reviewer Discussion:**

Reviewer C5fC noted a few typos and offered suggestions for improvement. Reviewer E13D highlighted the lack of demonstration for the "discretize-then-optimize" convergence issue, prompting the authors to conduct further experiments. This reviewer also inquired about the sensitivity of the "optimize-then-discretize" method to numerical ODE solvers, to which the authors responded with additional experiments comparing different solvers. Consequently, E13D raised their score. Reviewer Hphg inquired about convergence with polynomial activations and generalization beyond specific ODE forms, while also pointing out some missing references. The authors either provided clarification or revised the manuscript to address these points, leading the reviewer to increase their initial rating to 8.

---

### Decision · Program_Chairs · 2025-01-22

Accept (Oral)